# Giant lungfish genome elucidates the conquest of land by vertebrates

Axel Meyer[1,12,13 ✉], Siegfried Schloissnig[2,12], Paolo Franchini[1,12], Kang Du[3,4,12], Joost M. Woltering[1,12], Iker Irisarri[5,11], Wai Yee Wong[6], Sergej Nowoshilow[2], Susanne Kneitz[7], Akane Kawaguchi[2], Andrej Fabrizius[8], Peiwen Xiong[1], Corentin Dechaud[9], Herman P. Spaink[10], Jean-Nicolas Volff[9], Oleg Simakov[6,13 ✉], Thorsten Burmester[8,13 ✉], Elly M. Tanaka[2,13 ✉] & Manfred Schartl[3,4,13 ✉]

Lungfishes belong to lobe-fined fish (Sarcopterygii) that, in the Devonian period, 'conquered' the land and ultimately gave rise to all land vertebrates, including humans[1–3]. Here we determine the chromosome-quality genome of the Australian lungfish (*Neoceratodus forsteri*), which is known to have the largest genome of any animal. The vast size of this genome, which is about 14× larger than that of humans, is attributable mostly to huge intergenic regions and introns with high repeat content (around 90%), the components of which resemble those of tetrapods (comprising mainly long interspersed nuclear elements) more than they do those of ray-finned fish. The lungfish genome continues to expand independently (its transposable elements are still active), through mechanisms different to those of the enormous genomes of salamanders. The 17 fully assembled lungfish macrochromosomes maintain synteny to other vertebrate chromosomes, and all microchromosomes maintain conserved ancient homology with the ancestral vertebrate karyotype. Our phylogenomic analyses confirm previous reports that lungfish occupy a key evolutionary position as the closest living relatives to tetrapods[4,5], underscoring the importance of lungfish for understanding innovations associated with terrestrialization. Lungfish preadaptations to living on land include the gain of limb-like expression in developmental genes such as *hoxc13* and *sall1* in their lobed fins. Increased rates of evolution and the duplication of genes associated with obligate air-breathing, such as lung surfactants and the expansion of odorant receptor gene families (which encode proteins involved in detecting airborne odours), contribute to the tetrapod-like biology of lungfishes. These findings advance our understanding of this major transition during vertebrate evolution.

Lungfish (Dipnoi) share with land-dwelling vertebrates the ability to breathe air though lungs, which are homologous to our own. Since their discovery in the nineteenth century, lungfish have attracted scientific interest and were initially thought to be amphibians[6,7]. We now know that they are more closely related to tetrapods than to ray-finned fish. Of the extant lungfish species (of which there are only six), four live in Africa, one in South America and one (*N. forsteri*) in Australia. Lungfish appeared in the fossil record in the Devonian period, around 400 million years ago (Ma)[1]. Some scholarship has discussed lungfish as 'living fossils', because their morphology barely changed over millions of years: for example, >100-million-year-old fossils from Australia strongly resemble the surviving species (which represents one of the oldest known animal genera, discovered exactly 150 years ago)[2]. Owing to the ancestral characters (such as body shape, large scales and paddle-shaped fins) of *N. forsteri*, it resembles 'archetypal' extinct lungfish much more than the two other lineages of extant lungfish. The South American and, in particular, the African lungfish have almost completely lost their scales secondarily and have simplified their fin morphology into thin filaments, although they do show the alternating gaits that are typical of terrestrial locomotion.

Together with the coelacanths and tetrapods, lungfish are members of the Sarcopterygii (lobe-finned fish); however, owing to the short branch that separates these three ancient lineages it has remained difficult to resolve their relationships. Developments of powerful DNA sequencing and

[1]Department of Biology, University of Konstanz, Konstanz, Germany. [2]Research Institute of Molecular Pathology (IMP), Vienna, Austria. [3]Developmental Biochemistry, Biocenter, University of Würzburg, Würzburg, Germany. [4]The Xiphophorus Genetic Stock Center, Texas State University, San Marcos, TX, USA. [5]Department of Biodiversity and Evolutionary Biology, Museo Nacional de Ciencias Naturales (MNCN-CSIC), Madrid, Spain. [6]Department of Neuroscience and Developmental Biology, University of Vienna, Vienna, Austria. [7]Biochemistry and Cell Biology, Biocenter, University of Würzburg, Würzburg, Germany. [8]Institut für Zoologie, Universität Hamburg, Hamburg, Germany. [9]Institut de Génomique Fonctionnelle, École Normale Superieure, Université Claude Bernard, Lyon, France. [10]Faculty of Science, Universiteit Leiden, Leiden, The Netherlands. [11]Present address: Department of Applied Bioinformatics, Institute for Microbiology and Genetics, University of Goettingen, Goettingen, Germany. [12]These authors contributed equally: Axel Meyer, Siegfried Schloissnig, Paolo Franchini, Kang Du, Joost M. Woltering. [13]These authors jointly supervised this work: Axel Meyer, Oleg Simakov, Thorsten Burmester, Elly M. Tanaka, Manfred Schartl. ✉e-mail: axel.meyer@uni-konstanz.de; oleg.simakov@univie.ac.at; thorsten.burmester@uni-hamburg.de; elly.tanaka@imp.ac.at; phch1@biozentrum.uni-wuerzburg.de

computational methods enable us to now revisit long-standing evolutionary questions regarding these relationships using whole-genome-derived datasets with more robust orthology inferences than have hitherto been possible. Previous analyses using large transcriptomic datasets have tended to support the hypothesis that lungfish are the closest living relatives of tetrapods[4,5]. Lungfish are therefore crucial for understanding the evolution and preadaptations that accompanied the transition of vertebrate life from water to land. This major evolutionary event required a number of evolutionary innovations, including in respiration, limbs, posture, the prevention of desiccation, nitrogen excretion, reproduction and olfaction. Lungfish are known to have the largest animal genome (http://www.genomesize.com/search.php), but the mechanisms that led to and maintained their genome sizes are poorly understood. Therefore, the Australian lungfish might provide insights both into tetrapod innovations and evolution, and the structure of giant genomes.

## Genome sequencing, assembly and annotation

The largest animal genome sequenced so far is the 32-Gb[8] genome of the axolotl salamander (*Ambystoma mexicanum*). To overcome the challenges of sequencing and assembling the even-larger genomes of lungfish, we used long- and ultra-long-read Nanopore technology to generate 1.2 Tb in 3 batches: 601 Gb with an N50 read-length of 9 kb; 532 Gb with an N50 of 27 kb; and 1.5 Gb with an N50 of 46 kb, all from a juvenile Australian lungfish. We assembled these three batches into contigs using the MARVEL assembler[8] (Extended Data Fig. 1a, Methods). This yielded a 37-Gb assembly with an N50 contig size 1.86 Mb (Supplementary Table 1). To correct for insertions and/or deletions, gaps, single-nucleotide polymorphisms and small local misalignments in the primary assembly, we used 1.4-Tb DNA and 499.8-Gb RNA Illumina reads. The genome-correction DNA data—sequenced at more than 30× coverage—were used to estimate genome size through frequencies of *k*-mers (Extended Data Fig. 2). We ascertained the high completeness of the 37-Gb assembly by observing that 88.2% of the DNA and 84% of the RNA sequencing (RNA-seq) reads aligned to the genome, which gives an estimated total genome size of 43 Gb (about 30% larger than the axolotl[8]). This matches the *k*-mer value but is smaller than that predicted by flow cytometry (52 Gb[9]) and Feulgen photometry (75 Gb[10]).

Next, we scaffolded the contigs using 271-Gb chromosome conformation capture (Hi-C) Illumina PE250 reads to a chromosome-scale assembly with an N50 of 1.75 Gb (Extended Data Fig. 1d, Methods). We also used Hi-C data to detect misjoins, by binning Hi-C contacts along the diagonal and identifying points that were depleted of contacts (Extended Data Fig. 1e). The largest scaffolds correspond to the 17 macrochromosomes arms of the karyotype of *N. forsteri*. We also assembled all ten microchromosomes into single scaffolds (Supplementary Information).

We constructed a comprehensive multi-tissue de novo transcriptome assembly (BUSCO score of over 98% core vertebrate genes) using RNA extracted from the same individual lungfish. For annotation of protein-coding genes, we combined evidence from transcript alignments and homology-based gene prediction. This resulted in 31,120 high-fidelity gene models. We assessed the completeness of the genome assembly using the predicted gene set and the BUSCO pipeline, detecting 91.4% of core vertebrate genes (233 genes) and 90.9% of vertebrate conserved genes (2,586 genes) (Supplementary Table 2). We predicted 17,095 noncoding RNAs (ncRNAs), including 1,042 transfer RNAs (tRNAs), 1,771 ribosomal RNAs (rRNAs) and 3,974 microRNAs (Supplementary Table 3, Supplementary Information).

## Phylogeny of lungfish, coelacanth and tetrapods

Phylogenetic relationships among coelacanths, lungfishes and tetrapods have been debated[4,5,11]. We used Bayesian phylogenomics (Fig. 1) with 697 one-to-one orthologues for 10 vertebrates, with a complex

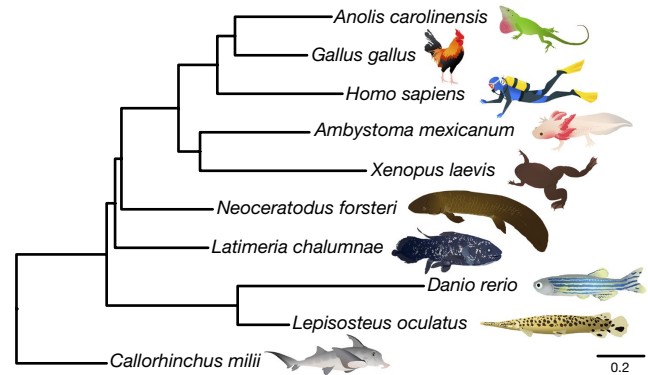

**Fig. 1 | Bayesian phylogeny based on 697 one-to-one orthologues.** This analysis used the CAT-GTR model in PhyloBayes MPI. All branches were supported by posterior probabilities of 1. The protein and a noncoding conserved genomic element datasets (Extended Data Fig. 3a) recovered identical and highly supported vertebrate relationships (posterior probability = 1.0 and 100% bootstrap for all branches). Scale bar is expected amino acid replacements per site.

mixture model that can overcome long-branch attraction artefacts[4] and also used noncoding conserved genomic elements (96,601 aligned sites) (Extended Data Fig. 3a). Both datasets unequivocally support lungfish[4,5] as the closest living relatives of land vertebrates, with which they shared a last common ancestor around 420 Ma (Extended Data Fig. 3b).

## Synteny conserved of macro- and microchromosomes

Lineage-specific polyploidy events are important evolutionary forces[12] that can also lead to genome expansions in lungfish[9,13]. Despite the massive genome expansion in lungfish relative to other animals, the lungfish chromosomal scaffolds strongly resemble the ancestral chordate karyotype (Fig. 2a, Extended Data Fig. 4 a, b). On the basis of 17 chordate linkage groups (CLGs)[14,15] and 6,337 markers mapped onto the lungfish genome, we uncovered conserved syntenic correspondence between lungfish chromosomes and CLGs (Fig. 2a). The ancestor of vertebrates underwent two rounds of whole-genome duplication. Lungfish also retained more ancient CLG chromosomal fusions through these two rounds of vertebrate duplication[15]. In lungfish, CLG fusions from before the second round of whole-genome duplications are preserved intact but substantially expanded (Fig. 2b). Almost all additional CLG fusions happened recently, as indicated by sharp syntenic boundaries (Fig. 2b). This, along with the 'vertebrate-typical' gene number of *N. forsteri*, confirms the diploidy of the genome.

All ten lungfish microchromosomes (inferred from karyotype[9] and our assembly (Extended Data Fig. 4)) could be homologized to the microchromosomes of chicken and gar (Fig. 2c, Extended Data Fig. 4c, d)—and even they mostly retained their co-linearity. This, along with the conservation of some microchromosomes in gar, chicken and green anole[15,16], suggests that microchromosomes may date back to the earliest vertebrates. The complete retention of microchromosomes in the massively expanded lungfish genome suggests that stabilizing selection maintains these ancestral units. In support of this, lungfish microchromosomes show—on average—higher gene densities and a lower density of long interspersed nuclear elements (LINEs), which are the major contributors to genome size (Extended Data Fig. 4b); this also suggests different expansion dynamics of vertebrate micro- and macrochromosomes.

## Hallmarks of the giant lungfish genome

A maximum likelihood reconstruction of the ancestral genome sizes of vertebrates shows 2 major independent genome-expansion events

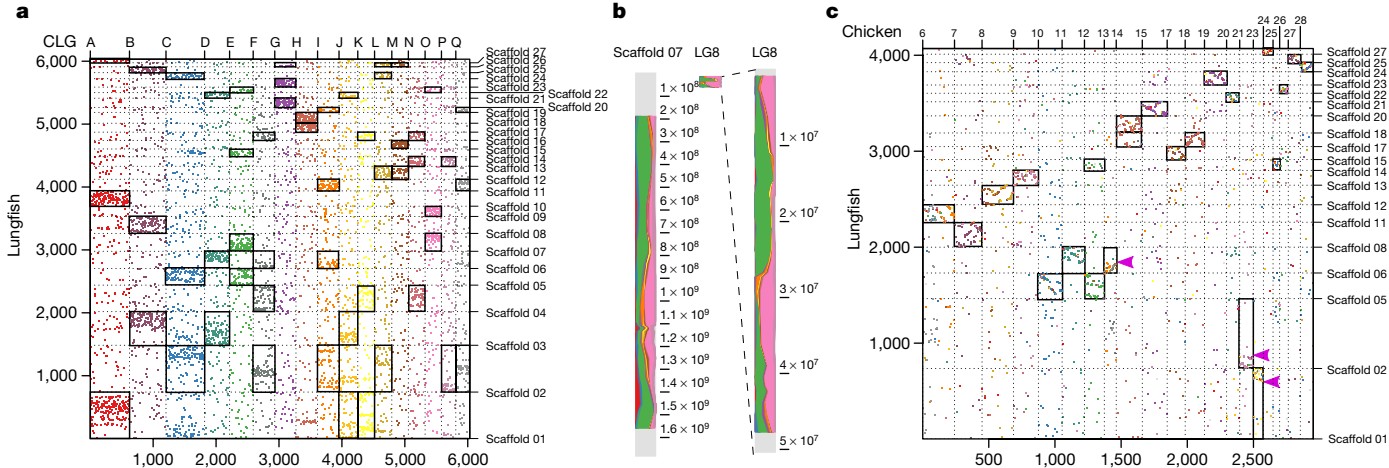

**Fig. 2 | Conserved synteny and chromosomal expansion in lungfish.**
**a**, Mapping of CLGs onto lungfish chromosomes. Orthologous gene family numbers are shown. Each dot represents an orthologous gene family, CLGs are as previously defined[15]. Scaffolds 01–17 represent lungfish macrochromosomes, and scaffolds 18–27 represent microchromosomes. Significantly enriched CLGs on lungfish chromosomes indicated by rectangles (for raw data, see Extended Data Fig. 4f). **b**, Expansion of homologous chromosomes in lungfish (left), compared to spotted gar (right) (here only LG8 is shown; the other chromosomes are in Extended Data Fig. 4a). Chromosomes are partitioned into bins and CLG content is profiled; chromosomal position is plotted next to each chromosome. LG8 in gar has a prominent jawed-vertebrate-specific fusion of the CLGs E and O, which is retained throughout the whole chromosome in lungfish (despite the latter being >30-fold larger). The small box in the middle is the unexpanded LG8 of spotted gar. **c**, Preservation of microchromosomes. Chicken microchromosomes are

plotted (for gar, see Extended Data Fig. 4d) along with their lungfish homologues with >50 orthologues. Scaffolds 01–17 represent lungfish macrochromosomes, and scaffolds 18–27 represent microchromosomes. For chicken, only microchromosomes are shown. Significantly enriched chicken microchromosomes on lungfish chromosomes indicated by rectangles (for raw data, see Fig. 4e). Most chicken microchromosomes are in one-to-one correspondence with lungfish, but some lungfish microchromosomes have recently been incorporated into macrochromosomes. These lungfish macrochromosomes (for example, scaffold 01 or scaffold 02) have significant association with both chicken macro- and microchromosomes. However, those fusions are recent in lungfish, because the positions of chicken orthologues are restricted to specific areas of the lungfish chromosomes, as is evident from the sharp syntenic boundaries (indicated by pink arrows on scaffold 01, scaffold 02 and scaffold 06). Silhouettes are from a previous publication[36]. Significances were determined by Fischer's exact test, *P* value ≤ 0.01.

in lungfish and salamander lineages (Extended Data Fig. 3c), initially at similar rates in both lineages (161–165 Mb per million years) but subsequently at slower rates in the Australian lungfish (about 39 Mb per million years), but possibly not in the other lineages of extant lungfishes. The genome expansion happened in early lungfishes (around 400–200 Ma), and slowed during the break up of Gondwana (from around 200 Ma to present) (Extended Data Fig. 3c). Independently, genome size increased in salamanders in two independent waves of DNA-repeat expansion (Fig. 3b, Extended Data Figs. 3c, 5). LINEs make up much of the recent genome growth of the lungfish (<15% divergence, around 9% (4 Gb), also in an earlier burst in lungfish but not axolotl) (Extended Data Fig. 5a). Because mobilized transposable elements can interrupt gene function, one might speculate that such bursts of activity of transposable elements might have caused novel gene functions.

Although syntenically highly conserved, the lungfish genome has undergone extreme expansion through the accumulation of transposable elements. We performed standard repeat-masking procedures on the 37-Gb genome assembly, which identified 67.3% (24.65 Gb) as repetitive (Fig. 3a, Supplementary Table 4). To our knowledge, this is the highest repetitive DNA content in a genome found in the animal kingdom. We tested whether the remaining 13 Gb of the genome have signatures of repetitiveness that are obscured by genome size by applying a second round of repeat annotation on the hard-masked genome. This revealed an additional 23.92% of repetitive DNA (Fig. 3a), which was mostly classified as 'unknown' (adding 11% to the unknown portion of repetitive DNA) or 'LINE' (8.5%) (Supplementary Tables 5, 6). In total, around 90% of the lungfish genome is repetitive, and it expanded in two waves (Fig. 3a, Extended Data Fig. 5).

To investigate whether transposable elements are still active, we analysed poly(A)-RNA-derived RNA-seq data that probably relates to proteins relevant for transposition activity. All major categories of transposable elements (1,106 out of 1,821 (60.7%)) were expressed

(Extended Data Fig. 6a). Transposable element families with higher copy numbers were also highly expressed in all three tissues we tested. This, and the finding of similar copies for many transposable element families, suggests that several types of transposable element remain active and contribute to the ongoing expansion of the lungfish genome. Identification of insertion polymorphisms between two, ideally relatively closely related lungfish species (such as *Protopterus* from Africa) are necessary to confirm transposable element activity. Apparently, the transposon silencing machinery did not adapt to reduce overabundant transposable elements by copy number expansion or structural changes (Supplementary Table 7).

The repeat landscape (proportions of major classes of transposable element) of lungfish resembles tetrapods (including axolotl), whereas the third extant sarcopterygian lineage (the coelacanths) is more 'fish'-like (Fig. 3b). The two largest animal genomes yet sequenced expanded through different temporal dynamics. Whereas long terminal repeat (LTR) elements are the most abundant class of transposable element (59%) in axolotl[8], LINEs (25.7%; mostly CR1 and L2 elements) dominate in lungfish (Extended Data Figs. 5, 6). These two retrotransposon classes belong to the same copy-and-paste (and not cut-and-paste) category, but propagate via different mechanisms[17]. Although global repeat compositions differ between lungfish and axolotl, the same LTR class affects their genic regions (Extended Data Fig. 6, Supplementary Information).

To further understand genome growth in lungfish, we compared the genome structure of *N. forsteri* with that of other genomes (Extended Data Figs. 6c, d, 7). Although compact genomes have small introns, intragenic noncoding regions usually increase with genome size[18]. The largest intron of the lungfish is 5.8 Mb (in the *dmbt1* gene) and average intron size is 50 kb as in axolotl, compared to 1 kb in fugu and 6 kb in human. Introns in the *N. forsteri* genome comprise about 8 Gb (21% of genome)—a similar proportion to that in human (21%), but half that

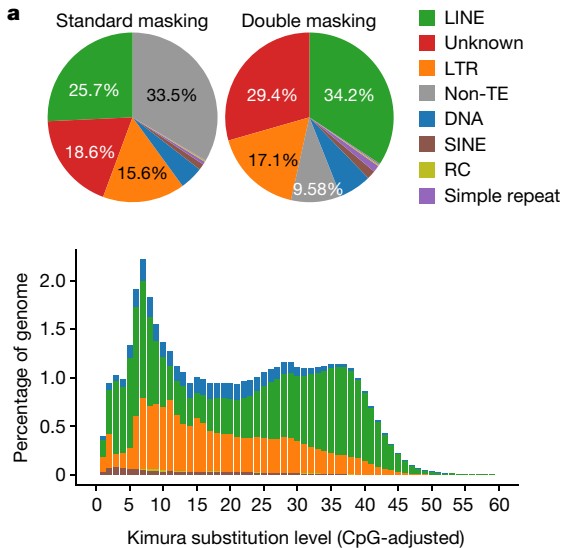

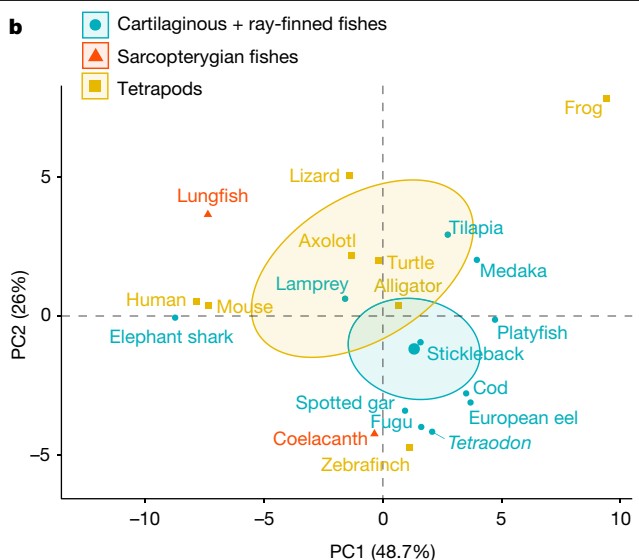

**Fig. 3 | Composition of repetitive elements in the lungfish genome. a**, The pie charts show overall composition of repetitive elements from unmasked assembly (first transposable element annotation) (left), together with the annotation from the hard masked genome (second transposable element annotation) (right). The bar chart shows the landscape of major classes of transposable elements. Kimura substitution level (%) for each copy against its consensus sequence used as proxy for expansion history of the transposable

elements. Older copies (old expansion) accumulated more mutations and show higher divergence from the consensus sequences. RC, rolling-circle transposons; SINE, short interspersed nuclear element; TE, transposable element. **b**, Principal component (PC) analysis of composition of repetitive elements (LTR, LINE, SINE, DNA and unknown, filtered by 80/80 rule) of vertebrates.

of fugu (40%). This suggests that similar mechanisms affect the genic and intergenic compartments, following expectations for genome size evolution[19].

In most genes, the first intron typically is the largest. The biological relevance of this remains unclear. The first introns in lungfish and axolotl are also much larger than downstream introns (Extended Data Fig. 7), which indicates that the relatively larger first introns in smaller genomes are probably not due to the space requirements of regulatory or structural motifs[20].

It has previously been suggested that the size of intragenic noncoding sequences and the extent of intron expansion are associated with organismal features (such as metabolic rate[18]) or functional categories of gene[8] (for example, developmental or nondevelopmental genes). Similar to axolotl[8], the introns in developmental genes in lungfish are smaller than in nondevelopmental genes ($P = 2.166 \times 10^{-8}$, Mann–Whitney $U$ test) (Supplementary Table 8).

## Genomic preadaptations in fish–tetrapod transition

Positive selection analysis uncovered 259 genes, many of which are related to oestrogen and categories related to female reproduction (Supplementary Information, Supplementary Table 9). We compared these rate dynamics (16,471 gene families) (Supplementary Tables 10,11), and found that in the lungfish lineage 24 families have contracted and 107 families have expanded—possibly related to evolutionary innovations.

## Air breathing and the evolution of lungs

All land-living vertebrates and adult lungfish are air breathers. The pulmonary surfactant protein B family of genes has expanded considerably in the lungfish genome. Surfactants are necessary components of the lipoprotein mixture that covers the lung surface and ensures proper pulmonary function. In lungfish, the number of surfactant genes increased to a number typical for tetrapods (2–3× more than in cartilaginous and bony fish) (Supplementary Table 12). This may

indicate an adaptation to air breathing in lungfish. We further investigated the expression of *shh*, which encodes an important regulator of lung development[21], during lungfish embryogenesis (Extended Data Fig. 8a). *shh* is strongly expressed in the developing lungs (embryos at stages 43–48), visualizing the development of the right-sided lung (*Neoceratodus* has a unilateral lung). This lung develops in a manner notably similar to those of amphibians[22]. Altogether, this highlights molecular signatures of lungs that were necessary for the conquest of land by sarcopterygians.

## Olfaction and evolution of the vomeronasal organ

We also noted expansions of genes involved in olfaction. The gene complement of receptors for airborne odorants (which is large and complex in tetrapods and small in fish) is considerably expanded in lungfish, whereas several receptor classes for waterborne odours have shrunk—in particular, zeta and eta receptors, which abound in teleost fishes (Supplementary Table 13). The vomeronasal organ (VNO) is present in most tetrapods[23,24], being linked to pheromone reception and expressing a large repertoire of vomeronasal receptor genes (particularly in amphibians). In *N. forsteri*, the vomeronasal receptor gene family—known from fish and even lampreys, although its function in these species is unknown—has expanded considerably. Lungfish possess a 'VNO primordium'[25]. The notable expansion of the vomeronasal receptor gene family (especially V2R genes) in *N. forsteri* (Supplementary Table 14) shows that the VNO is a tetrapod innovation, which emerged in the water-to-land transition.

## Lobed fins and evolution of terrestrial locomotion

Sarcopterygians have elaborated endochondral skeletons: lobed fins that are distally branched, forming digits that are suitable for substrate-based locomotion. Our analysis indicates sarcopterygian origins for 31 conserved tetrapod limb-enhancer elements[26] (Fig. 4a, Extended Data Fig. 8b). The hs72 (refs. [27,28]) enhancer (related to *sall1*) drives autopodal expression (Fig. 4b). We found *sall1* strongly

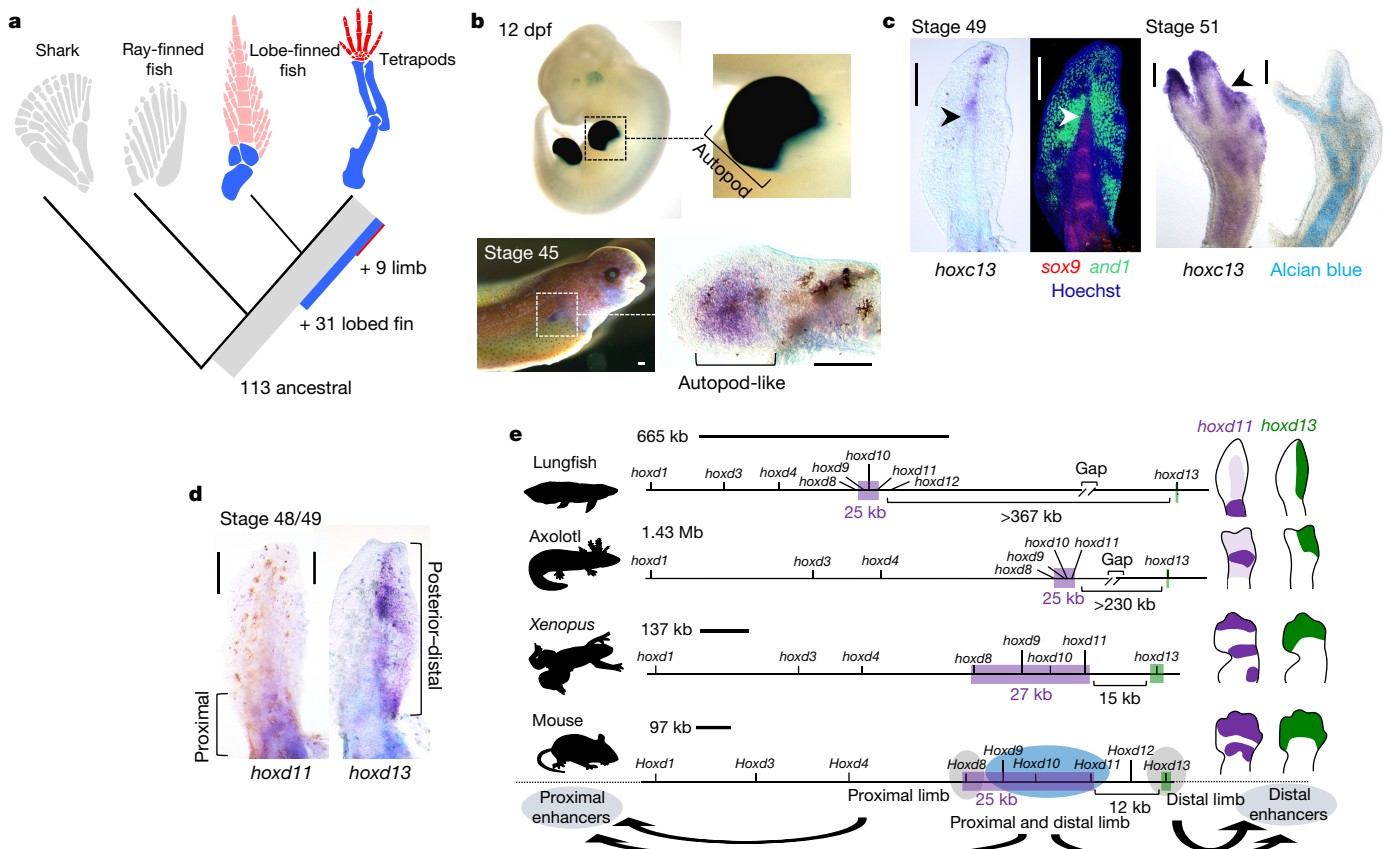

**Fig. 4 | Regulatory preadaptation of lobed fin and hoxd gene regulation.**
**a**, Analysis of 330 validated mouse and human limb enhancers shows deep evolutionary origin of the limb regulatory program; 31 enhancers are associated with the emergence of the lobed fin. **b**, The hs72 enhancer located near the *Sall1*[27,28] gene drives strong LacZ in mouse autopods (*n* = 3 out of 3 embryos, LacZ-stained embryos courtesy of VISTA enhancer[26]) (top). *sall1* is expressed in a similar autopodial-like domain in lungfish pectoral fins (*n* = 2 out of 2 fins) (bottom). dpf, days post-fertilization. **c**, Left, *hoxc13* is expressed in a distal lungfish area that overlaps with the central metapterygial axis (*sox9*) and fin fold (*and1*) (arrowheads) (*n* = 2 out of 2 fins). Right, similar expression present in axolotl limbs (arrowhead) (*n* = 4 out of 4 limbs), indicating a deep sarcopterygian origin for this expression domain. **d**, During lungfish fin development, *hoxd11* and *hoxd13* are expressed in mostly nonoverlapping proximal and posterior–distal fin domains (*n* = 4 out of 4 fins each). **e**, The lungfish hoxd cluster has increased in size compared to mouse and *Xenopus*,

but may be smaller than the axolotl hoxd cluster. In lungfish and axolotl expansion has occurred in the 3′ and 5′ regions of the cluster, whereas the central *hoxd8*, *hoxd9*, *hoxd10* and *hoxd11* region (lilac box) remained stable at approximately 25 kb, forming a separate 'minicluster'. The hoxd cluster is regulated by 3′ and long-range enhancers. *hoxd9*, *hoxd10* and *hoxd11* (lilac), and *hoxd13* (green), are subject to enhancer sharing[33] and co-expressed in the distal limb in mouse and *Xenopus*[33,37], whereas the increased genomic distance between *hoxd13* and *hoxd9*, *hoxd10* and *hoxd11* has disrupted their co-expression in the distal appendages of lungfish and axolotl. The preserved clustering of *hoxd8*, *hoxd9*, *hoxd10* and *hoxd11* can be explained by enhancer sharing 3′ of the cluster[33], which probably places constraints on their intergenic distances. Axolotl and *Xenopus hoxd11* and *hoxd13* after ref. [37]; lungfish *hoxd11* and *hoxd13* domains after ref. [36] and **d** (Supplementary Table 16 lists primers for probes). Scale bars, 0.2 mm. Silhouettes are from ref. [36].

expressed in lungfish embryos, in expression patterns similar to those reported for tetrapods[29] (Fig. 4b) but absent during zebrafish fin development[30]. Similar functions of *sall1* during mouse limb development[29] suggest that this gene contributed to the acquisition of sarcopterygian lobed fins already in lungfish.

## Hox clusters and te fin-to-limb transition

The 4 clusters of hox genes in *Neoceratodus* (hoxa, hoxb, hoxc and hoxd) comprise 43 genes (Extended Data Fig. 9); the presence of *hoxb10* and *hoxa14* in lungfish confirms their loss at the fish-to-tetrapod transition[11]. Our RNA-seq analysis of the expression of hox genes in the fins of larval *Neoceratodus* (Extended Data Fig. 8c) showed an unexpected expression of hoxc genes. The expression of hoxc genes in paired fins or limbs has previously been reported only for mammals[31], related to the nail bed. We observed *hoxc13* expression in axolotl limbs (Fig. 4c), but it was absent in the pectoral fins of ray-finned fish (Extended Data Fig. 8d). Transcript localization in *Neoceratodus* embryos showed expression of *hoxc13* in the distal fin (Fig. 4c). This indicates an early

gain of *hoxc13* expression in sarcopterygians, suggesting co-option of this domain in tetrapods to pattern dermal limb elements (such as nails, hooves and claws). Together with *sall1*, this demonstrates an early sarcopterygian origin of limb-like gene expression that was ready for tetrapod co-option, facilitating the fin-to-limb transition and colonization of the land.

## Hox cluster expansion versus regulation

Consistent with the overall genome expansion, the hox clusters of *Neoceratodus* are larger than in mouse, chicken and *Xenopus*, but have an uneven pattern of expansion (Extended Data Fig. 9). The clustering of hoxd genes results in their coregulation by enhancers 3′ and 5′ of the cluster, leading to co-expression of *hoxd9*, *hoxd10*, *hoxd11*, *hoxd12* and *hoxd13* in the distal appendages[32–35]. During fin development in *Neoceratodus*, expression of *hoxd11* is nearly absent from the *hoxd13* territory[36] (Fig. 4d) whereas in axolotl *hoxd9*, *hoxd10* and *hoxd11* are excluded from the *hoxd13* digit domain[37] (Extended Data Fig. 8e). Such apparent loss of coregulation between *hoxd13* and *hoxd9*, *hoxd10* and

*hoxd11* is similar to that caused by experimentally increased distances in the hoxd cluster[32], and suggests a disruption of enhancer sharing caused by the expansion of the intergenic regions between *hoxd11* and *hoxd13* (Fig. 4e). We performed additional analyses in mouse, *Xenopus*, lungfish and axolotl, which showed that—despite 5–10× differences in the size of the hoxd cluster—the region comprising *hoxd8*, *hoxd9*, *hoxd10* and *hoxd11* remained fixed at around 25 kb (Fig. 4e). This apparent constraint is probably due to sharing of enhancers located at the 3′ end of the cluster[33]. Altogether, this indicates that hoxd expansion has partially disrupted long-range enhancer sharing, but that—conversely— such mechanisms have locally also constrained intergenic distances.

We have sequenced and assembled at the chromosome level (Supplementary Table 15) the largest animal genome, and have substantiated the hypothesis that lungfish are the closest living relatives of tetrapods. Despite the unique genome expansion history of lungfish, genic organization and chromosomal homology is maintained even at the level of microchromosomes. Genomic preadaptations in lungfish for the water-to-land transition of vertebrates include a larger complement of lung-expressed surfactant genes, which might have facilitated the evolution of air-breathing through a lung. In addition, the number of VNO olfactory receptors (as well as other receptor gene families that permit detection of airborne odours) increased in the lineage that led to air-breathing lungfish. The uneven expansion of hox clusters demonstrates the regulatory consequences of, and constraints on, genome expansion. The evolutionary trajectory of limb enhancers shows an early-fish origin of the limb regulatory program, with important changes towards preadaptations for terrestrialization preceding the fin-to-limb transition. Gene expression domains that characterize the tetrapod limb, but which were previously presumed to be absent from fins (such as those of *sall1* and *hoxc13*), appeared in the lobe-finned lineage. Such novelties might have predisposed the sarcopterygians to conquer the land, demonstrating how the lungfish genome can contribute to a better understanding of this major transition in vertebrate evolution.

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

## Methods

No statistical methods were used to predetermine sample size. The experiments were not randomized and investigators were not blinded to allocation during experiments and outcome assessment.

### Biological materials

Biopsy material for DNA and RNA isolation was obtained from a juvenile Australian lungfish (*N. fosteri*) imported from Australia (CITES permit no.: PWS 2017-AU-000242). Owing to the immature status of the gonad, the sex could not be determined. The same specimen was used for genome sequencing (muscle), construction of the Hi-C library (spleen) and transcriptome sequencing of brain, gonad and liver. The second set of reads was generated from lungfish embryos (embryonic stage 52, GenBank accession numbers SRR6297462–6297470)[36]. Embryos were bred and collected under permit ARA 2009.039 at Macquarie University.

### DNA extraction, genome sequencing and assembly

High molecular weight (HMW) and ultra-HMW DNA was prepared by FutureGenomics and Nextomics, and sequenced using Nanopore technology (for statistics, see Supplementary Table 1).

gDNA for genome correction from snap-frozen lungfish muscle tissue (0.3 g) was isolated by a standard gDNA isolation protocol. Library preparation was performed using the Westburg NGS DNA library kit. The final library was excised by Pippin prep with 400-bp DNA size and sequenced (Illumina Nova-seq S2; PE150) at Vienna Bio Center NGS facility.

Hi-C library was generated as previously described[38,39], with modifications detailed in Supplementary Methods. Final Hi-C libraries were sequenced (Illumina Nova-seq SP; PE150) at Vienna Bio Center NGS facility.

### Genome assembly

Ninety-six million reads comprising 1.2 Tb were assembled using the MARVEL genome assembler[8]. We first aligned 1% of the reads against all other reads. From these 1%-against-all alignments, we derived information on the repetitive elements present in the reads and used transitive transfer to repeat-annotate all reads used in the assembly. Regions were deemed repetitive when the depth of the alignments for a given read exceeded the expected depth fourfold. Given the alignment of the 1% against every other read in the assembly, we then transferred the repeat annotation of the 1% using the alignments to the respective position in the aligned reads. Here, the assumption is that when region $(a, b)$ in read A aligns to $(c, d)$ in read B and for $a \le rb \le re \le b$ (in which rb and re are repetitive elements); this than can be mapped using the alignment to a corresponding region in B, which then can be tagged as repetitive as well. The final repeat-masking track covered 28.7% of the 1.2 Tb.

We then processed with an all-against-all alignment with repeat masking in place, yielding five billion alignments. On the basis of these alignments, we derived read qualities at 100-bp resolution, highlighting low sequencing quality regions in the reads. Using the alignments and the read qualities structural weaknesses (chimeric breaks, high-noise regions and other sequencing artefacts) in the reads were repaired (Supplementary Methods, Extended Data Fig. 10).

Repaired reads were then used for a new round of alignments, again with repeat masking, in place. After alignment, the default MARVEL assembly pipeline proceeded as shown in the included examples of the source distribution (Extended Data Fig. 1).

For the current MARVEL source code repository, see https://github.com/schloi/MARVEL. For sample execution scripts, see https://github.com/schloi/MARVEL/tree/master/examples.

### Scaffolding

We used an agglomerative hierarchical-clustering-based scaffolding approach using various normalizations (Extended Data Fig. 1). For details, see Supplementary Methods.

We created initial clusters by selecting the largest contigs with the fewest contacts between them, each contig serving as a single cluster. We then added contigs on the basis of unique assignability to clusters. This was followed by scaffolding the cluster separately, visual inspection of an approximate contact map derived during the scaffolding process and return of wrongly assigned contigs to the set of unassigned contigs. We created contact maps for all clusters and merged or split clusters on the basis of the signal within those. The process of assigning contigs, scaffolding, merging and splitting clusters was repeated until no more useful changes could be made to the clusters (Supplementary Table 15 for comparison of chromosome and scaffold DNA content).

For the public source code repository, see https://github.com/schloi/MARVEL/.

The MARVEL assembler and scaffolder has previously been used to obtain a chromosome-scale axolotl genome assembly, which has been validated in comparison to the previously published chromosome-scale meiotic scaffolding[40] and is available as previously described[41].

### Genome assembly correction

For correction of errors (insertions and/or deletions (indels), base substitutions and small gaps) remaining after the genome assembly, we applied a two-step procedure using DNA-sequencing and RNA-seq reads separately. In brief, we sequenced the same genomic DNA sample and generated 4,693,324,032 high-quality read pairs (2 × 150 bp) (30× coverage). Additionally, we used the RNA-seq reads from the de novo transcriptome assembly to correct indels, but not base substitutions, in transcribed regions (Supplementary Methods, Supplementary Results, Extended Data Fig. 10).

### Transcriptome assembly

RNA was isolated from brain, spinal cord, eyes, gut, gonad, liver, jaw, gills, pectoral fin, caudal fin, trunk muscles and larval fin. Libraries were constructed using NEBNext Ultra II Directional RNA library preparation kit (New England Biolabs), Illumina TruSeq RNA sample preparation kit (Illumina) or Lexogen Total RNA-seq Library Prep Kit V2 (Lexogen). Paired-end sequencing, performed with Illumina platforms, yielded approximately 1,150 million raw reads.

Raw reads, filtered and corrected using Trimmomatic v.0.36[42] and RCorrector v.1.0.2[43], were assembled using de novo and reference-guided approaches. For de novo assembly, only reads derived from poly(A)-selected RNA were processed using the Oyster River Protocol (ORP) v.2.2.8[44]. In brief, reads were assembled using Trinity v.2.8.4 ($k$-mer = 25), SPAdes v.3.13.3[45] ($k$-mer = 55), SPAdes ($k$-mer = 75) and Trans-Abyss v.2.0.1[46] ($k$-mer = 32). The four different assemblies were then merged using the OrthoFuser module[47,48] implemented in ORP. Completeness of the de novo-assembled transcriptome was assessed with BUSCO v.3[49] using core vertebrate genes and Vertebrata genes (vertebrata_odb9 database) in the gVolante webserver[50]. For reference-guided assembly, all reads were aligned to the *N. forsteri* genome (each sample independently) using the program HISAT2 v.2.1.0[51] (maximum intron length set to 3 Mb). The resulting mapping files were parsed by StringTie v.1.3.6[52] and transcripts reconstructed from each aligned sample were merged in a single consensus .gtf file.

### Repeats and transposable elements annotation

*Neoceratodus forsteri* repeat sequences were predicted using Repeat-Masker (v.4.0.7) with default transposable element Dfam database and a de novo repeat library constructed using RepeatModeler (v.1.0.10), including the RECON (v.1.0.8), RepeatScout (v.1.0.5) and rmblast (v.2.6.0), with default parameters. Transposable elements not classified by RepeatModeler were analysed using PASTEC (https://urgi.versailles.inra.fr/Tools/) and DeepTE[53]. Repeat sequences of *A. mexicanum* (AmexG_v3.0.0, https://www.axolotl-omics.org/) were predicted using the same approach. Repetitive sequences of *Anolis carolinensis* (GenBank accession GCA_000090745.2),

*Xenopus tropicalis* (GCA_000004195.4), *Rhinatrema bivittatum* (GCA_901001135.1), *Latimeria chalumnae* (GCA_000325985.2), *Lepisosteus oculatus* (GCA_000242695.1), *Danio rerio* (GCA_000002035.4) and *Amblyraja radiata* (GCF_010909815.1)) were identified using Dfam TE Tools Container (https://github.com/Dfam-consortium/TETools) including RepeatModeler (v.2.0.1) and RepeatMasker (v.4.1.0). To further examine the remaining intergenic sequences, we predicted repetitive sequences again using the same workflow on the genome hard-masked with repeats already predicted by RepeatMasker.

### Kimura distance-based distribution analysis and transposable-element-composition principal component analysis

Kimura substitution levels between the repeat consensus to its copies were calculated using a utility script calcDivergenceFromAlign.pl bundled in RepeatMasker. Repeat landscape plots were produced with the R script nf_all_age_plot.R and nf_am_rb_age_plots.R, using the divsum output from calcDivergenceFromAlign.pl. Principal component analysis on repetitive element composition was performed in R (v.3.6) using factoextra package (v.1.0.6). Repetitive element compositions (SINE, LINE, DNA, LTR and unknown) were calculated from the predicted libraries. Repetitive element copies were filtered by the 80/80 rule (equal or longer than 80 bp, equal or more than 80 per cent identity compared with the consensus sequence). Repetitive element composition of other vertebrates was obtained from ref. [54].

### Transposable element composition by gene length and LTR family analysis

Repetitive sequence composition within genes (grouped by length) was examined by calculating the coverage (in bp) of each class of repetitive element, normalized by gene length. We examined LTR family enrichment in genic regions. All calculations and visualizations are summarized in the jupyter notebook file te_general_analysis.ipynb. All python scripts ran on Python ≥3.7 and used the package gffutils (v.0.10.1) (https://github.com/daler/gffutils) to operate large gene and repetitive element annotation files from large genomes. Plots were generated using Plotly Python API (https://plot.ly).

### Transposable element content in genic regions

Intron position was calculated by GenomeTools (v.1.5.9). The sum of the coverage of the repetitive element (for example, LINE CR1) was normalized by the length of the genic feature considered (Supplementary Table 17) (for example, intron 8) using python script te_cnt_class.py.

### Transposable element expression

Transposable element expression was assessed with TEtools[55] on gonad, brain and liver poly(A)-RNA data. Because of the large size of lungfish genome, a random subset of 10% of all transposable element copies was used. Transposable-element-family counts were normalized by transposable-element-family consensus length (count × $10^6$/consensus length) and library size. Normalized counts were plotted against transposable-element-family copy numbers.

### Annotation of protein-coding genes

Protein-coding genes were predicted by combining transcript and homology-based evidence. For transcript evidence, assembled transcripts (as described in 'Transcriptome assembly') were mapped to the assembly using Gmapl v.2019-05-12[56] and the gene structure was inferred using the PASA pipeline v.2.2.3[57]. Expression of each transcript was measured using the whole RNA-seq dataset (as described in 'Transcriptome assembly') and the pseudoalignment algorithm implemented in Kallisto v.0.46.1[58]. For homology evidence, we collected manually curated proteins from UniProtKB/SWISSPROT database (UniProtKB/Swiss-Prot 2020_03)[59] and protein sequences of *Callorhinchus milii*, *L. chalumnae*, *L. oculatus* and *X. tropicalis* from

Ensembl (http://www.ensembl.org) and NCBI (https://www.ncbi.nlm.nih.gov/genome), and aligned them to the repeat-masked assembly using Exonerate v.2.2[60]. Transcript and homology-based evidence were then combined by prioritizing the former (homology-based predicted genes were removed when intersecting a gene predicted using the reconstructed transcripts). The combined gene set was then processed by two rounds of 'PASA compare' to add untranslated region (UTR) annotations and models for alternatively spliced isoforms. Low-quality gene models were removed by applying three further quality-filtering steps in an iterative fashion: (1) single-exon genes were retained only when no similarity with exons of multi-exonic genes was found (similarity was identified with the glsearch36 module implemented in the FASTA v.36.3.8g package[61] with *e*-value cut-offs of $1 \times 10^{-10}$ and identity cu-toffs of 80); (2) genes intersecting repeat elements were removed when >50% (single-exonic genes) and >90% (multi-exonic genes) were covered by repeats; and (3) genes with internal stop codon(s) were removed. The completeness of the predicted protein-coding gene set was assessed with BUSCO using the core vertebrate genes and the Vertebrata genes (vertebrata_odb9 database) in the gVolante webserver.

To annotate the lungfish hox clusters, hox genes were first identified using BLAST with vertebrate orthologues as query (Supplementary Methods).

### Annotation of ncRNA genes

ncRNA genes were annotated using tRNAscan-s.e. v.2.0.3[62] and Infernal v.1.1.2[63]. The same procedure was applied to the genomes of the nine other focal species. For each of the ten species, the corresponding microRNA sets (obtained from miRBase v.22[64] database) were used to predict microRNA target sites on 3′ UTRs of canonical mRNAs using miRanda v.3.3[65]. Further details are provided in Supplementary Information.

### Annotation of conserved noncoding elements

**Whole-genome alignments.** The masked versions of the genome assemblies of the ten species used for the phylogenetic tree (Fig. 1) were used to build a whole-genome alignment with the human genome as reference (ten-way whole-genome alignment). In brief, each pairwise alignment was constructed using Lastz v.1.03.73[66] and further processed using UCSC Genome Browser tools[67]. Multiple alignments were generated using as input the nine pairwise alignments in .maf format with the programs Multiz v.11.2 and Roast.v.3.0[68].

**Detection of conserved elements.** The phylogenetic hidden Markov model (phylo-HMM) implemented in phastCons[69] (run in rho-estimation mode) was used to predict a consistent set of conserved genomic elements in the ten-species whole genome alignment. A neutral model of substitutions was calculated using phyloFit[69] with the general reversible substitution model from fourfold degenerate sites. Raw conserved noncoding elements (CNEs) detected by phastCons were merged when their distance was <10 bp, and subsequently CNEs <50 bp were removed. Protein-coding CNEs and those intersecting ncRNA genes, pseudogenes, retrotransposed elements and antisense genes (annotated in the human genome) were removed.

### Expansion of the genome in intergenic regions

The final filtered set of CNEs was used to investigate expansion of intergenic spaces. We compared the distance of nonexonic elements that are conserved in lungfish and three tetrapods (human, chicken and axolotl). To obtain informative CNE pairs, we selected those CNEs that: (1) were present in all four genomes; (2) were located in intergenic space; (3) were located in the same contig or chromosome in each species; and (4) did not have a gene in between them. The remaining set of 223 CNE pairs were used to calculate intergenic distance and region-specific expansion of the lungfish genome (Supplementary Table 18).

## Lineage-specific acceleration of CNEs

The program phyloP was used to test each CNE for lineage-specific accelerated evolution[69,70] in the lungfish branch. A likelihood ratio test to compute the P value of acceleration with respect to a neutral model of evolution for each of the conserved elements in the alignment was used. CNEs showing false-discovery-rate (FDR)-adjusted P values < 0.05 were considered significantly accelerated. The accelerated CNEs were checked for overlap with a set of 1,978 experimentally validated human and mouse noncoding fragments with gene enhancer activity (data from 'VISTA Enhancer Browser'[26]) (Supplementary Table 19).

## Macrosynteny analysis

Amphioxus annotation[15] was mapped onto the lungfish assembly using TBLASTN. The CLG identity of amphioxus genes was used to determine CLG composition of lungfish chromosomal scaffolds. Dot plots were done using scripts available at https://bitbucket.org/viemet/public/src/master/CLG/.

## Comparison of intron size

Intron size was compared between lungfish, axolotl, human and fugu for one-to-one orthologues. Intron sizes of each gene were calculated from the .gff files of each genome. Genes without a start codon were removed to avoid the pseudo-intron order. The intron size was compared first in absolute bp, then in the value normalized by each genome size (lungfish, 44,032 Mb; axolotl, 32,768 Mb; human, 3,000 Mb; and fugu, 400 Mb).

## Orthology assignment

Protein sequences of *A. carolinensis*, *C. milii*, *D. rerio*, *Gallus gallus*, *Homo sapiens*, *L. chalumnae* and *L. oculatus* were downloaded from Ensembl (Lepisosteus_oculatus), and of *Xenopus laevis* from NCBI (https://www.ncbi.nlm.nih.gov/genome). Sequences of *A. mexicanum* were taken from ref. [41]. In cases of alternative splicing, we kept the longest sequence for the gene. All proteins were pooled together as the query and database for an all-versus-all BLASTP. From the result, we determined an *H*-score between each two proteins as representative of the distance for sequence similarity[71], and launched a clustering using Hcluster_sg[72]. Finally, for each cluster, a gene tree was built using TreeBeST and orthology between genes was assigned.

## Phylogeny inference

The phylogeny was inferred using the set of 697 orthologous proteins. Individual loci were filtered with PREQUAL[73], aligned with MAFFT ginsi[74] and highly incomplete positions (>80%) trimmed with BMGE[75]. Orthology was ensured by manual inspection of maximum likelihood gene trees (IQ-TREE) and alignments (MAFFT ginsi) for loci showing high branch-length disparity, and five individual sequences were removed. Loci were concatenated into a final matrix containing 10 taxa and 697 loci, totalling 383,894 aligned amino acid positions, of which 208,588 (54%) were variable. Phylogeny was inferred using PhyloBayes MPI v.1.7[76] under the site-heterogeneous CAT-GTR model, shown to avoid phylogenetic artefacts when reconstructing basal sarcopterygian relationships[4]. Two independent Markov chain Monte Carlo chains were run until convergence (>4,000 cycles), assessed a posteriori using PhyloBayes' built-in functions (maxdiff = 0, meandiff = 0, ESS >100 for all parameters, after discarding the first 25% cycles as burn-in). Post-burn-in trees were summarized into a fully resolved consensus tree with posterior probabilities of 1 for all bipartitions.

## Whole-genome-alignment-based phylogeny

The ten-species whole genome alignment was processed by MafFilter v.1.3.0[77] to keep only alignment blocks >300 bp that were present in all species. Filtered noncoding blocks were then concatenated and exported in .phylip format. Poorly aligned regions were removed using trimAl v.1.2 with option '-automated1'. The final dataset (99,601 aligned nucleotides) was used to reconstruct the phylogeny with RAxML v.8.2.4 under the GTRGAMMA model and 1,000 bootstrap replicates.

## Genome size evolution

Genome size evolution was modelled by maximum likelihood using the 'fastAnc' function in the phytools R package[78]. We used a time-calibrated tree representing all major jawed vertebrate lineages obtained from the phylotranscriptomic tree of ref. [5]; ages are a genome-wide estimates across 100 time-calibrated trees inferred from 100 independent gene jack knife replicates inferred in PhyloBayes v.4.1[79] under a log-normal autocorrelated clock model with 16 cross-validated fossils as uniform calibrations with soft bounds, the CAT-GTR substitution model and a birth–death tree prior. Genome size data (haploid DNA content or *c*-value) were obtained from ref. [80]. Genome size estimates were averaged per species (if several were available) and, in six species, genome size was approximated as the average of closely related species within the same genera. For *Neoceratodus*, the *k*-mer-based estimation was used (43 Gb; *c*-value = 43.97 pg). Ancestral genome sizes were used to calculate the rates of genome evolution for selected branches.

## Molecular clock analyses

Divergence times were inferred with a relaxed molecular clock with autocorrelated rates, as implemented in MCMCTree within the PAML package v.4.9h[81]. A total of six fossil calibrations were used as uniform priors[82]. For further details, see Supplementary Methods.

## Dynamics of gene family size

CAFE[83] was used to infer gene birth and death rates (lambda) and retrieve gene families under significant dynamics. As input, we took the species tree with divergence time from the output of MCMCTree and the results of gene clusters from Hcluster_sg. Each gene cluster was deemed to be a gene family. We ran CAFE under a model in which a global lambda was set across the whole tree. To symbolize each gene family, we took the longest member as representative and BLAST-searched with diamond[84] against SWISSPROT and NR databases. The best hit from both was retained.

To compare the repertoire of olfactory receptors, taste receptors and pulmonary surfactant proteins across all studied species, we followed the same procedure for each species. First, we collected sequences of olfactory receptors, taste receptors and pulmonary surfactant proteins from Swiss-Prot and NR database as query. For sequences from NR database, we only kept those with identifiers starting with 'NP_', which are supported by the RefSeq eukaryotic curation group. Second, we mapped the query set to each genome using Exonerate in server model (maxintron set to six million for lungfish and axolotl). The alignment was extended to start and stop codon when possible. Third, we BLAST-searched all retrieved sequences to NR database and removed those with a best hit that was not an olfactory receptor, taste receptor or pulmonary surfactant. The final result sequences had alignment coverage ranging from 32% to 100% (first quartile 95%), and percentage of identity from 17% to 100 (first quartile 62%) to its query.

Following a previous study[85], we separated the final sequences into three categories on the basis of their alignment to their query: (1) pseudogene, sequences with premature stop codon or frameshift; (2) truncated gene, sequences without premature stop codon and frameshift but broken open reading frame (ORF) (start or stop codon missing); and (3) intact gene, sequences with intact ORF.

## Positive selection analysis

Two models were calculated. Model 1 was used to find genes positively selected in lungfish and model 2 was used for genes commonly positively selected in tetrapods and lungfish. Genomes included were *N. forsteri* and *A. mexicanum* from this study, and the Ensembl genomes *D. rerio* (Danio_rerio.GRCz11), *A. carolinensis* (Anolis_carolinensis.

AnoCar2.0), *L. oculatus* (Lepisosteus_oculatus.LepOcu1), *L. chalum-nae* (Latimeria_chalumnae.LatCha1), *C. milii* (Callorhinchus_milii. Callorhinchus_milii-6.1.3), *X. tropicalis* (GCF_001663975.1_*Xenopus*_lae-vis_v2), *G. gallus* (Gallus_gallus.GRCg6a) and *H. sapiens* (Homo_sapiens. GRCh38). The *X. tropicalis* genome (GCF_001663975.1_*Xenopus*_lae-vis_v2) was downloaded from NCBI. Protein and cDNA files from all species were downloaded. To identify orthologous proteins, all pro-tein sequences were compared to lungfish using Inparanoid[86] (default settings). To match protein and cDNA, sequences were searched by TBLASTN and only 100% hits were kept. Codon alignments for the pro-tein and cDNA sequence pairs were constructed using pal2nal v.14[87]. Resulting sequences were aligned by MUSCLE[88] (option: -fastaout) and poorly aligned positions and divergent regions of cDNA were eliminated by Gblocks v.0.91b[89] (options: -b4 10 -b5 n --b3 5 --t = c). An in-house script was used to convert the Gblocks output to PAML format.

As a phylogenetic tree, we took the species tree with divergence times from MCMCTree as input for detection of positive selection with *C. milii* as outgroup. For the phylogenetic analyses by maximum likelihood, the 'Environment for Tree Exploration' (ETE3) toolkit[90]—which auto-mates CodeML and Slr analyses by using preconfigured evolutionary models—was used. For detection of genes under positive selection in lungfish, we compared the branch-specific model bsA1 (neutral) with model bsA (positive selection) using a likelihood ratio test (FDR ≤ 0.05). To detect sites under positive selection, naive empirical Bayes prob-abilities for all four classes were calculated for each site. Sites with a probability > 0.95 for either site class 2a (positive selection in marked branch and conserved in rest) or 2b (positive selection in marked branch and relaxed in rest) were considered. Two models were calculated. In model 1, only the branch for lungfish was marked; in model 2, all tetra-pods and lungfish were marked for positive selection.

Functional clustering was done with IPA (Qiagen, www.qiagenbio-informatics.com/products/ingenuity-pathway-analysis) and DAVID (https://david.ncifcrf.gov/home.jsp) using human homologues with default settings.

## In situ hybridization
In situ hybridization was performed as previously described[36,91], with modifications (Supplementary Methods).

## hox gene RNA-seq analysis
hox gene RNA-seq analysis was performed on a stage-52 lungfish larva RNA-seq dataset (SRR6297462–SRR6297470)[39] (Supplementary Meth-ods).

## Limb enhancer analysis
Three hundred and thirty nonredundant VISTA enhancer elements[26,92] were searched by BLASTN against *X. laevis*, *X. tropicalis*, *Nanorana park-eri*, axolotl, reedfish, sterlet, gar, elephant shark, coelacanth (LatCha1) and *Neoceratodus* genomes to determine conservation (Supplementary Methods).

## Reporting summary
Further information on research design is available in the Nature Research Reporting Summary linked to this paper.

## Data availability
Data are available from NCBI Bioproject under accession code PRJNA644903. All other relevant data are available from the corre-sponding authors upon reasonable request.

## Code availability
Custom code has been deposited at https://github.com/labtanaka/ meyer_lungfish. For the current MARVEL source code repository,

see https://github.com/schloi/MARVEL. For sample execution scripts, see https://github.com/schloi/MARVEL/tree/master/examples.

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

**Acknowledgements** We thank the late J. Clack and R. L. Carroll for their contribution to our understanding of the water–land transition of vertebrates. This work was supported by the German Science Foundation (DFG) through a grant to A.M., T.B. and M.S. (Me1725/24-1, Bu956/23-1, Scha408/16-1) and to J.M.W. (Wo2165/2-1), and core funding from the IMP to E.M.T. J.-N.V. and M.S. were supported by a joint grant of the French Research Agency (ANR Evobooster) and DFG (SCHA408/13-1). I.I. was supported by the Spanish Ministry of Economy and Competitiveness (MINECO) (Juan de la Cierva-Incorporación fellowship IJCI-2016- 29566) and the European Research Council (grant agreement no. 852725; ERC-StG 'TerreStriAL' to J. de Vries (University of Göttingen)). W.Y.W. and O.S. were supported by the Austrian Science Fund grants P3219 and I 4353. W.Y.W. is supported by Croucher Scholarships for Doctoral Study. A.K. was supported by a fellowship from the Japanese Society for the Promotion of Science (JSPS) postdoctoral fellowship for Overseas Researchers Program. We thank D. Ocampo Daza (http://www.egosumdaniel.se/) for generously sharing his vertebrate illustrations, J. Joss and P. Sordino for the gift of lungfish embryos, and L. Pennacchio for Vista enhancer images.

**Author contributions** A.M., T.B. and M.S. conceived the study and coordinated the work. A.M. and M.S. wrote the manuscript with contributions from all other authors. S.S. performed genome assembly into contigs and Hi-C scaffolding. P.F. undertook transcriptome analysis, annotation and CNE analyses. K.D. performed genome annotation, analysis of gene family dynamics and genome expansion. J.M.W. analysed and annotated hox clusters, and performed embryonal RNA-seq and in situ hybridization. I.I. generated phylogenetic analyses, and molecular clock and ancestral character state reconstruction. W.Y.W. performed repeat and syntenic analysis. S.N. undertook genome correction and initial transcript alignment. S.K. performed positive selection analysis. A.K. undertook Hi-C library preparation and library preparation for genome correction. A.F. performed transcriptome generation. P.X. annotated ncRNAs. C.D. and J.-N.V. performed transposon and repeat analysis. H.P.S. contributed resources. O.S. performed syntenic analyses. E.M.T. supervised Hi-C and genomic sequencing, and analysed data.

**Competing interests** The authors declare no competing interests.

**Additional information**
**Correspondence and requests for materials** should be addressed to A.M., O.S., T.B., E.M.T. or M.S.

**a**

Create seed clusters

↓

→ Assign contigs to best-fit clusters → Create scaffold graph

Scaffold clusters → Score edges and find most likely neighbours

Visual inspection ← Merge unambiguous chains of neighbouring nodes

↓

Merge/Split clusters

↓

Scaffolded contigs

**b**

Nx

(plot: Contig length (Mbp) vs x)

**c**

Nx

(plot: Contig length (Mbp) vs x)

**d**

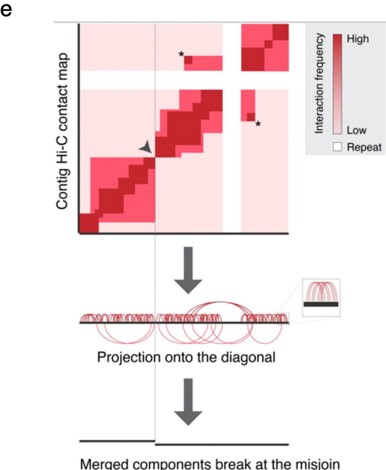

**e**

Contig Hi-C contact map

Interaction frequency: High / Low / Repeat

↓

Projection onto the diagonal

↓

Merged components break at the misjoin

**Extended Data Fig. 1** | See next page for caption.

**Extended Data Fig. 1 | Schematic overview of the scaffolding procedure.**
**a**, Scaffolding consists conceptually of two nested loops. The inner loop, depicted on the right, takes a list of contigs, their contact information and iteratively performs a global agglomerative clustering until convergence or until no more contigs can be joined. This loop is nested in the main procedure, which takes as input a list of seed contigs, assigns contigs these initial clusters, scaffolds these and allows for visual inspection and merging or splitting of the clusters. **b**, $N(x)$ plot of the assembled contigs. On the $y$ axis the contig length is shown, for which the collection of all contigs of that length or longer covers at least $x$ per cent ($x$ axis) of the assembly. **c**, $N(x)$ plot of the scaffolded genome. On the $y$ axis, the contig length is shown for which the collection of all scaffolds of that length or longer covers at least $x$ per cent ($x$ axis) of the assembly. **d**, Hi-C contact heat map of the scaffolded portion of the lungfish genome assembly, ordered by scaffold length. Blue boxes indicate the scaffold boundaries. The four largest scaffolds represent both chromosome arms on a single scaffold. Remaining scaffolds are split into chromosome arms or represent microchromosomes. **e**, Schema illustrating the contig misjoin detection process. Hi-C contacts are binned along the diagonal. Points that are not crossed by a sufficient number of contacts are deemed potential misjoins and are thus separated (dotted line).

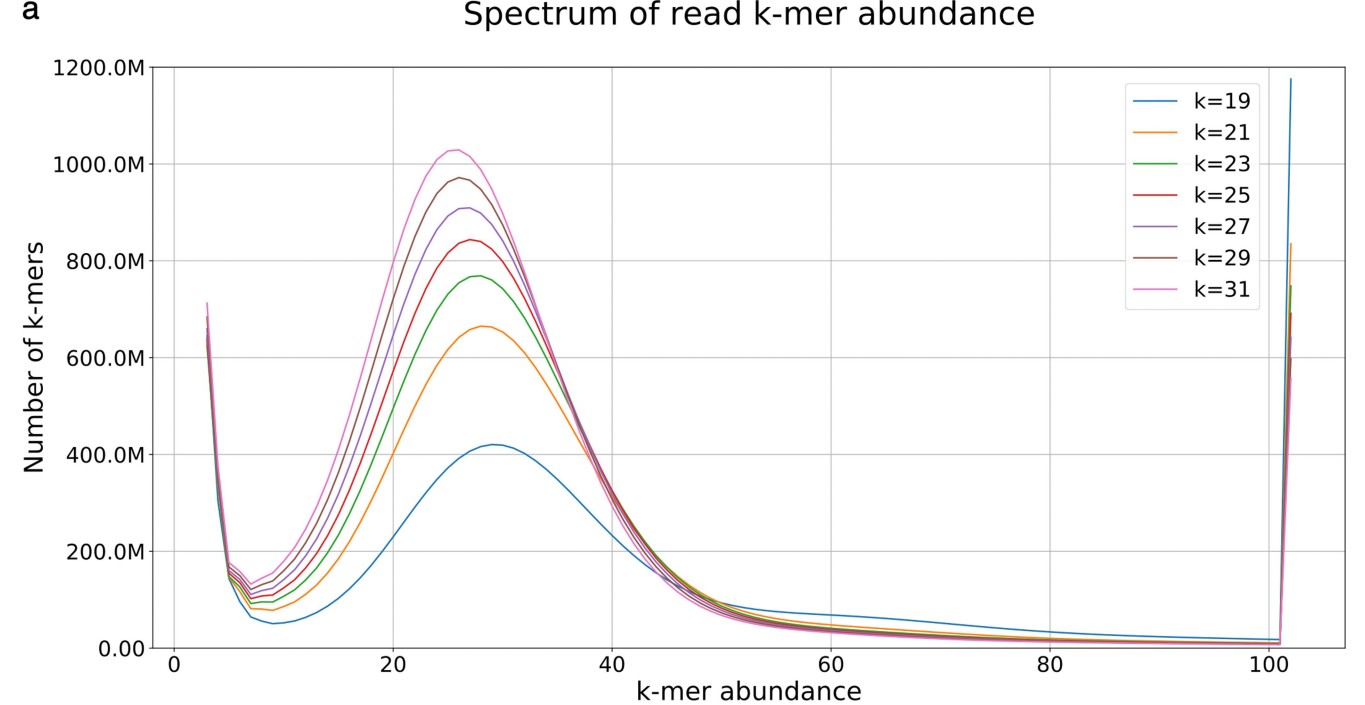

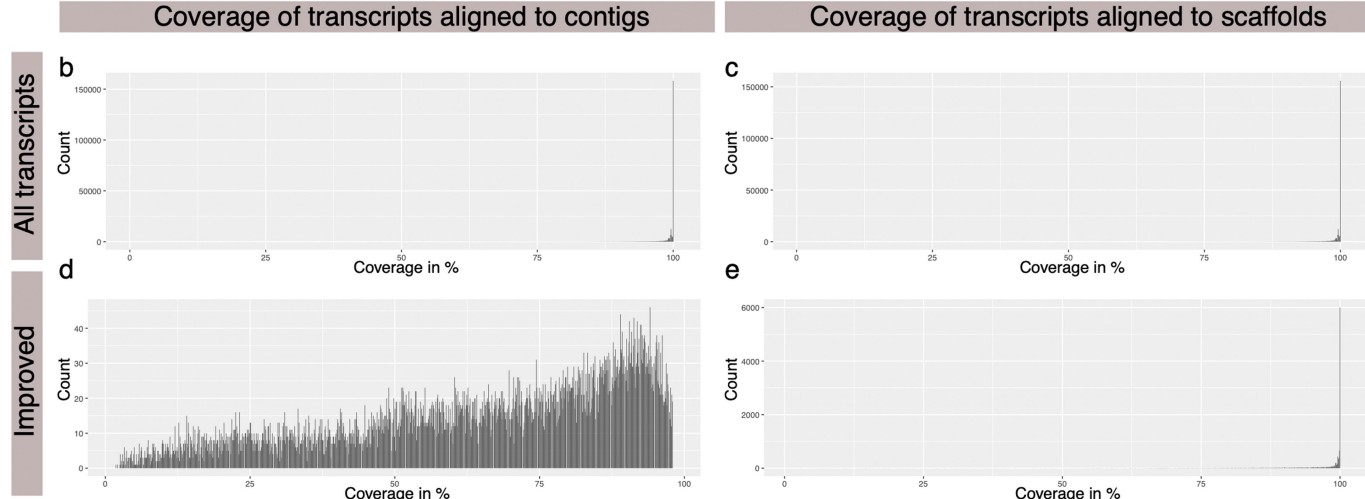

**Extended Data Fig. 2 | k-mer frequency analysis and transcript coverage by genomic sequences. a**, The Illumina dataset was used to generate the spectra of *k*-mer abundances using seven *k*-mer sizes. **b**–**e**, Transcript coverage by genomic sequences. **b**, Histogram of the proportion of all transcript lengths covered by the alignment to contigs. **c**, Histogram of the proportion of all transcript lengths covered by the alignment to scaffolds. **d**, **e**, Histogram of the proportion of the transcript lengths covered by the alignment to contigs (**d**) or to scaffolds (**e**) of those transcripts with alignments that were improved after scaffolding.

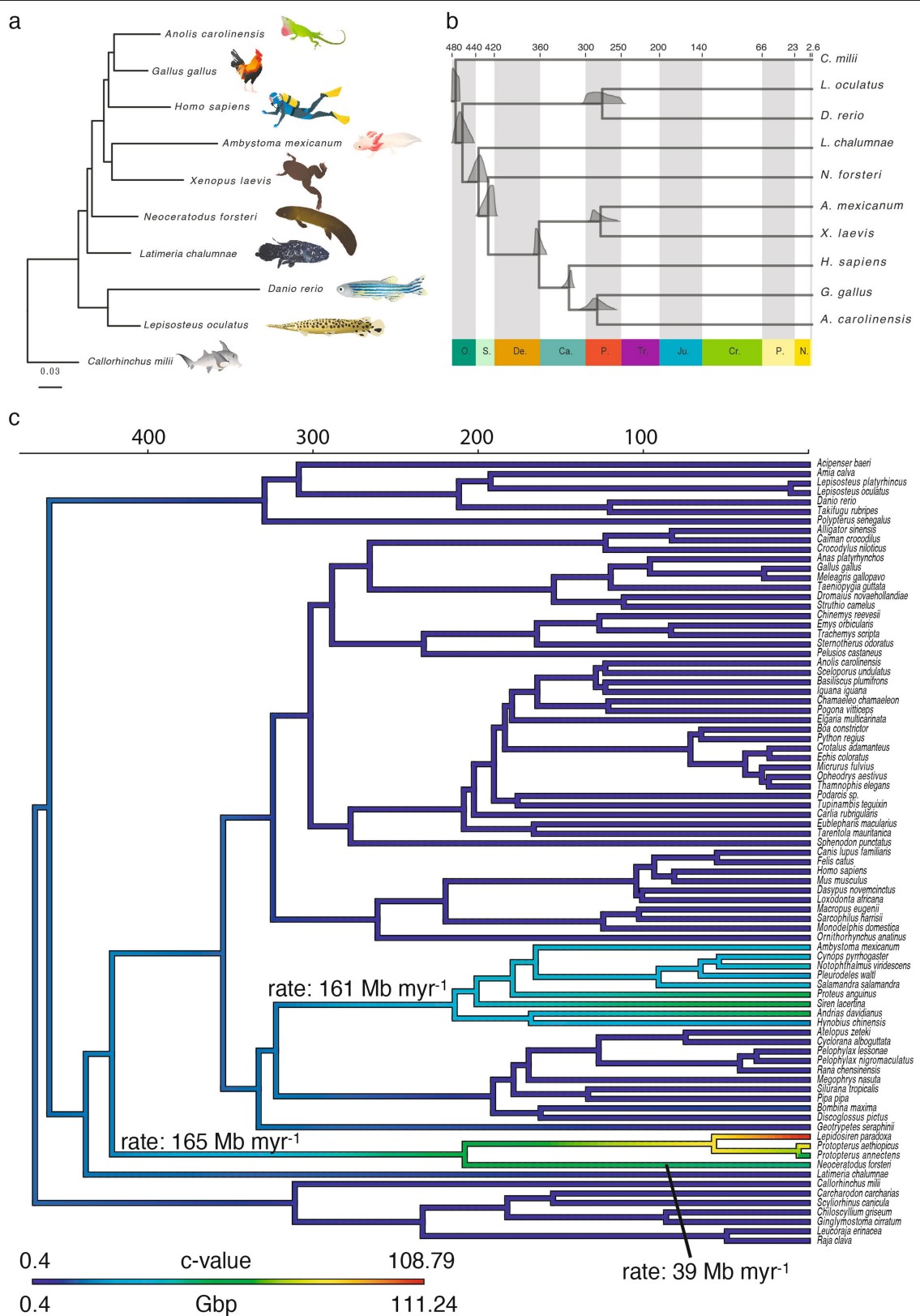

**Extended Data Fig. 3 | CNE-based phylogeny, divergence times and rates of genome evolution. a**, Maximum likelihood phylogeny from noncoding conserved alignment blocks totalling 99,601 informative sites (using RAxML; GTRGAMMA model). All branches were supported by 100% bootstrap value; scale bar is in expected nucleotide replacements per site. Branch lengths of the trees obtained by the CNE method or from the protein sequences show a high correlation ($R^2 = 0.84$, $P < 0.05$). **b**, Relaxed clock time-calibrated phylogeny (MCMCTree). Plots at nodes correspond to full posterior distribution of inferred ages. Scale is in Ma, and main geological periods are highlighted. Plot generated with MCMCTreeR (https://github.com/PuttickMacroevolution/MCMCtreeR). **c**, Evolution of genome size in jawed vertebrates. Maximum likelihood reconstruction of ancestral genome sizes using a time-calibrated phylotranscriptomic tree[8] and genome size values obtained from ref.[80]. Branch lengths are in Ma; colours denote genome size (c-value in pg or Gb). Rates of genome expansion are given for the ancestral branches of lungfishes and salamanders, as well as for the *Neoceratodus* terminal branch.

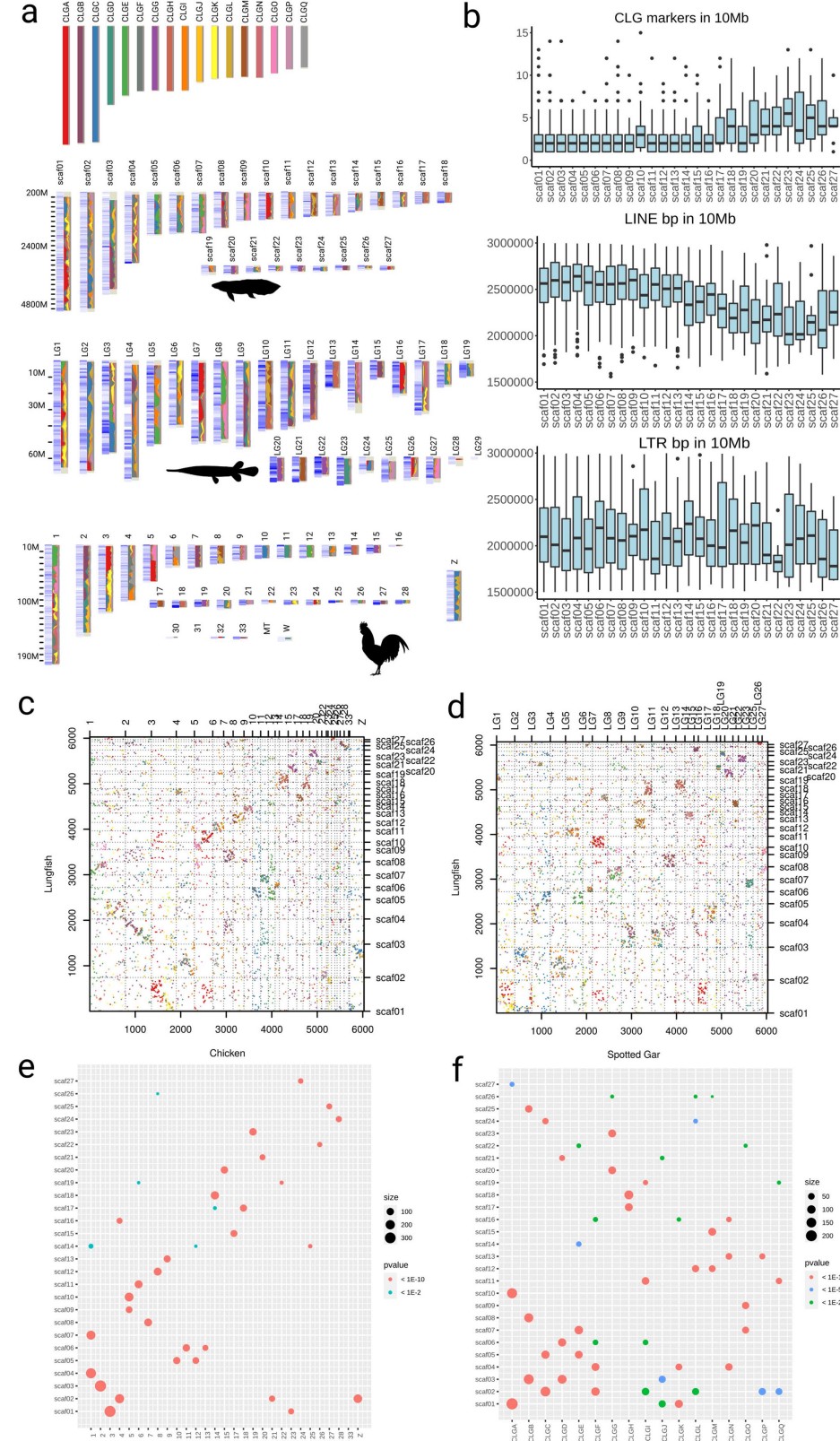

**Extended Data Fig. 4** | See next page for caption.

**Extended Data Fig. 4 | CLG, gene and repeat density along lungfish chromosomes. a**, CLG content profiled within windows of 20 genes with available orthology and CLG identity, and using a 10-gene sliding window. If genes were more than 10 Mb or 100 Mb apart in gar or lungfish, respectively (breaking the 20-gene window), the area is highlighted as grey, indicating areas that lack a sufficient amount of orthologous CLG markers. Blue bar indicates gene density (as measured by the 6,337 marker genes used in the CLG analysis) along 10-Mb windows. White or grey indicates gene desert; blue indicates gene-rich areas. Top row, previously reconstructed CLGs and their colour labels, followed by lungfish, spotted gar and chicken from top to bottom. **b**, Gene and repeat density along 10-Mb windows on lungfish chromosomes. The $y$ axis shows count of CLG genes, LINEs and LTRs per 10-Mb window in top, middle and bottom panels, respectively. Microchromosomes show higher gene density and lower LINE density, and LTR density remains stable. **c**, **d**, Conserved macrosynteny between lungfish and chicken (**c**) and spotted gar (**d**). Chromosomes of chicken (**c**) and gar (**d**) are plotted along with their homologous lungfish chromosomes. The majority of the chromosomes and co-linearity are retained one-to-one. Some recent incorporation of microchromosomes into lungfish macrochromosomes (scaffold 02) has occurred, as evident from sharp syntenic boundaries. **e**, **f**, Significance of the association (homology) between chicken and lungfish chromosomes. Colours correspond to the significance power of the association, or $-\log_{10}$(adjusted Fisher's exact test $P$ values). Fisher's test was run on the number of orthologous gene families shared between any given pair of chromosomes in chicken and lungfish, compared to the overall distribution of orthologous gene families on all other chromosomes. Most chicken microchromosomes (chromosome 6 onwards) are in one-to-one correspondence with lungfish, but some lungfish microchromosomes have recently been incorporated into macrochromosomes. These lungfish macrochromosomes (for example, scaffold 01 or scaffold 02) have significant association with both chicken macro- and microchromosomes. However, these fusions are very recent in lungfish, because the positions of chicken orthologues are restricted to specific areas of the lungfish chromosome (also seen as a clear boundary in Fig. 2c). 'Size' refers to the number of shared orthologous gene families between homologous chromosomes. **f**, Significance of the association (homology) between CLGs and lungfish chromosomes. Fisher's test was run on the number of orthologous gene families shared between any given pair of chromosomes in CLGs and lungfish, compared to the overall distribution of orthologous gene families on all other chromosomes. Silhouette of the lungfish is from ref. [36].

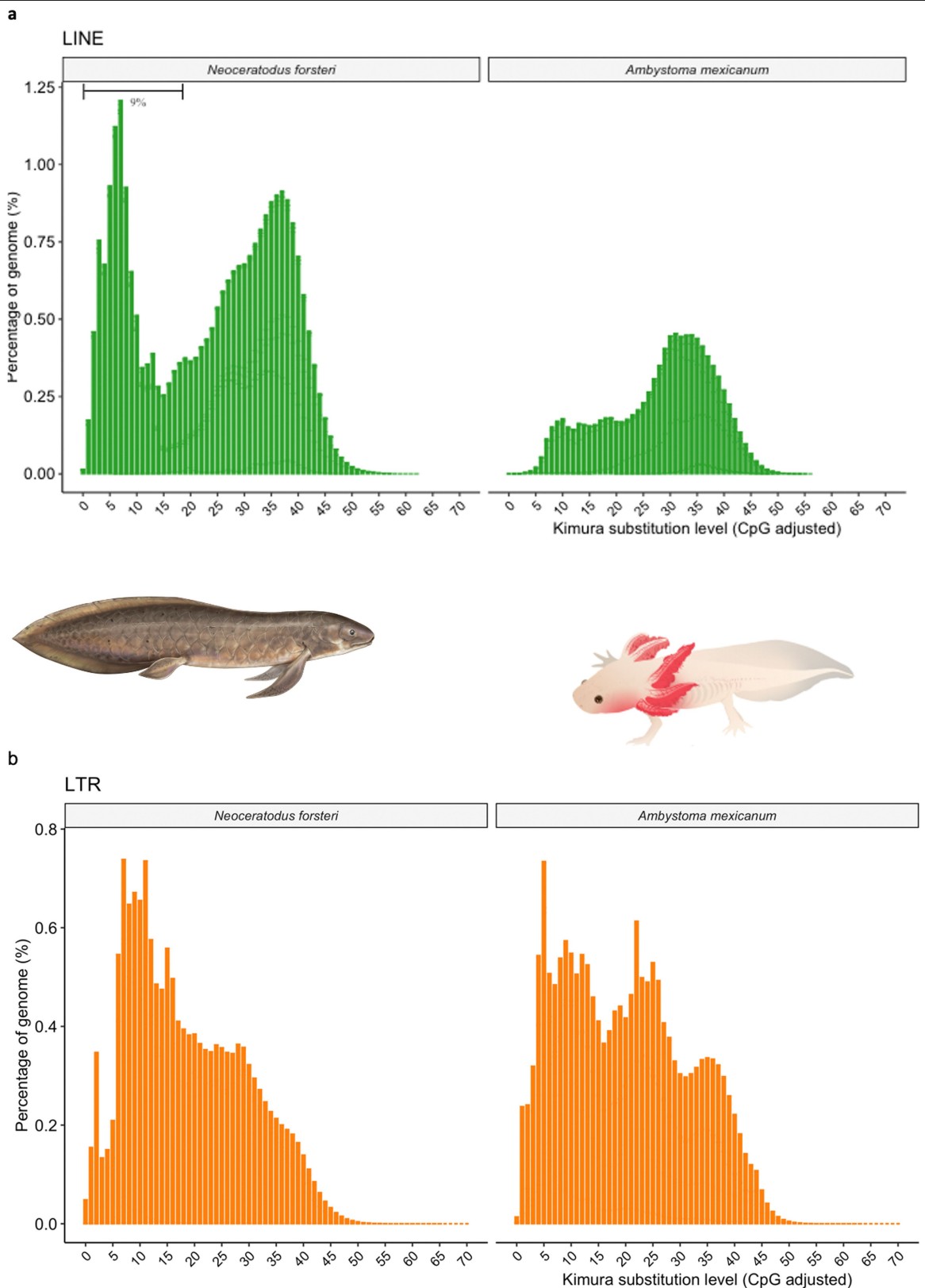

**Extended Data Fig. 5 | Age estimation plots on LINE and LTR classes in Kimura plots. a**, **b**, Repeat landscape of LINE (**a**) and LTR (**b**) of lungfish and axolotl. The two main peaks indicate there were two major LINE expansions in lungfish. The recent expansion (diverging ≤15% from the consensus sequences) contributed to 9% of the lungfish genome. The LTR landscapes are similar in these two species.

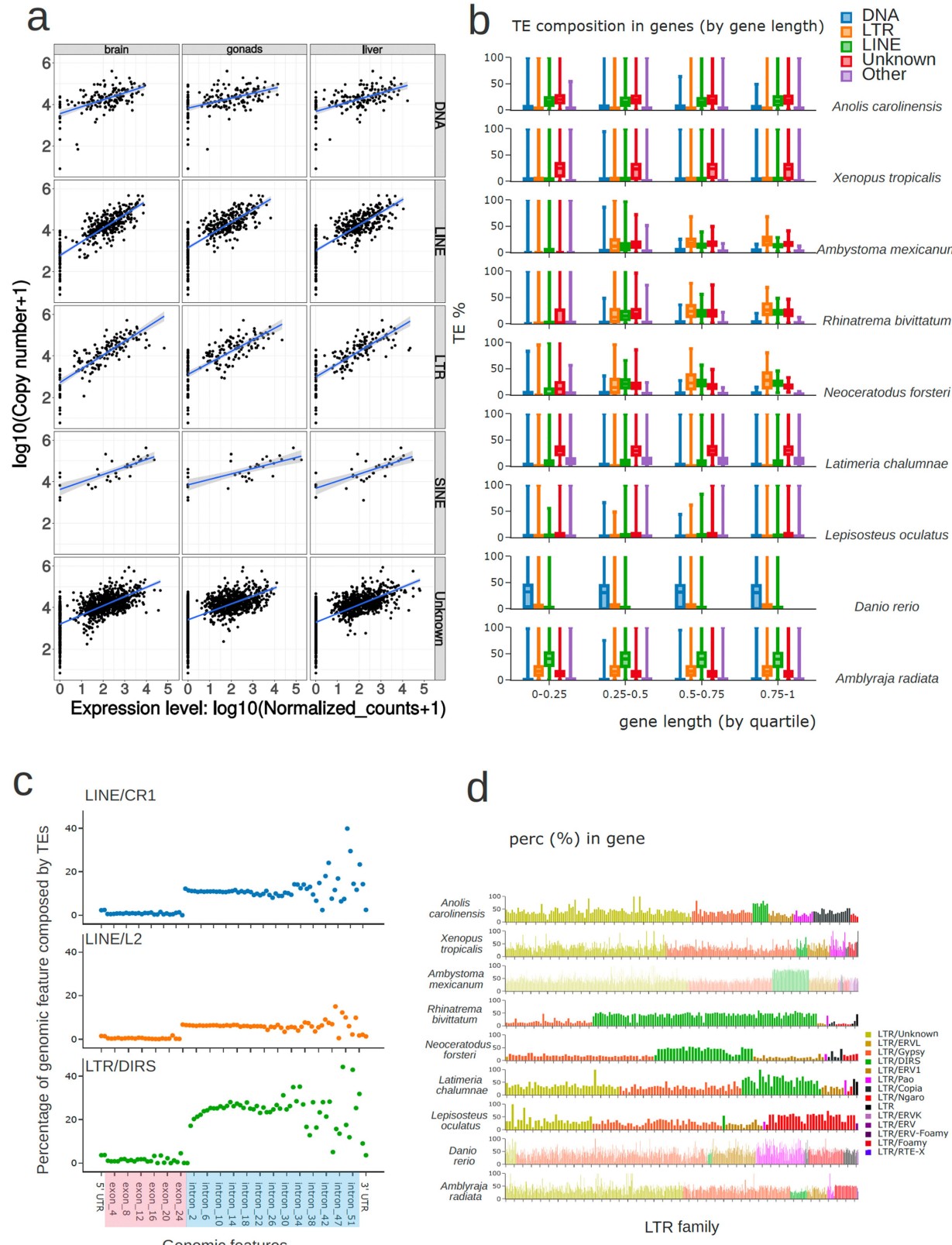

**Extended Data Fig. 6** | See next page for caption.

**Extended Data Fig. 6 | Correlation between expression of transposable element families and copy number in the genome. a**, Expression was estimated for each transposable element family using poly (A)-enriched RNA-seq data from gonad, brain and liver. For all tissues and transposable element classes, a positive correlation is observed between expression level and copy number. When a transposable element family is highly expressed, this family tend to have more copies. However, some families are distant from the correlation line, with a high expression and low copy number or vice versa. The expression levels of transposable element families are globally correlated in the three tissues. **b**, Composition of different classes of repetitive elements in genic regions. Gene and repetitive element annotations were obtained from published reference genomes (see 'Repeats and transposable elements annotation' in Methods). The percentage of different classes of repetitive elements in genic region (including UTRs, exons and introns) were calculated as percentage of the number of bp covered by the repetitive element, normalized by the size of the genes. Genes are grouped by length. As the size of genes varies across species, we grouped them by quartile division per species. The genic LTR percentage (orange) increases in longer genes in lungfish, axolotl and caecilian (vertical lines show the minimum and maximum of the percentage of transposable elements in genes). The box plot shows the median, and the 25% and the 75% quartiles; whiskers show 1.5× the interquartile range. Outliers extend beyond 1.5× interquartile ranges from either hinge. **c**, Percentage of the genic regions that are occupied by different classes of transposable elements. Top and middle, LINE CR1 and LINE L2 (which are classified in the same clade of LINE and are closely related) compose about 5.1% and 2.9% of the lungfish genome, respectively. Bottom, on average, introns (blue) contain a higher number of LTRs and DIRS (about 20 to 30%) than exons (red). **d**, Percentage of LTR families in genic regions (including UTRs). The LTRs and DIRS are enriched in genic regions in lungfish and axolotl.

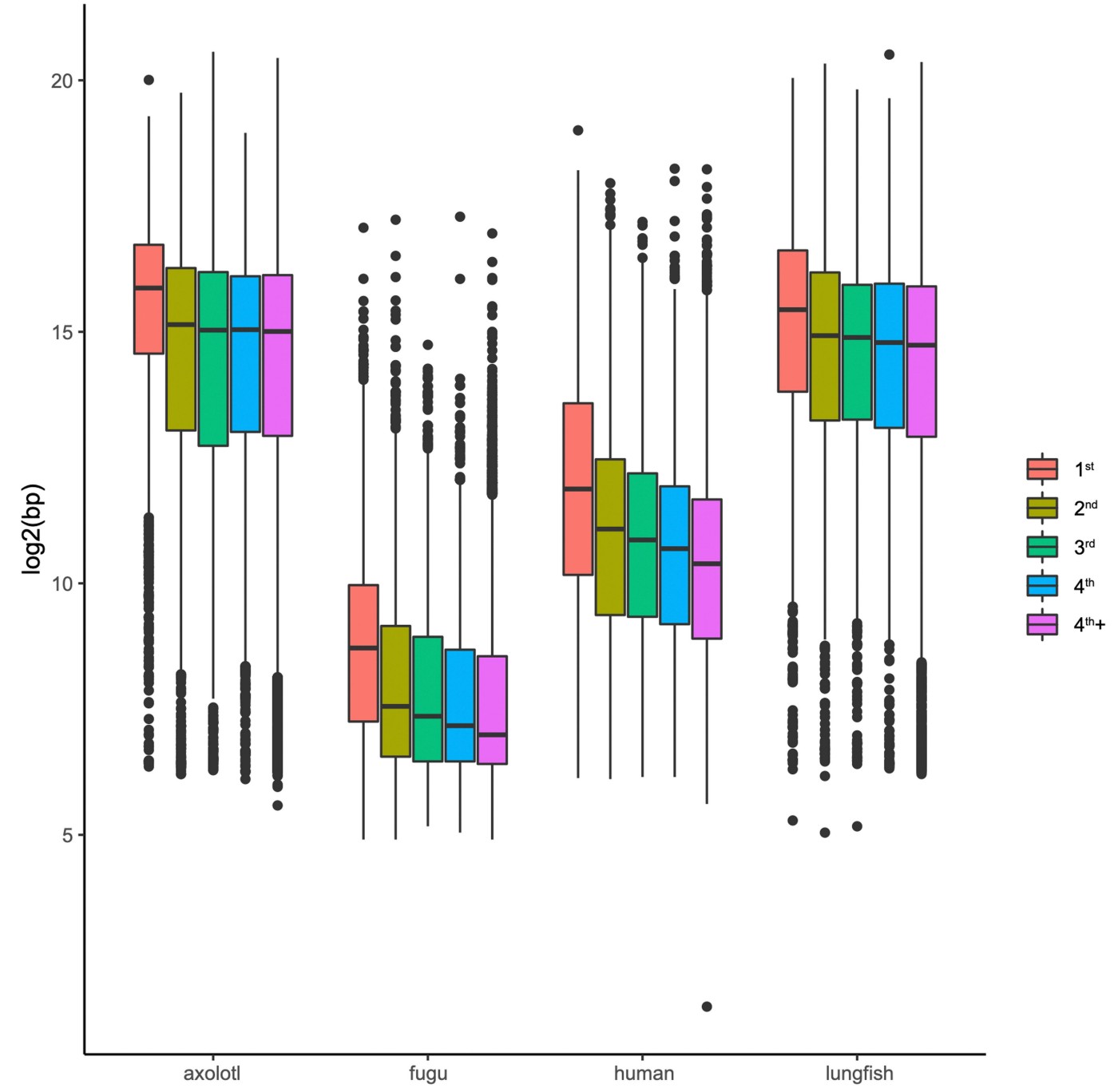

**Extended Data Fig. 7 | Box plot of intron sizes in axolotl, fugu, human and lungfish.** For axolotl, fugu, human and lungfish the lengths ($y$ axis is $\log_2$-transformed scale of base pairs) of the first, second, third, fourth and fifth (and above) introns show a consistent pattern, in which the first intron is always the longest intron—both in the giant lungfish and axolotl genomes as well as in the tiny fugu (400 Mb) genome.

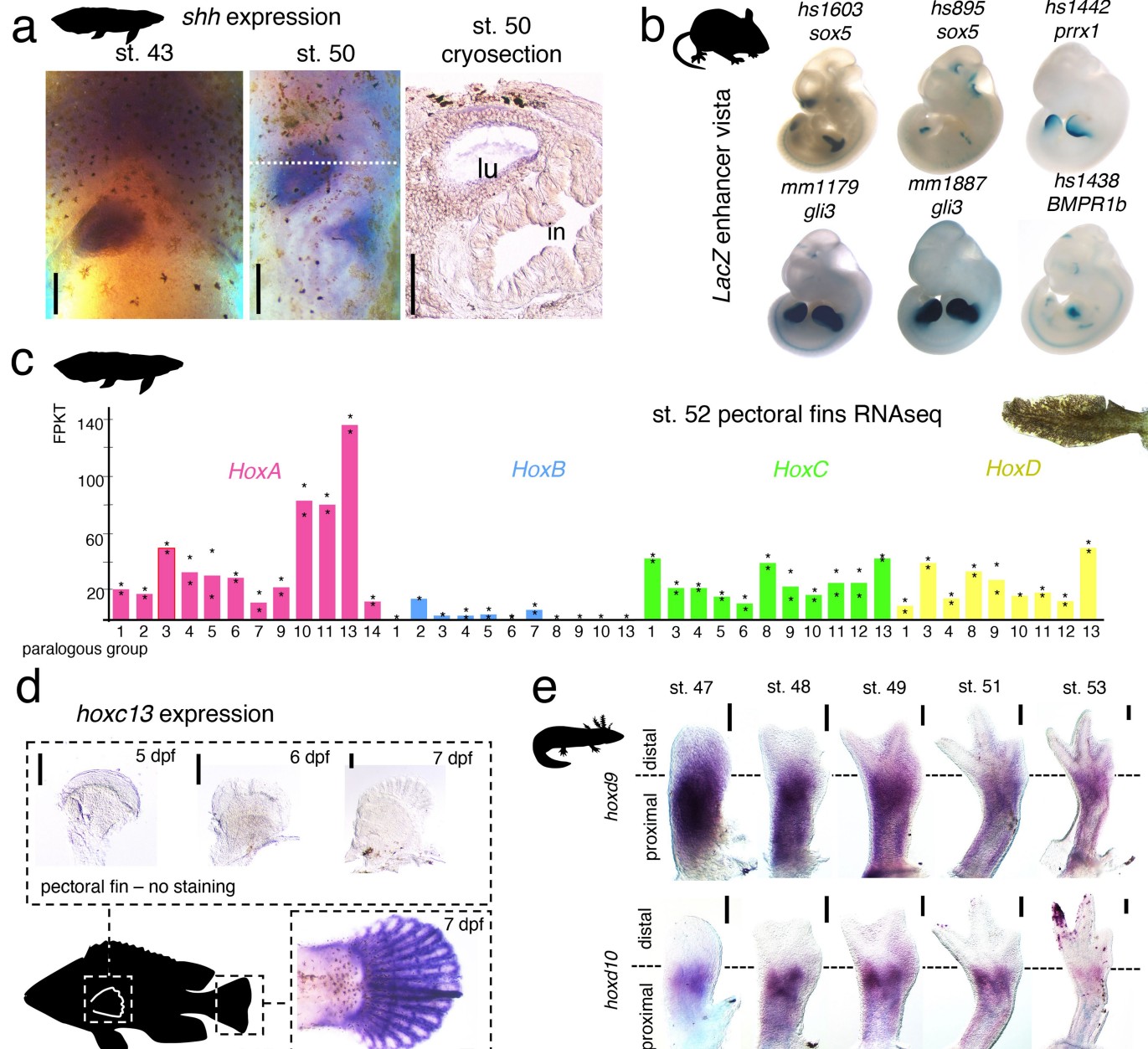

**Extended Data Fig. 8 | Gene expression data in Australian lungfish, ray-finned fish and axolotl salamader. a**, *Neoceratodus* only has a single right lung. *shh* expression in the *Neoceratodus* lung anlage. Stage (st.) 43 ventral view (*n* = 1 out of 1 embryo), anterior up. Stage-48 ventral view, anterior up (*n* = 1 out of 1 embryo). The appearance of the lung anlage and *shh* expression is similar to that in *Xenopus*. Transverse section across dotted line. lu, lung; in, intestine. Scale bars, 0.2 mm. **b**, LacZ enhancer assays in mouse 12-dpf embryos show the regulatory activity of several ultraconserved enhancers that emerged in association with the evolution of the lobed fin. These include elements located near important limb developmental genes that contribute to the sturdy sarcopterygian fin archetype (Supplementary Results). Reported LacZ limb expression: hs1603, *n* = 7 out of 7 embryos; hs895, *n* = 5 out of 8 embryos; hs1442, *n* = 10 out of 11 embryos; mm1179[93,94], *n* = 7 out of 7 embryos; mm1887, *n* = 6 out of 6 embryos; hs1438, *n* = 5 out of 11 embryos . **c**, hox gene expression from RNA-seq analysis of stage-52 pectoral fins (*n* = 2). Individual data points shown with asterisks; the height of the bar indicates average expression.

Overlapping data points indicated with a single asterisk. High expression of posterior hoxa and hoxd genes (except for *hoxa14*), low expression of hoxb genes and unexpectedly high expression of hoxc genes. **d**, Absence of *hoxc13* expression from pectoral, but not caudal, fins in the ray-finned cichlid *Astatotilapia burtoni*. A staging series of cichlid pectoral fins (5–7 dpf) does not show expression of *hoxc13*, whereas this gene stains strongly in the caudal fin (*n* = 4/4 embryos per stage). This result is consistent with a sarcopterygian origin of *hoxc13* expression in the distal paired fins and limbs. Scale bars, 0.1 mm. **e**, Non canonical patterns of *hoxd9* and *hoxd10* expression in axolotl limbs (*n* = 2/2 limbs per stage). Expression of *hoxd9* and *hoxd10* during axolotl limb development shows strong expression in a proximal limb domain but absence or low expression in the distal limb or digit domain. This noncanonical expression is similar to that previously reported for *hoxd11*[37,95], and suggests a loss of contact with the distal limb enhancers located 5′ of the hoxd cluster, caused by the expansion of the posterior hoxd cluster. Scale bars, 0.2 mm. Silhouettes are from ref. [36].

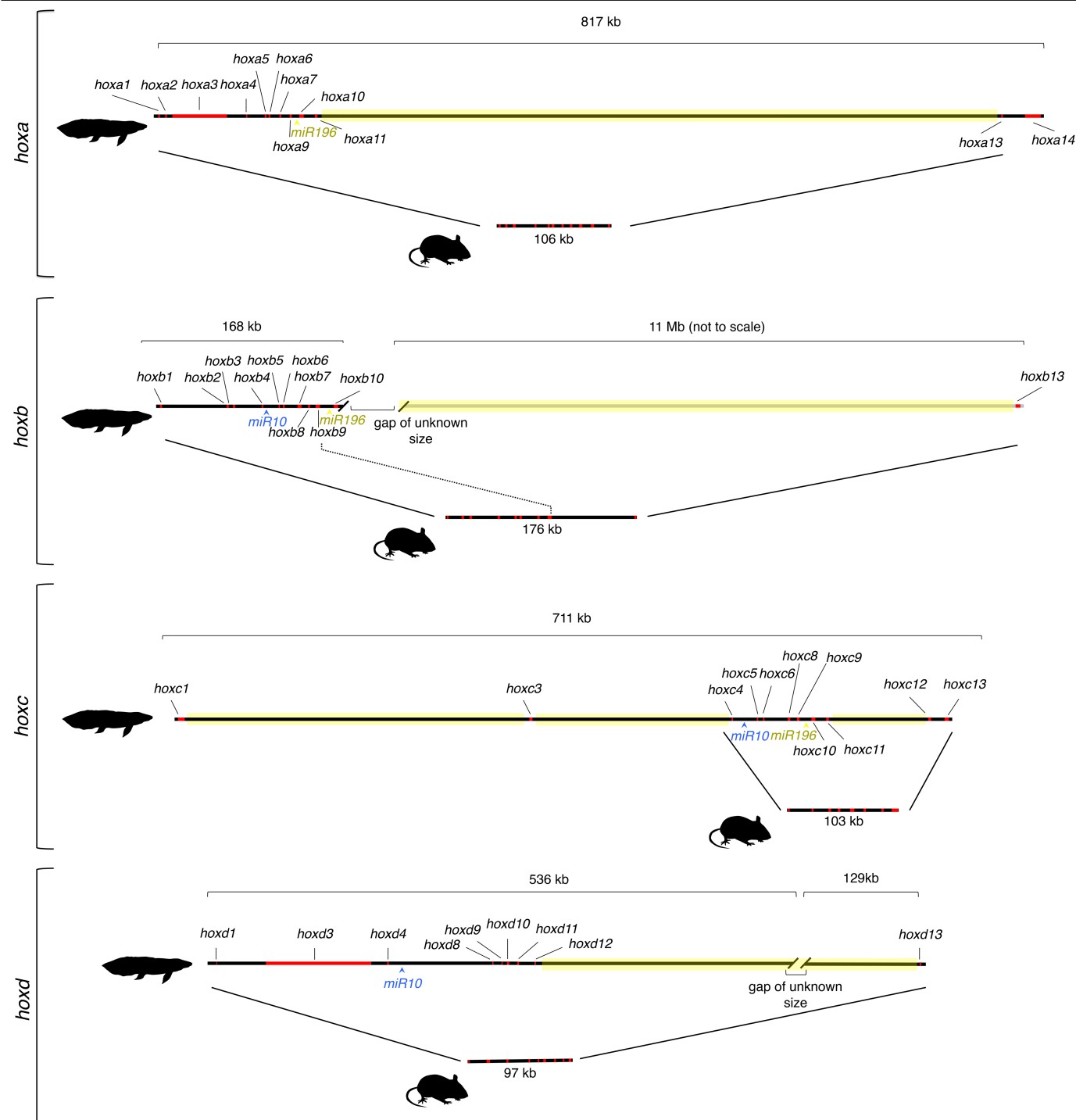

**Extended Data Fig. 9 | Comparison of *Neoceratodus* and mouse hox clusters.** Four hox clusters are present in the *Neoceratodus* genome (hoxa, hoxb, hoxc and hoxd), comprising 43 genes and 6 conserved miRNA genes (*miR10* and *miR196*). *Neoceratodus* preserves a copy of *hoxb10* and *hoxa14*, which are lost in tetrapods. The 3′ hoxc cluster contains the *hoxc1* and *hoxc3* genes, which are lost in several tetrapod lineages but have been shown to be part of the original tetrapod hox complement[96]. Consistent with the overall expansion of the *Neoceratodus* genome, its hox clusters are larger than their mouse counterparts. Expansion has occurred unevenly across the clusters and intergenic regions of highest expansion are indicated with yellow mark up (*hoxa11* to *hoxa13*; *hoxb10* to *hoxb13*; *hoxc1* to *hoxc3*; *hoxc3* to *hoxc4*; *hoxc11* to *hoxc12*; and *hoxd12* to *hoxd13*). Furthermore, the introns of *hoxa3* and *hoxd3*

are enlarged. All clusters shown (both mouse and *Neoceratodus*) are drawn to scale with the respective sizes indicated, except for the 11 Mb between *hoxb10* and *hoxb13*, which is drawn about 20-fold reduced. The *Neoceratodus hoxb13* and *hoxd13* are present on separate contigs and the exact genomic distance to their nearest neighbouring hox gene has not been determined. The sizes for the hoxb and hoxd clusters therefore represent a lower limit. The mouse has lost *hoxa14* and the indicated synteny for hoxa runs from *hoxa1* through *hoxa13*. Similarly, the mouse hoxc cluster lacks *hoxc1* and *hoxc3* and the comparative hoxc synteny runs from *hoxc4* through *hoxc13*. Gene labels are included for the *Neoceratodus* cluster, whereas in the mouse clusters genes are indicated only using red boxes. miRNAs are indicated only for the *Neoceratodus* clusters. Silhouettes are from ref. [36].

a

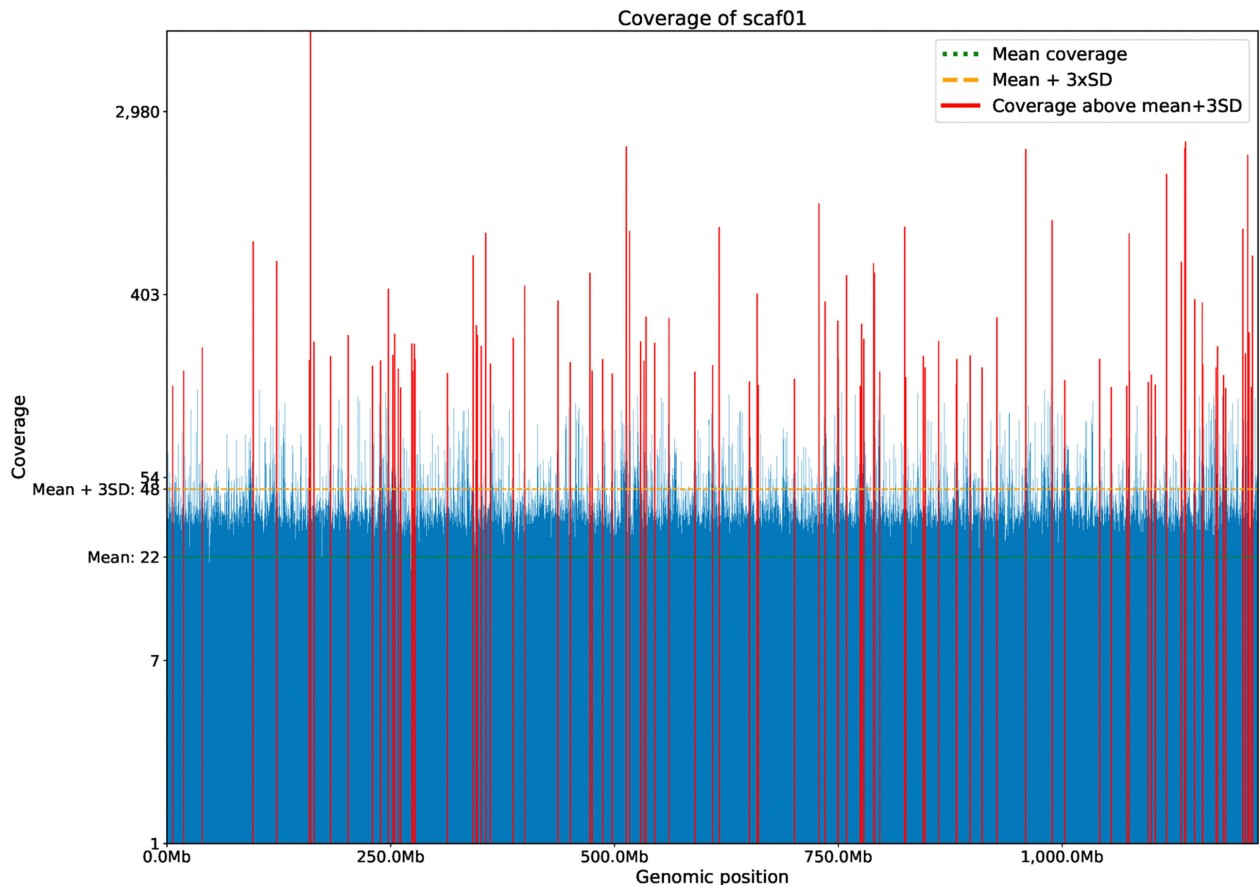

b

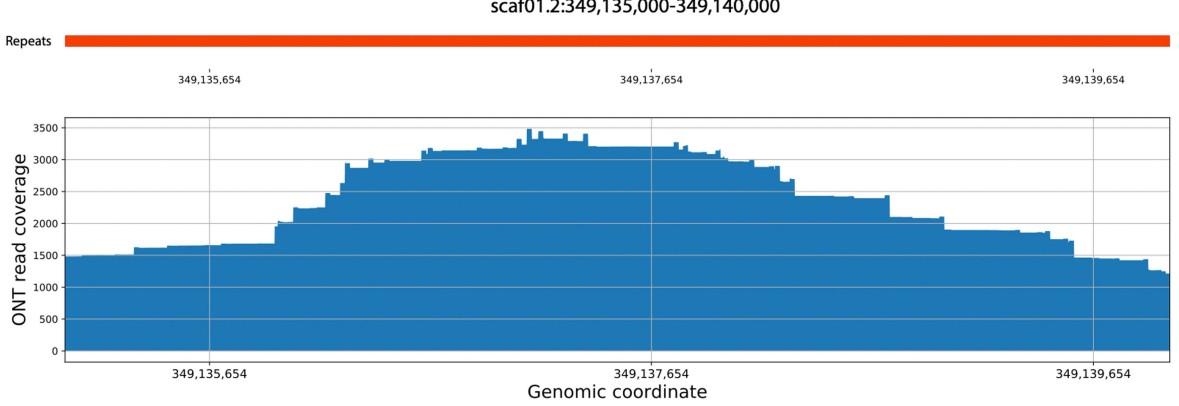

**Extended Data Fig. 10 | Validation of the assembly of the *Neoceratodus* genome. a**, Read coverage along the assembly showing a portion of scaffold 01. Red lines mark regions exhibiting a coverage >3 s.d. from the mean. Overall, these regions represent 0.09% of the genome. **b**, Representative region showing read pile-up with coverage in excess of 3 s.d. from the mean. The entire region is contained within a region annotated as repetitive by RepeatMasker (red interval).

# Reporting Summary

Nature Research wishes to improve the reproducibility of the work that we publish. This form provides structure for consistency and transparency in reporting. For further information on Nature Research policies, see our Editorial Policies and the Editorial Policy Checklist.

## Statistics

For all statistical analyses, confirm that the following items are present in the figure legend, table legend, main text, or Methods section.

| n/a | Confirmed | |
|---|---|---|
| ☒ | ☐ | The exact sample size (*n*) for each experimental group/condition, given as a discrete number and unit of measurement |
| ☒ | ☐ | A statement on whether measurements were taken from distinct samples or whether the same sample was measured repeatedly |
| ☒ | ☐ | The statistical test(s) used AND whether they are one- or two-sided<br>*Only common tests should be described solely by name; describe more complex techniques in the Methods section.* |
| ☒ | ☐ | A description of all covariates tested |
| ☒ | ☐ | A description of any assumptions or corrections, such as tests of normality and adjustment for multiple comparisons |
| ☒ | ☐ | A full description of the statistical parameters including central tendency (e.g. means) or other basic estimates (e.g. regression coefficient) AND variation (e.g. standard deviation) or associated estimates of uncertainty (e.g. confidence intervals) |
| ☒ | ☐ | For null hypothesis testing, the test statistic (e.g. *F*, *t*, *r*) with confidence intervals, effect sizes, degrees of freedom and *P* value noted<br>*Give P values as exact values whenever suitable.* |
| ☒ | ☐ | For Bayesian analysis, information on the choice of priors and Markov chain Monte Carlo settings |
| ☒ | ☐ | For hierarchical and complex designs, identification of the appropriate level for tests and full reporting of outcomes |
| ☒ | ☐ | Estimates of effect sizes (e.g. Cohen's *d*, Pearson's *r*), indicating how they were calculated |

*Our web collection on statistics for biologists contains articles on many of the points above.*

## Software and code

Policy information about availability of computer code

| Data collection | No software was used to collect the data. I don't know if we should list NanoPore software here. Probably not, as it was done by a company. For RNAseq data, I also don't think we need to list the sequencer software, it doesn't make sense. Therefore, I think the statement in the first sentence should be enough |
|---|---|
| Data analysis | MARVEL https://github.com/schloi/MARVEL |

For manuscripts utilizing custom algorithms or software that are central to the research but not yet described in published literature, software must be made available to editors and reviewers. We strongly encourage code deposition in a community repository (e.g. GitHub). See the Nature Research guidelines for submitting code & software for further information.

## Data

Policy information about availability of data

All manuscripts must include a data availability statement. This statement should provide the following information, where applicable:
- Accession codes, unique identifiers, or web links for publicly available datasets
- A list of figures that have associated raw data
- A description of any restrictions on data availability

PRJNA645042

# Field-specific reporting

Please select the one below that is the best fit for your research. If you are not sure, read the appropriate sections before making your selection.

☒ Life sciences ☐ Behavioural & social sciences ☐ Ecological, evolutionary & environmental sciences

For a reference copy of the document with all sections, see nature.com/documents/nr-reporting-summary-flat.pdf

# Life sciences study design

All studies must disclose on these points even when the disclosure is negative.

| | |
|---|---|
| Sample size | n/a |
| Data exclusions | No data were excluded |
| Replication | n/a |
| Randomization | The data were not randomized |
| Blinding | No blinding was necessary as we used all data for the genome assembly and analyses thereof |

# Reporting for specific materials, systems and methods

We require information from authors about some types of materials, experimental systems and methods used in many studies. Here, indicate whether each material, system or method listed is relevant to your study. If you are not sure if a list item applies to your research, read the appropriate section before selecting a response.

## Materials & experimental systems

| n/a | Involved in the study |
|---|---|
| ☒ | Antibodies |
| ☒ | Eukaryotic cell lines |
| ☒ | Palaeontology and archaeology |
| ☐ | ☒ Animals and other organisms |
| ☒ | Human research participants |
| ☒ | Clinical data |
| ☒ | Dual use research of concern |

## Methods

| n/a | Involved in the study |
|---|---|
| ☒ | ChIP-seq |
| ☒ | Flow cytometry |
| ☒ | MRI-based neuroimaging |

## Animals and other organisms

Policy information about studies involving animals; ARRIVE guidelines recommended for reporting animal research

| | |
|---|---|
| Laboratory animals | *For laboratory animals, report species, strain, sex and age OR state that the study did not involve laboratory animals.* |
| Wild animals | *Provide details on animals observed in or captured in the field; report species, sex and age where possible. Describe how animals were caught and transported and what happened to captive animals after the study (if killed, explain why and describe method; if released, say where and when) OR state that the study did not involve wild animals.* |
| Field-collected samples | *For laboratory work with field-collected samples, describe all relevant parameters such as housing, maintenance, temperature, photoperiod and end-of-experiment protocol OR state that the study did not involve samples collected from the field.* |
| Ethics oversight | *Identify the organization(s) that approved or provided guidance on the study protocol, OR state that no ethical approval or guidance was required and explain why not.* |

Note that full information on the approval of the study protocol must also be provided in the manuscript.

