## [Peer Review File · Nature]

Manuscript Title: Giant Lungfish genome elucidates the conquest of land by vertebrates

Editorial Notes:

Reviewer Comments & Author Rebuttals**Reviewer Reports on the Initial Version:**Referee #1 (Remarks to the Author):

The manuscript describes the new genome assembly for the Australian lungfish. The genome is compared to several other species, especially the large genome of salamanders. The findings include describing the Hox gene cluster, repetitive element expansion, and phylogenomic placement of the lungfish at the base of the tetrapods. The surfactant encoding gene expansion is quite interesting, considering the biology of the lungfish. The genome is an exciting addition to understanding vertebrate and tetrapod evolution.

There is a large mismatch between the flow cytometry and Feulgen photometry estimates of genome size and the k-mer estimate from the reads. Is the assembled genome missing an additional 15+ Gb of repetitive DNA? Certainly, the genome is missing at least 5 Gb of information given the mismatch between the k-mer estimate and the assembled size.

The novelty of the phylogenomic method was unclear. The implemented method did not appear to be developed and presented as a part of this manuscript.

Some outstanding questions that came to mind while reading the manuscript:

How was RNA used for gaps, indels and misalignments?

What is the BUSCO score on the gene models?

The rationale and need for double masking for repetitive elements was unclear. The inference of two waves of repeat element expansion seems to be based somewhat on the two rounds of repeat masking, which is insufficient evidence. However, the accumulation of substitutions shown in Figure 3b suggests two rounds.

Generally, Figure 2 is unclear. Scaf01 is shown in (a) and (c), is it a macro or microchromosome? The mentioned sharp syntenic boundaries are not clear. The sorting of the lungfish scaffolds in panel (a) is unclear.

Is there any reason to think the axolotl and lungfish genome would have similar expansions? There is a large emphasis on the comparison between the two species because of the large genomes but given their phylogenetic placement, in my mind, there is no reason to presume that the repeat expansion would have been shared.

There are many writing issues which were distracting, including misspelling of evolutionary and phrases such as "hugely larger", "including besides", etc. The first time CNE is defined the word non-coding is excluded. The phrase "indicating the existence of island of genome gigantism" is unclear.

Referee #2 (Remarks to the Author):

In this manuscript, Meyer et al describe their findings resulting from sequencing the genome of the Australian lungfish, making it the largest vertebrate genome sequenced to date. They confirm the lungfish status as sister group of tetrapods and show that their chromosomal make up contains the ancient chordate linkage groups and can be explained by the 2 rounds of genome duplication experienced by jawed vertebrate lineage. Lungfish genome is diploid and contains microchromosomes highly syntenic with those in chicken and Gar. The lungfish genome expanded via accumulation of transposable elements in intronic and intergenic regions; the principal category being LINES in two waves that happened independently from that observed in salamander genomes. The general composition of TE is most similar

to that seen in tetrapods. Surfactant and Vomeronasal Receptors experienced expansion in the number of protein coding genes. Finally, the authors examine Hox gene cluster composition and expression and propose that Hoxc13 expression domain in the developing lungfish fin is homologous to interdigital expression seen in the mouse. The authors also show that posterior Hox genes such as Hoxd13 have been “pushed” away from other genes in the cluster and propose that, as a result, distancing may have caused loss of influence of late-phase regulatory elements on Hoxd11 gene expression, explaining lack of digital expression of this gene in lungfish and axolotls.

In all, the manuscript is an important contribution to our understanding of the evolution of vertebrate genomes and potential preadaptations to the vertebrate water-to-land transition. Whereas the findings are of broad interest and the experiments and analyses are largely sound, I outline below issues that require attention.

1. In the Summary:

- First line is confusing. "Sarcopterygii" no more left the water than Osteichthyans did. One lineage within sarcopterygians radiation did. The way it's written implies lungfishes left the water.
- Line 55, "...mechanisms than salamanders, the other vertebrate lineage with enormous genomes". I would recommend saying "another vertebrate lineage with enormous genome", as the "enormous genome" category is not well defined.
- Line 60: word "evolutionary" is missing the "y".

2. Introduction:

- Line 70: This sentence is incorrect as written. The clade is not "freshwater fish." Much of the lineage's history includes occupancy of marine environments. Living representatives, however, are freshwater.
- Line 73: "...initially thought not to be fish but amphibians". Please provide citation.
- Line 79: "and is one of the oldest genera". Please provide citation.
- Line 83: "...filaments that do not aid in locomotion^{1,4,5}". This statement is incorrect. Please see and cite King et al. (PNAS 2011), and Aiello et al. (J Exp Biol. 2014).
- Line 88: Marjorie Courtenay-Latimer's name is misspelled in the text.
- Line 92: "...it is now possible to revisit and resolve this long-standing crucial evolutionary question using whole-genome-derived datasets". Paleontological and previous molecular work up to the coelacanth genome (Amemiya 2013 Nature) have reached this conclusion before. I am not convinced this is a "long-standing" unresolved matter. Consider toning down the statement.

3. Results and discussion

- Line 108: "But lungfish genomes are even bigger". Please provide citation.
- Line 146: "...that has been shown to overcome phylogenetic artefacts". Please specify which artifacts.
- Fig 1: Dedicating this figure to Jenny Clack and Robert Carroll is quite nice. Nevertheless, I feel I should point out that this may be a sensitive issue. The authors should very carefully consider the optics of only including white European scientists as representatives of the human species.
- Line 166: "The lungfish chromosomal scaffolds show a remarkable conservation of the ancestral chordate karyotype". What is remarkable about the amount of conservation? Is it greater than expected? Please clarify or reword.
- Line 180: "This suggests, by comparisons to gar, anolis, and chicken, that a typical set of microchromosomes existed in the ancestral vertebrate". This conclusion has been reached before, see the Gar genome paper. Please rephrase, for instance, indicating that the results "confirms previous reports"... And cite Braasch 2016 Nature Genetics.
- Line 181: Anolis, as a genus name, should be capitalized, italicized.
- Line 228: "We find that the global repeat landscape (overall amount of the major TE classes)..." the amount as a proportion of the genome or by bp counts? Please clarify.
- Line 231: "Interestingly, the two largest animal genomes, lungfish and axolotl...". Given that extant lungfish species vary greatly in their genome size, I recommend specifying Australian lungfish as opposed to using "lungfish" to refer to all species.
- Line 256: delete the word "besides". And consider revising the entire sentence. As is, it is very confusing.
- Fig 3a: Some categories on the pie charts are described in minuscule font unreadable if printed.
- Line 279: "The conserved element (CNE) dataset...". Missing the word non-coding.
- Line 272: "random process of repeat, transposon and junk DNA expansion". I suggest avoiding using the term "Junk DNA", as it presumes, without evidence, absence of function.
- Line 301: "This approach allowed us to identify 308 genomic regions with signatures of accelerated evolution in lungfish". This is interesting. Were some examined more closely? Any with potential relevance for lungfish physiology/evolution?
- Line 345/346: "...high expression of posterior hoxa and hoxd genes (except for hoxa14 as previously reported 42,43)". Reference 42 is a manuscript in press and therefore yet unavailable. Without access to the data I cannot evaluate the statement regarding hoxa14.
- Line 354: This large section is a bit difficult to follow, but if I understood correctly, the hypothesis is that genome expansion promoted a distancing of Hoxd13 from the minicluster of genes spanning Hoxd8-d12.

This in turn, resulted in maintenance of regulatory influence of late-phase hox enhancers only over *hoxd13*. This would explain why *Hoxd11* in *axolotl* and lungfish are not expressed in the same domain as *d13*. Regarding this:

o Reference 42 (paper in press) is cited for *Hoxd11* expression in lungfish fins. As I cannot access the primary data (only the diagram from fig 4), I cannot critically evaluate the claim. Can the authors share an image of *Hoxd11* in situ in lungfish?

o If this loss of regulatory influence is correct for both lungfish and *axolotl*, it would presumably impact other *Hoxd* genes such as *d12*, *d10*, *d9*. In reference 49, only *Axolotl d11* and *d13* were examined. If the authors have in situ data for other *Hoxd* genes showing similar exclusion of digit (*axolotl*) or distal fin (lungfish) expression, it would strengthen the claim.

4. Conclusion:

- Line 421: sentence needs revising.

5. Materials and Methods

- In Biological Materials section, specify where did the embryos come from and, if pertinent, permits for breeding. Also, specify if the animal used for genome sequencing was male or female. Is there sexual dimorphism in genome size -- sex chromosomes? Does that impact the results?

Supplementary Figure 2: I suggest also adding Jenny Clack to this phylogeny. Supplementary Figure 11: The word intron in the figure legend is misspelled.

Supplementary Figure 11: "(Y-axis is log scale of bp s.". Close the parenthesis and remove "s").

Referee #3 (Remarks to the Author):

The authors describe the sequencing, assembly and analysis of a lungfish genome. The work is technically impressive and full of interesting analyses that shed light on the evolution both of large genomes and also early vertebrate evolution, both phylogenetically and from a genome architecture perspective. It is also nice to see it confirm some of the observations made in the analysis of the *axolotl* genome about the evolution of large vertebrate genomes. The scope and scale of this work make it appropriate for a high profile publication. Practically, it is worth mentioning that the paper is very large and it was challenging to review comprehensively. What follows is our best attempt!

Comments on the assembly:

The assembly created is impressive! A 2.2GB scaffold N50 is amazing. However, there is little evaluation of the resulting assembly quality. The paper says "88.2% of the DNA reads and 84% of the RNAseq reads aligned to the genome.", but this analysis is not (as far as I could find - I may have missed it) described in the methods, so it is unclear how the reads were aligned, if this was all the long-reads and short-reads, and if the reads were aligned with high-mapping quality. Furthermore, it is surprising given the repetitiveness of the genome, that there is no evaluation of misassemblies or assembly collapses. I would very much like to know how many of the genes assembled were subject to assembly collapse. Given that the authors strived for a high-quality analysis, it would be good to see some more in-depth, honest analysis of the assembly and its limitations.

How many chromosomes were there predicted to be via karyotype and other prior work? How many macrochromosomes vs microchromosomes, and how big are each? (Is the huge final scaffold N50 of 2.2 Gb reasonable / expected?) What are the estimated sizes of the true chromosomes in lungfish, and for each chromosome what is the distribution and ordering of assembly contig sizes? Also, your assembly N50 may be accurate because of the size of the genome, but it would be good to know if your scaffolds agree with previous estimates of chromosome size by Rock et al 1996 or others (if the individual chromosome masses are available)

Perhaps include an N(x) plot of the assembly (pre/post scaffolding) to get a sense of the distribution of contig and scaffold sizes.

What is the heterozygosity of the genome? There is no heterozygous peak on the kmer count figure (Supplementary Figure 1)

The scaffolding process in particular seems to take into account the difficulties of scaffolding such a highly repetitive genome, but the final scaffold N50 of 2.2 Gb seems potentially too good to be true, particularly given this was done with only 7.55x coverage of HiC data. Are all of the joins correct? Validation via an additional sequencing type (such as BioNano) would be good. How many gaps are there in the final assembly? Could a final round of gap closing be done, if it was not already?

What is an estimate of the final Q value of the assembly? (Could estimate via kmers with a tool like Merqury)

More detail is needed in the description of the methods for the genome assembly, which is currently very brief. How was the 1%-against-all alignment of reads used to find repetition? What are key features of the MARVEL assembler? (we ended up finding a description in the Supplementary Information section of the axolotl paper, Nowoshilow 2018). How is repeat masking used during the read correction, read overlap, and graph layout stages of assembly? Why was the MARVEL assembler used as opposed to any of the other long-read assemblers out there (such as Flye which is generally well-suited for handling repeats)? This is easily justified, but worth commenting on.

Perhaps include a figure/flowchart of some sort to help with the explanation of the HiC scaffolding method? It is a bit hard to follow in the text.

What were the runtimes of the assembly, scaffolding, and polishing stages? How much disk space/memory is required?

Comments on the alignment and conservation analysis:

The generated multiple sequence alignment relies upon reference guided alignment to the evolutionarily extremely distant human reference genome, which is separated from the lungfish by well over one neutral substitution per site! This must inevitably mean that only very highly conserved elements, such as core genes and conserved non-coding elements, will be present in the alignment. Indeed, despite the genome of the lungfish being more than 30 billion bases long just less than 100 thousand bases (1/300,000 bases of the lungfish!) were used for the phylogenetic tree building from the WGA. While this is probably sufficient for the purposes of establishing the ancestry relationships, it is worth mentioning that these highly, highly filtered elements are not representative of a neutral evolutionary process, and so it is possible that they could create misleading phylogenetic reconstruction, certainly, the inferred branch lengths must be very questionable.

Similarly, I am suspicious that the use of 4d sites to estimate neutral rates for phastcons for this phylogeny will be heavily biased by the lack of truly neutral evolution in the elements (even degenerate codon positions) conserved in this alignment - how many sites were these estimates made from? Were there enough sites to estimate a model for? I was surprised to see that you ran a different model for each human chromosome, which only vaguely resembles the ancestral synteny structure of the MRCA of these species. How many sites did you use, for example, to estimate the rate for human chromosome 22!? I'm guessing the interchromosomal variation will be a good reflection of the variance in the estimate of these rates.

Why did you use the space between CNEs to estimate intergenic distances? It seems like you could have had a higher density set of markers (you state 229 pairs) simply by looking at the distance between protein-coding genes?

The analysis of potential positive selection within CNEs and protein-coding genes is potentially confused by misassignment of orthology. This could happen because of collapses within the assembly itself, or because of misalignment. This is known to plague MultiZ alignments, which frequently pick the wrong sequence to align to the reference genome. What tests were performed to check that misalignments and misassemblies did not affect these results? I could not find any in the methods, but may have missed them. Without some further scrutinization of these alignments I am skeptical of all the positive selection results reported.

Comments on the repeat / functional analysis:

Is there any evidence of the speculation that "such bursts of TE activity might also have been involved in the origin of novel gene functions" (line 225)?

The standard repeat masking procedure of the 37 Gb genome assembly identified 57.3% of the genome as being repetitive (Fig.3a). This corresponds to 24.65 Gb of repetitive DNA, that together with the axolotl, is the highest repeat content in the animal kingdom. After masking the repeats using standard procedure (Supplementary Table 2), we tested whether the remaining unmasked DNA contained more repetitive sequences. This second round of repeat annotation on the hard-masked genome revealed an additional 23.92% of repetitive DNA (Fig.3a). Most of this additional repetitive DNA was classified as "Unknown" (and added 11% to the total unknown portion of repetitive DNA) as well as "LINE" (8.5% added, respectively) (Supplementary Table 3). This means that at least 2/3 of the lungfish genome is repetitive and expanded in two waves (Fig.3b).

This would suggest that 1/3 is non repetitive, or about 13Gb of sequence. If 13gb is truly non-repetitive, what is its function (other than the 31k genes, and 17k ncRNA). Later in Fig3A, the plot labelled "Double

TE Masking" claims there is 6.23% "DNA" or about 2.4Gb. It's not totally clear how this is calculated and what is the meaning of "DNA", please clarify. We believe it refers to MER1, MER2 and mariners?

It is also worth explaining why 2 iterations of RepeatModeler + RepeatMasker were needed, or perhaps using a different method for identifying repeats if RepeatModeler does not perform well in this case. Is it not possible to change the parameters of the modeler so that 1 iteration is enough?

We would argue that the expansion of gene families relative to fish is one of the most interesting results in the paper, and should be worth its own figure (even if it is supplementary). These genes could also perhaps be compared in terms of Kimura substitution levels.

Is there evidence of transcriptional repression of transposable elements? Is there a visible history of evolution of TEs in response to their repressors?

Are there any mutations or gene expansions that suggest that the lungfish has adapted to having a physically larger genome? A 43Gb genome must be difficult to replicate during S phase and align during metaphase.

Minor comments

Typos: evolutionar → evolutionary (abstract line 60), GENECODE → GENCODE (throughout materials & methods), not a random → not random (line 289)

Supplementary Figure 1: The "above" and "below" plots should be in the other order, to match legend Fig 2b: Label the little box in the middle - is that unexpanded LG8? Perhaps add dotted lines to the lungfish genome, like was done for the gar?

What is the "standard repeat masking procedure"? (line 199, 202)

Increase font sizes in Figure 3 (all panels, but especially axes on 3B)

Supplementary Figure 8 has a text selection highlight left in on top panel, and no labels on axes

Error bars are so large in Supplementary Figure 9

Figure 4c - the box mentioned in the legend seems more red than "lilac"

Link to hic scaffolder github leads to 404 not found error

Would be helpful to define "Ur-vertebrate" and "Ur-karyotype"?

Fig 3A/B annotations and axis labels are unreadable, this panel needs to be fully revised it is just inscrutable.

Supplemental figure 8 has no y axis label and has remnants of a screenshot taken from inside a document editor

Author Rebuttals to Initial Comments:

General comments to reviewer's comments/suggestions (Manuscript 2020-07-12964B) major additional analyses for the revised manuscript

We are glad that all three reviewers seem to like our work and are impressed by the "scope and scale of this work that make it appropriate for a high profile publication" (reviewer 3). We obtained a chromosome-level assembly of the largest genome (43Gb = ~14x the human genome) sequenced so far, that of the Australian Lungfish *Neoceratodus forsteri* (Kreffft 1870). Since this species was described exactly 150 years ago it is a timely sesquicentennial celebration of this species' discovery to have its genome sequenced determined in 2020.

Our analyses of the genome of the Australian Lungfish resulted in a number of novel insights for genome biology. Next to the retained synteny of many of the 17 macrochromosomes we also found that all ten microchromosomes (some of which are 500Mb large, each larger than some complete fish genomes) of the Australian lungfish genome show much conserved synteny to microchromosomes of other vertebrates suggesting that also microchromosomes were a component of the Ur-vertebrate's ancestral karyotype.

The lungfishes are our closest relatives among the "fishes". This evolutionary key-position has been recovered before, but we ascertained it in our study based on two large and novel phylogenetic data sets of protein-coding genes and CNEs based on the chromosome-level assembly and therefore reliable orthology assessments that had not possible before this genome had been sequenced. In fact, long branch-attraction (LBA) artifacts continue to plague the

reconstruction of this phylogeny with long ancient uninterrupted lineages and many studies, also by ourselves (Brinkmann et al. 2004 PNAS) have been led astray by LBA. Also, we have been misled initially in this study and first seem to recover a sistergroup relationship of lungfish with the coelacanth – seemingly finding that both these sarcopterygian lineages are equally closely related to us. Only by using sophisticated CATGTR models were we finally able to recover the lungfishes as sistergroup to tetrapods.

Lungfish occupy a crucial position in the vertebrate phylogeny, therefore, its HiC-based chromosome level assembly will be a crucial reference for all kinds of future analyses of tetrapod biology. But, beyond a reference genome, we conducted a number of additional analyses that help in understanding the conquest of land and the pre-adaptations that permitted this major transition in evolution for this revised version of the manuscript. We analyzed the biology of the genome thoroughly and obtained a number of novel insights that explain both the expansion history of TEs and repeats that make this the largest animal genome and show that the many TEs are still active and the genome is still growing. The two giant genomes, that of *Neoceratodus* and that of the axolotl are composed of different “mobilomes” that differ in TE composition and dynamics of their expansion during evolution. Connecting the bioinformatics analyses with the comparative developmental work we now added an analysis of limb-specific enhancers.

We much expanded the treatment of **biological insights** in the revised manuscript. Several aspects of the biology of the sarcopterygian conquest of land were expanded on or entirely new in the revised version of the manuscript: **locomotion**: particularly the question how our sarcopterygian fish ancestors were pre-adapted to colonise land with their fleshy and branching fins, as opposed to the ray-finned fish lineage. Additional *in silico* analyses of a set of validated tetrapod limb enhancers across gnathostome genomes identified enhancers related to the fleshly-lobed fin phenotype that arose newly in the sarcopterygian lineage. Further analyses of these candidate genes led us to *sall1*, an important gene for the distal branching of the limb and involved in human limb pathologies (Townes_Brocks syndrome). This gene is reportedly absent from ray-finned fish fins but our analysis of *sall1* expression in embryonic fins of the Australian lungfish shows a tetrapod-like expression in a limb-like autopodial domain. Therefore we identify the gain of an expression domain that is phenotypically directly related to one of the critical assets that “preadapted” the sarcopterygian lineage to conquer the land. Furthermore we identify the distal expression of *hoxc13*, which importantly contributed to the adaptive potential of the tetrapod limb by allowing nails, claws and hoofs to evolve, as a novel feature that arose in the sarcopterygian lineage only and was already present as a genetic “pre-adaptation” in the fish ancestors of tetrapods (see revised Figure 5 which now also reports on enhancer evolution related to fin-to-limb transitions, and the discovery of the tetrapod-like limb expression pattern of *sall1* in the fin). In regards to **air respiration** we report on the surfactant gene family expansion, in obligate air-breathing lungfish and now in the revised version also *shh* expression in the developing lung of lungfish embryos. In this revised version of this manuscript we added in-situ hybridization data with *shh* genes in the Lungen-Anlage in stages 43-48 lungfish embryos, and found a pattern that, so far, had only been found in tetrapods - adding a new Fig. 4. **Olfaction**: expansion analyses of OR gene families, particularly in receptors for smelling air-born odors suggest that lungfish are already more tetrapod-like than „fish-like“.

We hope that this largest genome and the multitude of novel biological insights that we’ve added to the revised version of the genome analysis of this evolutionary key species will make our manuscript even more interesting to the wide readership of *Nature*.

Point-by-point response:

Referee #1 (Remarks to the Author):

The manuscript describes the new genome assembly for the Australian lungfish. The genome is compared to several other species, especially the large genome of salamanders. The findings include describing the Hox gene cluster, repetitive element expansion, and phylogenomic placement of the lungfish at the base of the tetrapods. The surfactant encoding gene expansion is quite interesting, considering the biology of the lungfish. The genome is an exciting addition to understanding vertebrate and tetrapod evolution.

There is a large mismatch between the flow cytometry and Feulgen photometry estimates of genome size and the k-mer estimate from the reads. Is the assembled genome missing an additional 15+ Gb of repetitive DNA? Certainly, the genome is missing at least 5 Gb of information given the mismatch between the k-mer estimate and the assembled size.

Response: A mismatch between flow cytometry and Feulgen photometry and k-mer estimates always raises the question which one is more correct. Such mismatches have been observed in many other cases before. For instance, the cytometry and Feulgen photometry estimates of the axolotl genome size vary from 21 to 47 Gb in the literature (5 independent studies! see <http://www.genomesize.com/search.php>) while the genome size estimates from k-mer analysis range between 33.2 and 34.8 Gb. For rainbow trout measurements from more than 20 papers go from 1.8 to 2.9 Gb, and for the house mouse are between 2.4 and 4. The data from the cytometry and photometry estimates are often from older reports that did not use modern equipment. We are not in the position to evaluate how accurate flow cytometry and Feulgen photometry are, but we have trust that the k-mer estimate, which is widely used and an established method gives a value that is closer to reality than the older cytological methods do. We find that about 88% of the genomic reads map to the genome, which is similar to high-quality datasets for other well-studied species. Even if the remaining 12% would map perfectly too, the true estimate would be $37 * 100 / 88 = 42 \text{ Gb}$, which is close to our k-mer estimate. We made some further estimates regarding expected genome size independently of the k-mers (using the average pileup depth of the alignments during the assembly process) and end up roughly at the same genome size. Of course, there will be pieces missing. However, the missing 5Gb in the lungfish genome is not a surprising number given the extreme repetitiveness of the genome and small discrepancies such as these are a common feature of large, assembled genomes.

The novelty of the phylogenomic method was unclear. The implemented method did not appear to be developed and presented as a part of this manuscript.

Response: The phylogenomic analyses were based on a set of ~700 protein-coding genes and ~10,000 bp of non-coding conserved elements. Both datasets were newly assembled and analyzed for this study. This is now mentioned in the main text. Compared to previous studies using transcriptomic data (e.g., Irisarri & Meyer 2016), our analyses are the first to address the phylogenetic position of lungfish using a robust set of orthologs obtained exclusively from whole genome comparison and an independent set of non-coding markers. In particular, non-coding data have not been previously employed to revisit this phylogenetic question. We note that these data were only possible thanks to the availability of the new lungfish genome. It is reassuring that our tree found lungfishes to be the sister group of tetrapods, in agreement with previous transcriptomic evidence (e.g., Irisarri & Meyer 2016), but we note that this relationship has continued to be challenged by some recent studies (e.g., Shan et al. 2011, Yoshida et al. 2017) and that trees with

ancient lineages generally are sensitive to methodological choices such as the outgroup or evolutionary model (Takezaki & Nishihara 2016, Irisarri & Meyer 2016).

Some outstanding questions that came to mind while reading the manuscript: How was RNA used for gaps, indels and misalignments?

Response: Both DNaseq and RNAseq data were mapped to the genome using bowtie2 or hisat2, respectively. Arguably, using RNAseq data can only fix errors in the assembly in the regions of the genome that ultimately end up in the mRNA, but the correction of exactly that portion of the genome is vital for gene annotation and especially for proper ORF identification and prediction. This is described in the Supplementary materials, section Genome Assembly Correction.

What is the BUSCO score on the gene models?

Response: The BUSCO score on the gene models is 91%, this is mentioned in the main text and details reported in the new Supplementary Table 2. We have now also calculated the completeness using the 2586 conserved vertebrate genes in addition to the 233 core vertebrate.

The rationale and need for double masking for repetitive elements was unclear. The inference of two waves of repeat element expansion seems to be based somewhat on the two rounds of repeat masking, which is insufficient evidence. However, the accumulation of substitutions shown in Figure 3b suggests two rounds.

Response: The interpretation of two waves of TE expansion is derived from the Kimura analysis we show in Fig. 3b. Our main rationale for double masking was the fact that the first round of masking (as is normally done in all genomes) generated a left-over of 13Gb of the genome as “unmasked”. We asked whether this remainder DNA had signatures of repetitiveness that was simply not detected due to the vast genome size. In fact, we found that a substantial proportion of this DNA is repetitive, representing also many previously unrecoverable LINES and unknown elements. For the revision we conducted additional analyses to annotate these TEs (see new Supplementary table 5). This approach highlights the necessity to be cautious about “classical” masking procedure of genomes and reveals that much (up to 90%) of the so-called genomic “dark matter” in the lungfish genome is repetitive. We modified the manuscript to make this more clear.

Generally, Figure 2 is unclear. Scaf01 is shown in (a) and (c), is it a macro or microchromosome? The mentioned sharp syntenic boundaries are not clear. The sorting of the lungfish scaffolds in panel (a) is unclear.

Response: In the figure legend we have now defined which scaffolds represent macro or microchromosomes. While the majority of microchromosomes are conserved, some ancestral microchromosomes were recently (as indicated by sharp syntenic boundary, pink arrowheads in Fig. 2 and Supplementary Figures) incorporated/fused into macrochromosomes such as scaf01/02. We have rephrased the text and figure legends to make this point more clearly and we provide a statistical quantification of association in Supplementary figures 6 and 8.

Is there any reason to think the axolotl and lungfish genome would have similar expansions? There is a large emphasis on the comparison between the two species because of the large genomes but given their phylogenetic placement, in my mind, there is no reason to presume that the repeat expansion would have been shared.

Response: In Supplementary Fig. 4 we reconstruct (with ML) the rate of genome size change in the vertebrate phylogeny. The independent genome expansions in salamanders and lungfish are confirmed by the maximum likelihood reconstruction of genome sizes. A priori one might have hypothesized that in the common ancestor of lungfish+tetrapods the genome expansion that is found in salamanders and lungfish in particular (but also comparatively large anuran and caecilian genomes) originated once and then repeatedly genome sizes shrank in the other tetrapod lineages.

But, it is more parsimonious that the two genome expansions, at the fastest rates (that we estimate in Suppl. Fig. 4) happened independently both in the salamander and lungfish lineages. Yet, this finding does not necessarily imply that these two genome expansions happened by the same mechanism, based on the same TE-type and same temporal dynamics. That is why we focused for part of this study on this comparison in particular and ended up concluding, which was not a priori known, that the gigantism of the genomes in salamanders and lungfish happened by different mechanisms (i.e., is based on different TE types) since our analyses identify different classes of TEs as the most abundant in lungfish and axolotl (LINE vs. LTR), suggesting that their genomes expanded by bursts of different TE classes and different temporal dynamics.

There are many writing issues which were distracting, including misspelling of evolutionary and phrases such as “hugely larger”, “including besides”, etc. The first time CNE is define the word non- coding is excluded. The phrase “indicating the existence of island of genome gigantism” is unclear.

Response: We have carefully revised the text for mistakes. The unclear phrase was omitted.

Referee #2 (Remarks to the Author):

In this manuscript, Meyer et al describe their findings resulting from sequencing the genome of the Australian lungfish, making it the largest vertebrate genome sequenced to date. They confirm the lungfish status as sister group of tetrapods and show that their chromosomal make up contains the ancient chordate linkage groups and can be explained by the 2 rounds of genome duplication experienced by jawed vertebrate lineage. Lungfish genome is diploid and contains microchromosomes highly syntenic with those in chicken and Gar. The lungfish genome expanded via accumulation of transposable elements in intronic and intergenic regions; the principal category being LINES in two waves that happened independently from that observed in salamander genomes. The general composition of TE is most similar to that seen in tetrapods. Surfactant and Vomeronasal Receptors experienced expansion in the number of protein coding genes. Finally, the authors examine Hox gene cluster composition and expression and propose that Hoxc13 expression domain in the developing lungfish fin is homologous to interdigital expression seen in the mouse. The authors also show that posterior Hox genes such as Hoxd13 have been “pushed” away from other genes in the cluster and propose that, as a result, distancing may have caused loss of influence of late-phase regulatory elements on Hoxd11 gene expression, explaining lack of digital expression of this gene in lungfish and axolotls.

In all, the manuscript is an important contribution to our understanding of the evolution of vertebrate genomes and potential preadaptations to the vertebrate water-to-land transition.

Whereas the findings are of broad interest and the experiments and analyses are largely sound, I outline below issues that require attention.

In the Summary:

- First line is confusing. "Sarcopterygii" no more left the water than Osteichthyans did. One lineage within sarcopterygians radiation did. The way it's written implies lungfishes left the water **Response:** This sentence was reworded to make clear that only one lineage of the Sarcopterygii left water and gave rise to tetrapods.

- Line 55, “...mechanisms than salamanders, the other vertebrate lineage with enormous genomes”. I would recommend saying “another vertebrate lineage with enormous genome”, as the “enormous genome” category is not well defined.

Response: changed as suggested.

Line 60: word “evolutionary” is missing the “y”.

Response: corrected

Introduction:

Line 70: This sentence is incorrect as written. The clade is not "freshwater fish." Much of the lineage's history includes occupancy of marine environments. Living representatives, however, are freshwater.

Response: We reworded the sentence to make it clear that not all extinct lungfishes lived in freshwater. We meant to say that the extant lineage are only found in freshwater.

Line 73: “...initially thought not to be fish but amphibians”. Please provide citation.

Response: Citations have been added (Kreffft 1870 and Gunter 1871).

Line 79: “and is one of the oldest genera”. Please provide citation.

Response: A citation has been added (Kemp 1986).

Line 83: “...filaments that do not aid in locomotion^{1,4,5}”. This statement is incorrect. Please see and cite King et al. (PNAS 2011), and Aiello et al. (J Exp Biol. 2014).

Response: Both citations have been added indicating that even the pelvic appendage-filaments show bipedal-like locomotory patterns

Line 88: Marjorie Courtenay-Latimer’s name is misspelled in the text.

Response: corrected

Line 92: “...it is now possible to revisit and resolve this long-standing crucial evolutionary question using whole- genome-derived datasets”. Paleontological and previous molecular work up to the coelacanth genome (Amemiya 2013 Nature) have reached this conclusion before. I am not convinced this is a “long-standing” unresolved matter. Consider toning down the statement.

Response: The statement was toned down. It now reads: “For seven decades (since the sensational discovery of a living coelacanth by Ms. Marjorie Courtenay-Latimer in 1939 in South Africa), the relationship among these three living lineages was hotly disputed. Due to the short internode dating back almost 400 million years that separates the three lineages and long-branch attraction artifacts due to these long uninterrupted lineages, it remained difficult to resolve these relationships. Since about 10 years (Amemiya 2013, Irissari and Meyer 2016, Irissari et al. 2017) the hypothesis of lungfish being the sister lineage of tetrapods was most strongly supported.”. We also write in the Results part: “The phylogenetic relationships among coelacanths, lungfishes and tetrapods have been a long- standing controversy. Recent phylogenomic analyses of transcriptomic data favored lungfish and not coelacanths as the closest living relatives of tetrapods. Our high-quality genome of the Australian lungfish now allows us to revisit this question using whole genome derived orthologous protein sequences and conserved non-coding regions. “ Reviewer 2 is correct to point out that the close affinity of lungfish and coelacanth has been already supported by some previous studies. However, the lungfish genome newly allowed us to revisit this question using both non- coding elements (Supplementary Fig. 2) and a new set of protein orthologs derived exclusively from whole genomes (thereby avoiding biases in transcriptomic data such as missing or alternatively spliced genes, Fig. 1). However, we would like to highlight that the close affinity of lungfish and tetrapods has continued to be challenged by some recent studies (see response to Reviewer #1). We point out that “ Recent phylogenomic analyses of transcriptomic data favored lungfish and not coelacanths as the closest living relatives of tetrapods” and “These (i.e. our) results, together with the previous transcriptome-based analyses

unequivocally support lungfish as the closest living relatives of land vertebrates that last shared a common ancestor with tetrapods about 420 Ma”. We make reference to the relevant publications. We believe that with our new analyses, that was possible with the genome sequence in hand, this chapter can be closed and ends the discussions.

Results and discussion

Line 108: “But lungfish genomes are even bigger”. Please provide citation.

Response: We make now reference to the animal genome database which is the most comprehensive source for such information. We addressed a similar point raised by reviewer 1 who questioned the accuracy of flow cytometry data.

Line 146: “...that has been shown to overcome phylogenetic artefacts”. Please specify which artifacts.

Response: We have specified this now and say: “...we performed a Bayesian phylogenomic analysis (Figure 1) with a complex mixture model that has been shown to overcome long-branch attraction artefacts (Irisarri & Meyer, 2016).” Those in the past ended up resulting in wrong phylogenetic inferences.

Fig 1: Dedicating this figure to Jenny Clack and Robert Carroll is quite nice. Nevertheless, I feel I should point out that this may be a sensitive issue. The authors should very carefully consider the optics of only including white European scientists as representatives of the human species.

Response: We have removed the pictures of Jenny Clack and Robert Carroll (keeping the dedication of the manuscript to them). Unfortunately, even after extensive searches of the internet we did not find a picture or symbol that represents all humans and is neutral in every aspect. Thus we decided to leave the human branch in the phylogenies without a picture or symbol.

Line 166: “The lungfish chromosomal scaffolds show a remarkable conservation of the ancestral chordate karyotype”. What is remarkable about the amount of conservation? Is it greater than expected? Please clarify or reword.

Response: One might have expected that due to the massive genome expansion much of the syntenic signal would be diluted or gone. The fact that we find significant association of chordate linkage groups to specific lungfish chromosomes, provides evidence of strong selective pressure to keep those linkage together. This is especially true for the highly expanded but still retained microchromosomes – together the lungfish microchromosomes are on the order of several Gb, which is larger than most animal genomes. To further address this, we now provide statistical quantification (new Supplementary Figs. 6, 8) of the overall conservation. In addition we have removed the word “remarkable”.

Line 180: “This suggests, by comparisons to gar, anolis, and chicken, that a typical set of microchromosomes existed in the ancestral vertebrate”. This conclusion has been reached before, see the Gar genome paper. Please rephrase, for instance, indicating that the results “confirms previous reports”... And cite Braasch 2016 Nature Genetics.

Response: We have now rephrased this to indicate the previous knowledge: “While previous reports found conservation of some microchromosomes in gar, chicken and green anole²³, which suggested that a limited set of microchromosomes may have existed in the ancestral vertebrate, the retention of all microchromosomes in lungfish, despite the genome expansion, highlights the selective pressure to maintain those ancestral units” and make reference to the Braasch et al. 2016 paper. Please note that Braasch et al found conserved synteny for only two microchromosomes of gar and chicken while we report the synteny of all 10 microchromosomes.

Line 181: *Anolis*, as a genus name, should be capitalized, italicized.

Response: corrected, to be consistent in this sentence we choose to use the common name “green anole” instead.

Line 228: “We find that the global repeat landscape (overall amount of the major TE classes)...” the amount as a proportion of the genome or by bp counts? Please clarify.

Response: We are sorry for the confusion. What was meant here was “proportion of the genome”. The sentence has been changed for increased clarity.

Line 231: “Interestingly, the two largest animal genomes, lungfish and axolotl...”. Given that extant lungfish species vary greatly in their genome size, I recommend specifying Australian lungfish as opposed to using “lungfish” to refer to all species.

Response: changed as suggested

Line 256: delete the word “besides”. And consider revising the entire sentence. As is, it is very confusing.

Response: done

Fig 3a: Some categories on the pie charts are described in minuscule font unreadable if printed.

Response: Fig 3a has been revised using bigger lettering for labelling the pie chart categories.

Line 279: “The conserved element (CNE) dataset...”. Missing the word non-coding.

Response: corrected

Line 272: “random process of repeat, transposon and junk DNA expansion”. I suggest avoiding using the term “Junk DNA”, as it presumes, without evidence, absence of function.

Response: the term “junk” was omitted and the sentence reworded to “sequences with no obvious function”

Line 301: “This approach allowed us to identify 308 genomic regions with signatures of accelerated evolution in lungfish”. This is interesting. Were some examined more closely? Any with potential relevance for lungfish physiology/evolution?

Response: We intersected the accelerated lungfish CNEs with a set of experimentally validated human and mouse noncoding fragments with gene enhancer activity, as assessed in transgenic mice (data from “VISTA Enhancer Browser”: <https://enhancer.lbl.gov/>). We found that our accelerated CNEs intersected two enhancers: “hs2142”, flanking a Plasminogen activator (PLAU) and a Vinculin gene (VCL) gene, with expression domains in forebrain, heart, hindbrain, midbrain and somite; “hs607”, flanking a Glutathione S-transferases (MGST1) and a LIM domain (LMO3) gene, with expression domains in hindbrain and neural tube. These results are now reported in the main text and described in the Methods sections.

Line 345/346: “...high expression of posterior *hoxa* and *hoxd* genes (except for *hoxa14* as previously reported 42,43”. Reference 42 is a manuscript in press and therefore yet unavailable. Without access to the data I cannot evaluate the statement regarding *hoxa14*.

Response: Previous reference 42 refers to the expression of *hoxa13* and *hoxd13* only - the reference for *hoxa14* is now published: Woltering, J.M., Irisarri, I., Ericsson, R., Joss, J.M.P., Sordino, P., and A. Meyer 2020. Sarcopterygian fin ontogeny elucidates the origin of hands with digits. *Science Advances*. 6: eabc3510. This has been updated.

Line 354: This large section is a bit difficult to follow, but if I understood correctly, the hypothesis is that genome expansion promoted a distancing of *Hoxd13* from the minicluster of genes

spanning *Hoxd8-d12*. This in turn, resulted in maintenance of regulatory influence of late-phase *hox* enhancers only over *hoxd13*. This would explain why *Hoxd11* in axolotl and lungfish are not expressed in the same domain as *d13*.

Response: Yes, this is what we wanted to say. We have shortened and rewritten the large section to make it more clear.

Regarding this: Reference 42 (paper in press) is cited for *Hoxd11* expression in lungfish fins. As I cannot access the primary data (only the diagram from fig 4), I cannot critically evaluate the claim. Can the authors share an image of *Hoxd11* in situ in lungfish?

Response: The reference for *Hoxd11* expression is now published: Woltering, J.M., Irisarri, I., Ericsson, R., Joss, J.M.P., Sordino, P., and A. Meyer 2020. Sarcopterygian fin ontogeny elucidates the origin of hands with digits. *Science Advances*. 6: eabc3510

If this loss of regulatory influence is correct for both lungfish and axolotl, it would presumably impact other *Hoxd* genes such as *d12*, *d10*, *d9*. In reference 49, only Axolotl *d11* and *d13* were examined. If the authors have in situ data for other *Hoxd* genes showing similar exclusion of digit (axolotl) or distal fin (lungfish) expression, it would strengthen the claim.

Response: Published data is available for *hoxd8* (and similar conclusion for *hoxd11* in this paper) in Bickelmann et al. 2018. These two genes show a similar strong proximal expression, but absent distal expression, which for *hoxd11* is very different from the canonical patterns reported in mouse and chicken. Material scarcity in the lungfish has precluded the analysis of additional genes. However, we have now provided *in situ* hybridization data for *hoxd9* and *hoxd10* in developing axolotl limbs in Supplementary figure 21 and figure below (please note that axolotl has no *hoxd12*). Both genes show strong expression in the proximal limb and are near absent in the digit domain where *hoxd13* is strongly expressed. These non-canonical expression domains contrast sharply with the expression in mouse where *hoxd9*, *hoxd10* and *hoxd11* are strongly co-expressed with *hoxd13* in the digits. Such "proximal-limb-only" expression is known from mouse mutants in which the 5' gene desert containing the distal limb enhancers has become experimentally displaced (e.g. Spitz et al. 2005, relevant data shown in figure below). Therefore expression domains found in the limbs of mice with an experimentally induced condition resemble those of the axolotl that has naturally evolved an analogous genomic organisation through the posterior expansion of its *hoxd* cluster. Therefore, given the available data, it is justified to interpret the non-canonical expression of the axolotl *hoxd9-hoxd11* as the result from this analogous "pushing away" of the 5' limb enhancers.

The remaining distal expression, which appears stronger for *hoxd9* and *hoxd11* than for *hoxd10*, likely results from remaining expression of the proximal phase of expression which initially extends into the distal limb domain but becomes subsequently silenced through trans-regulatory interactions with the *hoxa13* and *hoxd13* proteins, that repress the *hoxd8-hoxd11* part of the cluster (Beccari et al. 2016). In mouse, the renewed activation of *hoxd9-hoxd11* in the distal domain is the result of the co-activation of these genes together with *hoxd13*. It is this latter phase of re-activation that in axolotl appears to be absent for *hoxd9-hoxd11*.

Response: Please note that we do not claim that the enhancers cannot act over long enough distance to reach their target genes. From the expanded organisation of the axolotl and lungfish genomes it is clear that such enhancer- target gene interactions must occur over longer than normal circumstances, as is also shown in the accompanying manuscript of the Tanaka group (that we uploaded as additional information for the reviewers). What we expound here is that the enhancer sharing, by which a regulatory region simultaneously contacts several genes at the same time, is affected when the distance between several target genes is increased. In fact, such enhancer sharing is an accepted explanation for why the *hox* genes have remained clustered during

evolution. In the axolotl and lungfish genomes we identify a remarkable exception on the vertebrate pattern of *hox* clustering (expansion of intergenic *hoxd11-hoxd13*), as well as striking evidence for constraints (clustering of *hoxd8-hoxd11*). Both of these genomic patterns can be directly linked to conserved aspects and departures in the expression of these genes. Additional experiments that would provide detailed insight into the mechanisms and constraints acting in these expanded clusters such as HiC are outside of the scope of this manuscript.

Top panel, expression of *hoxd9*, *hoxd10*, *hoxd11* and *hoxd13* in developing axolotl limbs. *Hoxd11* and *hoxd13* images were taken from Woltering et al. 2019, similar data for these genes as well as for *hoxd8* was published in Bickelmann 2018. Lower panel, the absence of distal *hoxd10* expression in axolotl resembles the expression pattern in a mutant mouse model in which the 5' enhancers were experimentally displaced. Therefore, analogous genomic conditions in mouse (experimentally expanded) and axolotl (naturally expanded) result in similar expression domains.

Conclusion:

Line 421: sentence needs revising.

Response: This sentence has been revised.

Materials and Methods

- In Biological Materials section, specify where did the embryos come from and, if pertinent, permits for breeding.

Response: the information has been added to the Biological Materials section.

Also, specify if the animal used for genome sequencing was male or female. Is there sexual dimorphism in genome size -- sex chromosomes? Does that impact the results?

Response: We do not know the sex of the sequenced individual. Due to the immature stage of the gonad we could not determine if it was male or female (age at maturity is about 80cm). This is now mentioned in the materials section. There is no information on a difference of genome size between male and female genomes. The cytogenetic study (Rock et al, 1996) of eight specimen did not uncover heteromorphic chromosomes that would indicate differentiated sex chromosomes. Thus, there is no sexual difference in genome size to be expected.

Supplementary Figure 2: I suggest also adding Jenny Clack to this phylogeny.

Response: We have omitted any photos or graphic representation for “human”, see comment above.

Supplementary Figure 11: The word intron in the figure legend is misspelled.

Response: corrected, now Supplementary Fig. 14.

Supplementary Figure 11: “(Y-axis is log scale of bp s.”. Close the parenthesis and remove “s”).

Response: the text of the figure legend was edited to correct for typos, now Supplementary Fig.15.

Referee #3 (Remarks to the Author):

The authors describe the sequencing, assembly and analysis of a lungfish genome. The work is technically impressive and full of interesting analyses that shed light on the evolution both of large genomes and also early vertebrate evolution, both phylogenetically and from a genome architecture perspective. It is also nice to see it confirm some of the observations made in the analysis of the axolotl genome about the evolution of large vertebrate genomes. The scope and scale of this work make it appropriate for a high profile publication. Practically, it is worth mentioning that the paper is very large and it was challenging to review comprehensively. What follows is our best attempt!

Comments on the assembly:

The assembly created is impressive! A 2.2GB scaffold N50 is amazing. However, there is little evaluation of the resulting assembly quality. The paper says “88.2% of the DNA reads and 84% of the RNAseq reads aligned to the genome.”, but this analysis is not (as far as I could find - I may have missed it) described in the methods, so it is unclear how the reads were aligned, if this was all the long-reads and short-reads, and if the reads were aligned with high-mapping quality.

Response: Thank you for your comments and appreciation of the assembly. Yes it is hard to concisely but comprehensively describe all necessary information to evaluate the genome and we are thankful for the chance to address your specific queries. We would first like to address an embarrassing mistake we have made in the main text. The scaffold N50 was stated as 2.2 Gb, whereas it has been now finally determined to be 1.75 Gb. At this level of contiguity and low number of scaffolds the N50 has a tendency to jump fairly quickly if even a single scaffold larger than the N50 is modified. The 2.2 Gb that was reported in the manuscript was based on the second to last set of scaffolds, which are essentially the same as the ones included in the manuscript, but which underwent a final pass of breaking full length chromosomes at centromeres, at which the join of the arms was potentially ambiguous.

The metrics of “88.2% of the DNA reads and 84% of the RNAseq reads aligned to the genome” refer to the mapping rate of the 30x coverage 150 bp Illumina reads that were used for genome correction to the Nanopore contig assembly, and of the RNASeq datasets that were mapped to the Illumina-corrected assembly respectively.

For the DNA reads, we used Bowtie2 in the “-very sensitive” mode, which influences the parameters like the seed length and the number of re-tries for mapping. We used the default parameters for the alignment score threshold, which is -90 for the given read length, and would allow the reads with up to 16 mismatches per 150bp to still be aligned. Examination of the distribution of scores showed that 55% of the reads mapped with 0-3 mismatches per 150 bp. Based on our experience with the axolotl genome, we expect the uncorrected contigs to have an

error rate of around 2%. This base error rate varies along the genome depending on the Nanopore coverage at any particular site.

For the RNASeq mapping, we used Hisat2 with the default alignment score threshold of -30, which roughly corresponds to up to 6 mismatches per 150 bp but 91% of the reads mapped with up to 3 mismatches per 150 bp. This text has now been included in the Materials and Methods.

Furthermore, it is surprising given the repetitiveness of the genome, that there is no evaluation of misassemblies or assembly collapses. I would very much like to know how many of the genes assembled were subject to assembly collapse. Given that the authors strived for a high-quality analysis, it would be good to see some more in-depth, honest analysis of the assembly and its limitations.

Response: Thank you for the comment. In this and the following responses, we describe how the assembly process and assembly has been checked for errors. In general, the conserved mapping of the chordate linkage groups suggests an overall structural soundness of the assembly and scaffolds.

Assembly collapse (i.e. repeat collapse) is not an issue using OLC (overlap layout consensus) assemblers, and MARVEL belongs to this family of assemblers. The repeat annotation generated during the assembly process guarantees that reads entering repeats are properly annotated and contigs terminate if there is no unique path leaving the repeat. Additionally, for this assembly we didn't use any sort of heuristic to resolve repeats, which can occasionally result in repeat collapse.

Mis-join detection is always an issue especially when there are no assembled close relatives available or the mis-joins are very "subtle" in nature. We developed an experimental detection tool that uses the HiC in order to detect contact arrangements that are indicative of mis-joins. When running this on the final contigs we detect 1583 potential mis-joins that would require extensive further manual validation to be eliminated. However, as described below, Bionano is not an appropriate means for such validation. This information is now included in the Methods section. Please, see also our response to your query "The scaffolding process in particular seems to take into account the difficulties" below for more detailed information on scaffolding and mis-join detection, and the Bionano issue.

As one way of assessing the quality of genome contigs and scaffolds, we examined the alignment of the de novo assembled transcriptome contigs to the genome contigs and the genome scaffolds. 260,478 out of 283,820 contigs align to the genome contigs using GMAP, while 253,781 align to the scaffolds. This small differential is due to genome contigs that were not included in the scaffolds.

The new Supplementary Fig.25 shows the proportion of the transcript sequence covered by the alignment to the contigs (Supplementary Fig.25a) and scaffolds (Supplementary Fig.25b) on the X-axis and the number of transcripts that have this coverage on the Y-axis. Most of the transcripts align fully to contigs and scaffolds as can be seen from the large peak at 100% in both figures.

Transcripts that did not align completely to the contigs but did either align completely or to greater extent to the scaffolds are of particular interest. Supplementary Fig.25c, d show the distribution of the 17,549 transcripts that had partial alignment to the contigs and improved alignment to the scaffolds. These plots show that the scaffolding shifted the coverage of many transcripts to 100%, validating the accuracy of the scaffolding.

Supplementary Fig 25: Transcript coverage by genomic sequences. **a.** Histogram of the proportion of all transcript lengths covered by the alignment to contigs. **b.** Histogram of the proportion of all transcript lengths covered by the alignment to scaffolds. **c,d.** Histogram of the proportion of the transcript lengths covered by the alignment to contigs (**c**) or to scaffolds (**d**) of the transcripts whose alignment was improved after scaffolding.

As a final assessment, we specifically focused on roughly 4400 transcripts that showed 40-60% alignment to the contigs. Of those approximately 1208 did not show marked improvement in alignment to the scaffolds. We wanted to determine whether this might be due to scaffolding problems, or due to artifactual contigs in the de novo assembled transcriptome – a well-known problem. When we examined these 1208 further, only 478 were longer than 500 bp, and therefore the majority of those 1208 sequences most probably represented unidentifiable bits and pieces that are known to plague de novo transcriptome assemblies. We examined the longest transcript contig (10kB) in this class, and extensive BLAST alignment of this transcript and its sub-parts to NR showed signs of a mis-assembled transcript contig, another hallmark of de novo transcriptome assemblies. Although we cannot state that all genome contigs were correctly placed, we conclude from this analysis that the genome scaffolds show very good accuracy in ordering the genome contigs.

How many chromosomes were there predicted to be via karyotype and other prior work?

How many macrochromosomes vs microchromosomes, and how big are each? (Is the huge final scaffold N50 of 2.2 Gb reasonable / expected?)

Response: Rock et al 1996 give a $2N=54$. These 27 chromosomes, when ordered by size, appear to include 3 very large, 1 large, 13 small, and 10 microchromosomes. Given their Fig. 1. and an expected genome size of 42-44 Gb the assumption that around half of the genome is represented by the very large and large chromosomes is reasonable, thereby making the scaffold N50 not unexpected. The N50 was corrected to 1.7 Gb, see above.

To facilitate evaluation, we provide here and in the manuscript a table of expected chromosome DNA content, and the DNA content of our 39 scaffolds. We expect this set of 39 scaffolds to represent some entire chromosomes, some chromosome arms, microchromosomes, and unincorporated scaffolds. A meiotic map is not available to specifically assign the scaffolds to chromosomes, but the synteny observed with the chordate linkage groups provides some anchorage to the data. Chromosome sizes (Gb) were estimated by physically measuring the area occupied by each chromosome in the karyotype shown in Fig. 1 Rock et al. 1996 and calculating the fraction of the total assuming a 43 Gb genome. This was compared to our chromosome-level assembly scaffolds, ordered by size.

What are the estimated sizes of the true chromosomes in lungfish, and for each chromosome what is the distribution and ordering of assembly contig sizes?

Response: This can at best be guessed from the karyotyping in Rock et al 1996 due to the lack of individual chromosome sizes. Based on the estimated genome we estimate the 3 very large chromosomes to be around 5 Gb each, the large around 3 Gb, the 13 small in the range of 1-2 Gb each and the 10 microchromosomes at around 500 Mb each.

Also, your assembly N50 may be accurate because of the size of the genome, but it would be good to know if your scaffolds agree with previous estimates of chromosome size by Rock et al 1996 or others (if the individual chromosome masses are available)

Response: See the two responses above. Given the karyotyping and the scaffold sizes we believe these to be in agreement.

Perhaps include an N(x) plot of the assembly (pre/post scaffolding) to get a sense of the distribution of contig and scaffold sizes.

Response: N(X) plots for the contig assembly and the scaffolded genome have been added, see new Supplementary Figs. 20 and 22.

What is the heterozygosity of the genome? There is no heterozygous peak on the kmer count figure (Supplementary Figure 1)

Response: Using the genomescope (<http://genomescope.org/>) analysis the heterozygosity is estimated to be only 0.0965-0.1278 %. One likely explanation for this low level of heterozygosity could be inbreeding as the species is highly endangered and, thus, the population size is quite small (Hughes et al. PLoS One 2015 <https://doi.org/10.1371/journal.pone.0121858>, Bishop et al, Conservation Genetics 19, 2017 DOI: 10.1007/s10592-017-1034-7). We do not detect a secondary peak in the kmer data to estimate heterozygosity. We believe this is for two reasons—one technical and the other biological. Based on our experience working with smaller genomes, a ≥ 100 -fold coverage of the genome is needed for the peak to become detectable in organisms with a reasonable heterozygosity rate. Our lungfish dataset represents 30x coverage, (which represents a huge sequencing volume), and therefore our sensitivity to detect this peak is reduced. Second, as mentioned above, we expect this species to have a small population size and high inbreeding, so we expect the heterozygosity to be low which further reduces the chances to see the heterozygosity peak in the kmer count data.

The scaffolding process in particular seems to take into account the difficulties of scaffolding such a highly repetitive genome, but the final scaffold N50 of 2.2 Gb seems potentially too good to be true, particularly given this was done with only 7.55x coverage of HiC data. Are all of the joins correct?

Validation via an additional sequencing type (such as BioNano) would be good. How many gaps are there in the final assembly? Could a final round of gap closing be done, if it was not already?

Response: Thanks for the appreciation of and interest in the scaffolding efforts. We describe below how we determined the HiC coverage to use for scaffolding and address the Bionano validation.

Thinking about an appropriate “coverage value” for the HiC data is a bit different than for all the other datasets used for assembly and error correction. First this is because HiC involves selective precipitation of cross-linked and sheared entities, so HiC libraries sequence only those parts of genomes where contacts take place. Therefore, the true “coverage” of those

parts of the genome is higher than a calculation of a blanket coverage of the genome. Second, the coverage of HiC required to accurately scaffold a genome is highly dependent on the lengths of the contigs input into the scaffolding process. This is because the frequency of contacts between two contigs is used to assess the likelihood they are neighbors (and the relative frequencies found in the two halves of a contig give orientation information). Long contigs will have many contacts while short ones have fewer. In the extreme case, let's say that our base assembly consisted of three very long contigs that cover the sequence of an entire chromosome. We would have needed "1x coverage" or less of the chromosome in order to correctly orient and order these three contigs because so many sites of contact would have existed between the contigs. We have therefore removed "7.55x coverage" from the text describing the number of HiC reads.

We gained our scaffolding experience in preceding work HiC scaffolding the 30 Gb axolotl genome (manuscript in submission, available upon request). There, we used roughly four-times the number of contacts to scaffold, but we also had an eight-times lower contig N50 of 216 Kb. We could validate the axolotl assembly, as the sizes of the scaffolds correspond to expected chromosome number and sizes, and gene order correspond to available meiotic mapping data. We have also extensively worked with the gene models in the axolotl assembly and have observed very few problems with gene structure.

Prior to generating the lungfish HiC data we used the lungfish contig N50, its expected genome size and the experience from scaffolding the axolotl as a guide on how much to sequence to generate high-quality scaffolds. Admittedly, more data would have resulted in fewer unassigned sequences (1.97 Gb). But this quickly results in diminishing returns due to the unassigned sequences comprising ever smaller and more highly repetitive sequences.

There are 48505 gaps at the contig joins in the scaffolds.

Important to the scaffolding procedure is a misjoin detection tool in MARVEL that allows for visual inspection of the scaffolds by means of the contact map. This allows the exclusion of larger structural defects.

In terms of using Bionano to validate scaffolds, here again, our experience with the axolotl genome guided our perspective on this. The original axolotl genome assembly reported in Nowoshilow et al. (2018), was scaffolded using Bionano. When our axolotl HiC data became available, we initially scaffolded the Bionano scaffolds, but then analysis of the HiC contact maps indicated the presence of a number of intra-/inter chromosomal misjoins in the Bionano scaffolds which perturbed our ability to accurately scaffold the genome. When HiC scaffolding, any Bionano misjoin is then carried along and amplified by HiC so we found it was far better to scaffold the base contig assembly. An interesting aspect of the HiC data, is the sensitivity to detect misjoins. A misjoin can often result in a kind of chain reaction effect which becomes easily visible in the contact maps. Therefore, the misjoin detection tool seems to be a quite sensitive way to detect misjoins.

So, in the end, we concluded that the HiC data are more reliable, and ended up being a tool to assess the Bionano data, rather than the other way around. It is likely that in such repeat-rich large genomes, Bionano is prone to more misjoins than in standard organisms due to a higher chance of similar restriction site patterns.

Gap filling was not attempted. Given the way the assembly was conducted there is fairly low

chance when mapping the reads to the assembly to add additional sequence between the contigs. Proper gap filling would require additional ultra-long-read sequencing. Given the giant genome size, the current project involved an enormous amount of sequencing, and such gap-filling efforts represent future polishing efforts under different funding schemes.

What is an estimate of the final Q value of the assembly? (Could estimate via kmers with a tool like Merqury) **Response:** We computed a Q-value of 26 for the lungfish assembly. For comparison, the axolotl Q-value, based on our current unpublished scaffolds, is 30. The slightly lower number observed for the lungfish assembly can be attributed to the increased repeat content over the axolotl genome, leading to additional regions in the genome that can only be partially corrected. This is due to ambiguities in the mapping of the Illumina reads to such repetitive regions. How the unassembled parts of a genome influence the Q value computation is currently unknown.

More detail is needed in the description of the methods for the genome assembly, which is currently very brief. How was the 1%-against-all alignment of reads used to find repetition?

Response: The text in the supplement has been expanded and hopefully makes this a bit clearer. Overall we would also like to point to the linked sample execution scripts which can serve as template to run the assembly.

What are key features of the MARVEL assembler? (we ended up finding a description in the Supplementary Information section of the axolotl paper, Nowoshilow 2018).

Response: MARVEL has a modular design, which in essence is a tool-kit that allows fine control over each step of the assembly process which is necessary to adapt the assembly procedure for highly repetitive, large genomes. Other assemblers that treat repetitive sequences as comprehensively as MARVEL typically do not run in reasonable timescales on such large genomes. Another characteristic feature for the long-read assembly part is the read-patching procedure which uses other reads to patch-correct stretches of artefacts and maintains read length rather than cutting the reads at those regions as many other assemblers have done.

Several aspects of the MARVEL HiC scaffolder are notable. First, the algorithm runs very quickly, so it was possible to rapidly go through scaffolding attempts and evaluate their quality, even on such a large genome. These iterations allowed us to learn that the axolotl Bionano scaffolds were not ideal for HiC scaffolding. Second, a normalization procedure for calculating the contact frequencies between contigs was developed. This is very important in the landscape where most of the genome consists of contact-depleted repetitive sequence, which has a strong effect on contact frequency calculation. Third, we developed a misjoin evaluation tool to detect and aid in the resolution of misjoins.

How is repeat masking used during the read correction, read overlap, and graph layout stages of assembly? **Response:** This is a very technical request that would require its own document. We are happy to provide a document as a separate publication, or better yet, discuss these details at an appropriate point.

Why was the MARVEL assembler used as opposed to any of the other long-read assemblers out there (such as Flye which is generally well-suited for handling repeats)? This is easily justified, but worth commenting on.

Response: In Nowoshilow et al (2018) we made a direct comparison of CANU and MARVEL on 30x Arabidopsis and MARVEL outperformed CANU in contiguity, (Suppl.

Fig 11). More recently, a colleague benchmarked FLYE, CANU, SHASTA, WTDBG2 and MARVEL using a high coverage Nanopore dataset of the drosophila genome aiming to achieve very accurate and complete assembly particularly of the highly repetitive piRNA cluster regions. CANU, which can treat repeats comparably to MARVEL did not finish within the given time frame. FLYE and WTDBG2 assemblies showed defects in chromosome 3 and 2 respectively. The assembly from SHASTA generally looked very good except the piRNA regions and was outcompeted for accuracy/contiguity in the piRNA cluster region by MARVEL. In conclusion MARVEL has so far provided the best performance in difficult-to-assemble, repeat-rich genomes. Specific metrics and figures are available upon request.

Perhaps include a figure/flowchart of some sort to help with the explanation of the HiC scaffolding method? It is a bit hard to follow in the text.

Response: We added a flow chart to the supplement (new Supplementary Fig.21).

What were the runtimes of the assembly, scaffolding, and polishing stages? How much disk space/memory is required?

Response: For correction, 842G (743Gb DNaseq + 99Gb RNAseq) are required for the mapping data (bam files). Additionally, 493Gb and 1.5Tb are required for pilon output for DNaseq and RNAseq, respectively. The runtime is hard to estimate since the data had to be split into 300 pieces, which were processed in parallel. Each job took about 20-30min to complete using a single core and 70Gb of RAM each. The assembly used at peak 1.5TB of disk space and was performed on our in-house compute cluster with nodes averaging at around 128GB of main memory and 16-32 CPU cores. Due to upgrades to our IT infrastructure the assembly had to be moved at someone point to our new cluster making exact runtime measurements hard to come by. Our best estimate would be roughly 80.000 CPU/hours. The computational part of the scaffolding is not CPU intensive at all and was performed on a standard desktop workstation. The manual inspection of the contact maps and currently unautomated task of joining of chromosome arms, which usually are put into separate cluster at the beginning of the process, take their time though.

Comments on the alignment and conservation analysis:

The generated multiple sequence alignment relies upon reference guided alignment to the evolutionarily extremely distant human reference genome, which is separated from the lungfish by well over one neutral substitution per site! This must inevitably mean that only very highly conserved elements, such as core genes and conserved non-coding elements, will be present in the alignment. Indeed, despite the genome of the lungfish being more than 30 billion bases long, just less than 100 thousand bases (1/300,000 bases of the lungfish!) were used for the phylogenetic tree building from the WGA. While this is probably sufficient for the purposes of establishing the ancestry relationships, it is worth mentioning that these highly, highly filtered elements are not representative of a neutral evolutionary process, and so it is possible that they could create misleading phylogenetic reconstruction, certainly, the inferred branch lengths must be very questionable.

Response: We have seriously considered this potential limitation for the revised version of the manuscript. However, as reported in the text, we used a stringent pipeline to remove all coding alignment blocks and keep only non-coding

elements that should better represent a neutral evolutionary process. The phylogenetic relationships inferred with this dataset were used to provide an additional line of evidence to support the phylogeny built with the protein-coding genes (Fig. 1), and only the branch

lengths from this latter tree were used for downstream analyses. That said, we found a high correlation between the branch lengths estimated with the two different approaches ($R^2=0.84$; $p<0.05$), which we now mention in the legend to Supplementary Fig.2. This suggests that both protein and non-coding regions are likely to represent the true variation of evolutionary rate between the studied species.

Similarly, I am suspicious that the use of 4d sites to estimate neutral rates for phastcons for this phylogeny will be heavily biased by the lack of truly neutral evolution in the elements (even degenerate codon positions) conserved in this alignment - how many sites were these estimates made from? Were there enough sites to estimate a model for? **Response:** We see the point of the reviewer. However, just because our dataset contains very distant species, which align predominantly in coding sequences, the estimation from the non-conserved model from 4d sites in coding regions is a common procedure (and suggested in the phyloFit tutorial). In the phastCons paper (Siepel et al. 2005. Genome Research 15: 1034-1050) the authors showed that the 4d method is better suited for distant species because it is less sensitive to alignment biases than their normal "unsupervised" (EM) learning method. They reported that the branch lengths for distant species (e.g., human-zebrafish) are systematically underestimated, but that nevertheless the 4d method can readily discriminate between conserved and non-conserved regions, which is what we wanted to achieve in our study (rather than inferring the models very accurately). In fact, they showed that the distortions of the distant species (which are comparable to our taxon sampling) do not have a pronounced effect on the elements that are predicted.

Because many coding regions were recovered in our 10-way multispecies alignment, we were able to estimate the non-conserved model with a reasonable number of fourfold sites (from 1,913 to 15,716 4d sites depending on the chromosome the model was estimated for). These numbers are now added in the Material and Methods section.

I was surprised to see that you ran a different model for each human chromosome, which only vaguely resembles the ancestral synteny structure of the MRCA of these species. How many sites did you use, for example, to estimate the rate for human chromosome 22!? I'm guessing the interchromosomal variation will be a good reflection of the variance in the estimate of these rates.

Response: As correctly pointed out by the reviewer, the karyotype of the human genome does not strongly resemble the ancestral synteny structure of the most recent common ancestral genome.

However, we decided to use this information, even considering that for each of the 22 human autosomes we were able to extract a relatively high number of 4d sites to calculate neutral models of evolution. The lowest number of 4d sites was for chromosome 21 (1,913 sites), followed by chromosome 22 (3,695). In total, we were able to extract 179,959 4d sites from the alignment. It is important to note that the separately estimated parameters for the 22 autosomes were then averaged using phyloBoot in order to create a combined nonconserved model to use in the phastCons analysis and thus minimize the effect of outliers. We have now better described the analytical workflow in the Material and Methods section.

Why did you use the space between CNEs to estimate intergenic distances? It seems like you could have had a higher density set of markers (you state 229 pairs) simply by looking at the distance between protein-coding genes?

Response: We agree with the reviewer and actually we originally thought about adding these additional data points to complement the analysis carried out using intergenic CNEs.

However, approximately one protein-coding gene per contig has been predicted in the contig-level assembly of one of the four species used in our analysis (Axolotl). This would have prevented us from calculating the intergenic expansion in the Axolotl genome, a critical species as its genome has expanded independently in comparison to the lungfish genomes, and thus allowed us to test the dynamic of such independent expansion events. As an alternative approach, we could have used the chromosome-level Hi-C scaffolded version of the Axolotl genome, but the unknown length of the “N” gaps used to bridge the contigs would have biased the calculation. These are the reasons why we used the contig-level assembly of the lungfish and Axolotl genomes and only the CNE dataset.

We are aware that the low number of inferred CNE pairs can only suggest a trend, and more data points would be ideal to make our results more robust and thereby strengthen our conclusions.

The analysis of potential positive selection within CNEs and protein-coding genes is potentially confused by misassignment of orthology. This could happen because of collapses within the assembly itself, or because of

misalignment. This is known to plague MultiZ alignments, which frequently pick the wrong sequence to align to the reference genome. What tests were performed to check that misalignments and misassemblies did not affect these results? I could not find any in the methods, but may have missed them. Without some further scrutinization of these alignments I am skeptical of all the positive selection results reported.

Response: For the selection of potential positively selected genes no genomic sequences were used. Homologous genes were found using Inparanoid based on peptide sequences, subsequently peptide sequences were blasted to transcript sequences for each species separately (tblastn, 100% identity). After this step sequences of homologous genes are combined, multiple sequences of proteins and the corresponding mRNA sequences are aligned into a codon alignment (Pal2Nal) and in the last step poorly aligned positions and divergent regions of mRNA are eliminated (Gblocks). Thus, we expect no problems here. To infer selection in non-coding regions we used whole-genome based alignments. To this end, we strictly followed the UCSC pipeline to first align independently each of the nine species to the reference genome (i.e., the hg38 assembly version of the human genome), and then merge each pairwise alignment in a 10-way multispecies alignment. We then calculated a neutral model of evolution (see answers above) and used phastCons to infer conserved elements (CNEs) across the genome. After removing CNEs located in coding regions, we identified those non-coding CNEs with potential signature of accelerated evolution in the lungfish lineage. We agree with the referee that misalignments and misassemblies are a common source of bias in this type of analysis. To minimize the effect of misassemblies (quality check on contig and Hi-C scaffold was discussed in other points above), we used the contig version for both lungfish and Axolotl genomes to carry out the alignments. To test the quality of the alignment, we have now added a new test. Briefly, we checked whether the orthology inferred from protein coding genes (see first part of this answer) matched the orthology in the multispecies alignment. We performed this analysis by intersecting the genomic coordinates of the protein coding genes (data from gene predictions) with those of the alignment blocks in the multispecies alignment for four species (human, chicken, Axolotl and lungfish). Of the genes retrieved in both datasets, 94.5% were found in both approaches to be orthologs in human and chicken; 93.5% in human and Axolotl, and 94.9% in human and lungfish. The high concordance between the two independent approaches provided us with an indirect measurement of the quality of our alignment. This test has now been included in the Material and Methods section.

Comments on the repeat / functional analysis:

Is there any evidence of the speculation that “such bursts of TE activity might also have been involved in the origin of novel gene functions” (line 225)?

Response: We have replaced the reference by a more recent review (Chuong et al. Nat. Rev. Genet. 18:71, 2017) and added a reference to the milestone paper that reports bursts of ERV as substrate for their exaptation for a new function in the mammalian placenta (Lynch et al. Cell Reports 10:551, 2015).

The standard repeat masking procedure of the 37 Gb genome assembly identified 57.3% of the genome as being repetitive (Fig.3a). This corresponds to 24.65 Gb of repetitive DNA, that together with the axolotl, is the highest repeat content in the animal kingdom. After masking the repeats using standard procedure (Supplementary Table 2), we tested whether the remaining unmasked DNA contained more repetitive sequences. This second round of repeat annotation on the hard- masked genome revealed an additional 23.92% of repetitive DNA (Fig.3a). Most of this additional repetitive DNA was classified as “Unknown” (and added 11% to the total unknown portion of repetitive DNA) as well as “LINE” (8.5% added, respectively) (Supplementary Table 3). This means that at least 2/3 of the lungfish genome is repetitive and expanded in two waves (Fig.3b).

This would suggest that 1/3 is non repetitive, or about 13Gb of sequence. If 13Gb is truly non-repetitive, what is its function (other than the 31k genes, and 17k ncRNA). Later in Fig. 3A, the plot labelled “Double TE Masking” claims there is 6.23% “DNA” or about 2.4Gb. It’s not totally clear how this is calculated and what is the meaning of “DNA”, please clarify. We believe it refers to MER1, MER2 and mariners?

Response: We are grateful for this comment, because it was also brought up by another reviewer as well. We completely agree with the surprising observation of at least 13Gb of ‘nonrepetitive’ sequence after doing the conventional analysis. We asked whether this remainder DNA had signatures of repetitiveness that was simply not detected due to the vast genome size. To have a closer look on the remaining fraction of the genome our paper provides the novel approach of double masking as a way to assess the repetitiveness in the remainder 13Gb. In fact, we found that indeed a significant proportion of this initially unmasked DNA has repeat signatures. In total, we estimated that the lungfish genome is up to 90% repetitive. In fact, we found that a substantial proportion of this DNA is repetitive, representing also many previously unrecoverable LINES and unknown elements. To be more conservative, we provided “at least 2/3” estimate in the text of the original submission. We rephrased the text to be more clear and have clarified the figure legend to explain that it refers to retroelements (LINE/LTR) and DNA transposons (DNA) as whole classes of repeat elements. Our approach highlights the necessity to be cautious about “classical” masking procedure of genomes and reveals that much (up to 90%) of the genomic “dark matter” in lungfish genome is repetitive, which we now explain better in our manuscript.

It is also worth explaining why 2 iterations of RepeatModeler + RepeatMasker were needed, or perhaps using a different method for identifying repeats if RepeatModeler does not perform well in this case. Is it not possible to change the parameters of the modeler so that 1 iteration is enough?

Response: Taking into account this and similar comments above, we have now rephrased and explained the need for double masking in uncovering repetitiveness in the unmasked version. Currently, no parameter can be used to adjust RepeatModeler to such a large genome. This indicates that more investigation into the methodology of repeat detection is

needed. However, it is consistent among species, e.g., genomes with smaller genome size (like *Drosophila*) or even human almost completely captures repeat complement in a single round of masking. Double masking is thus usually not needed for “conventional” genomes.

We would argue that the expansion of gene families relative to fish is one of the most interesting results in the paper, and should be worth its own figure (even if it is supplementary). These genes could also perhaps be compared in terms of Kimura substitution levels.

Response: We have checked those interesting expanded gene families for selection using site model in codeml (PAML) but did not find a signal of positive selection.

Is there evidence of transcriptional repression of transposable elements? Is there a visible history of evolution of TEs in response to their repressors?

Response: To address this point we have now looked for presence of TE sequences in the transcriptomes generated from poly A+ selected RNA of gonad, liver and brain. We find quite some expression of almost all types of TE's. Families with the higher copy number were also highly expressed in all three tissues tested. This, added to the presence of very similar copies for many TE families, strongly suggests that several types and families of transposable elements are still active in the lungfish genome and contribute to its ongoing expansion. This result is now included in the manuscript and presented in the new Supplementary Fig. 10.

We also studied the complement of genes from the transposon silencing machinery (Ago, piwi etc.). We found that neither in lungfish nor axolotl this pathway shows adaptation to the overabundance and activity of transposons (new Supplementary Table 7). Both giant genomes present the typical vertebrate repertoire with no sign of positive selection or copy number expansion. Moreover, there is no correlation between number of genes and genome size.

Are there any mutations or gene expansions that suggest that the lungfish has adapted to having a physically larger genome? A 43Gb genome must be difficult to replicate during S phase and align during metaphase.

Response: To address this question we screened our lists of positively selected genes and gene family dynamics for genes that are possibly involved in replication and cell division. We additionally analyzed the lungfish and axolotl gene sets for signatures of convergent evolution. We retrieved 3737 genes with about 11000 convergent substitutions. Of these 108 genes passed the test for being non-random hits. associated with GO terms related to replication, DNA-polymerase, nucleotide metabolism, DNA repair, cell cycle, cell division and cytokinesis. We checked 1456 genes from these three lists manually and with screening tools if they are associated with GO terms related to replication, DNA-polymerase, nucleotide metabolism, DNA repair, cell cycle, cell division and cytokinesis. We did not find a single significant hit in the gene family expansion (1348 genes), positive selection (159 genes) and convergent evolution (108 genes) gene lists.

We shared the same expectation as this reviewer that intuitively genes involved in replication and cell division should undergo mutations or expansions to cope with the challenges to function in physically giant genomes. One may reason, however, based on observing cell cycle characteristics in the axolotl, which has a 30 Gb genome, that considerable gene expansions or mutations that impact on gene function may not be necessary to execute the cell division and replication of large genomes. The axolotl embryonic cell cycle is 6 hours (compared to 30 minutes in the clawed frog *Xenopus*) while the somatic cell cycle is about 72 hours long. This probably means that in these organisms DNA replication can be

regulated by other mechanisms e.g. by increasing the number of origins of replication, which may be done by controlling gene expression level. These giant genome animals have proportionally larger cells and longer cell cycles as well, so the availability of components for DNA replication and mitosis can likely accumulate through controlling gene expression.

Minor comments

Typos: evolutionar → evolutionary (abstract line 60), GENECODE → GENCODE (throughout materials & methods), not a random → not random (line 289)

Response: typos have been corrected

Supplementary Figure 1: The “above” and “below” plots should be in the other order, to match legend

Response: the error has been corrected

Fig 2b: Label the little box in the middle - is that unexpanded LG8? Perhaps add dotted lines to the lungfish genome, like was done for the gar?

Response: We have revised Fig. 2b and labeled the little box and mention in the legend that this is the unexpanded LG8 for gar. We considered putting also dotted lines to the lungfish genome but finally refrained from doing so because it could have been misinterpreted as identity like it is the case for the middle box and the upper box.

What is the “standard repeat masking procedure”? (line 199, 202)

Response: The conventional procedure is a single masking round with RepeatModeler and RepeatMasker. We have modified this sentence to specify that.

Increase font sizes in Figure 3 (all panels, but especially axes on 3B)

Response: Figure 3 has been revised by increasing font size.

Supplementary Figure 8 has a text selection highlight left in on top panel, and no labels on axes **Response:** the text selection has been removed and labels for the axis are provided, now Supplementary Fig.12.

Error bars are so large in Supplementary Figure 9

Response: We apologize for a mistake in the legend of previous Supplementary Fig. 9. The vertical bars of the box plots show the range (minimum and maximum) of the percentage of TE's in those genes. This has been corrected, now Supplementary Fig.11.

Figure 4c - the box mentioned in the legend seems more red than “lilac

Response: we might redesign this figure and its coloring after this revision to enhance visibility for all readers.

Link to hic scaffolder github leads to 404 not found error

Response: We cannot explain this problem. For us the link works ok. If the error persists, please, try a different URL <https://bit.ly/30hNBxS>. Maybe the link in the document you received is broken.

Would be helpful to define “Ur-vertebrate” and “Ur-karyotype”?

Response: by those words we mean the LCA (Last common ancestor) of all vertebrates and its reconstructed karyotype ancestral state (see Simakov et al. Nature Ecology+Evolution 2020). The etymology of “Ur” of the German word means “original” or “primordial”.

Fig 3A/B annotations and axis labels are unreadable, this panel needs to be fully revised it is

just inscrutable.

Response: We have revised Fig 3A/B by using bigger letters.

Supplemental figure 8 has no y axis label and has remnants of a screenshot taken from inside a document editor

Response: Supplemental figure 8 has been revised, Y axis are labeled and the remnants of the screenshot removed.

Reviewer Reports on the First Revision:

Referee #1 (Remarks to the Author):

I appreciate the thorough response to reviewers and the revisions to the manuscript and supplementary materials. The manuscript is much easier to read and does a better job justifying the analyses.

There are a few points that still need addressing or correction.

There is a missed opportunity to analyze the raw reads for repeat elements, which are inevitably missed in the assembly. T-lex2 <https://pubmed.ncbi.nlm.nih.gov/25510498/> is an example of a program that can do this.

The k-mer analysis for genome size is described incorrectly. Based on the code provided in the supplement, it is implemented correctly but described incorrectly (section starting line 1120) and the corresponding figure (Supplementary Fig.1) has an incorrect x-axis label and figure legend. See <https://bioinformatics.uconn.edu/genome-size-estimation-tutorial/> for clarification. There is only one k-mer size that is used. It is recommended to explore multiple k-mer sizes for the analysis (this may also reveal a heterozygosity peak, although I agree that the coverage may be a limitation here). Also, the figure is truncated and a large bin (> largest point) will be informative and further demonstrate the repeat content of the genome.

The statement "This indicates that genome expansion similarly affects the genic and intergenic compartments of the genome" (line 292) needs additional context/development, what is the expectation (e.g. from Lynch expectations of genome size evolution), is this counter to an expectation?

Line 321, what were the results for metabolic rate vs non-metabolic rate?

Positive selection section is weak. Is there a need / reason to include it? Given the phylogenetic distance between organisms there is likely saturation of sites. Is 158 genes under positive selection more than expected given the number of tests? There is no correction for multiple tests. It seems the model comparing neutral to positive selection wasn't considered so those genes may just be neutrally evolving.

Please include a HiC scaffolding figure.

Minor points:

The paragraph starting line 164 seems out of place.

The "Ur-vertebrate" is never defined.

The use of the word "novel" is unnecessary in places. The dataset is novel, but aren't all new datasets novel?

Italicize species names.

Why does genome size affect repeat masking?

Line 219, are the reported percentages of the total or of the remaining genome?

Figure 2C, not all lungfish chromosomes are shown, it would be informative to include a similar supplementary figure with all chicken and all lungfish chromosomes. Different from Supplementary Fig.6.

What is "size" in supplementary Figure 6 and 8?

Supplementary Fig.17 needs improvement.

The text jumps from expansion dynamics to shh expression back to expansion dynamics. (around lines 349-370). Also, why focus on shh and not other important regulators of lung development?

Violin plots may be more informative for Supp Fig 11.

Remove lines from Supp Fig 12 and only include dots since the x-axis is not ordered sequentially.

Be sure to include github URL.

Referee #2 (Remarks to the Author):

The Authors have responded to all my concerns. The manuscript is much improved.

Referee #3 (Remarks to the Author):

Response to Lungfish Rebuttal

We originally reported in our review that this is a substantial and important body of work and continue to believe this. We find the manuscript to be improved, and the responses to the reviews to be mostly convincing, however, we still have some concerns and comments that we'd like to see addressed.

"The scaffold N50 was stated as 2.2 Gb, whereas it has been now finally determined to be 1.75 Gb"

We appreciate the update to the scaffold N50. The inclusion of the table with the scaffold sizes later on in the discussion about karyotyping helps with the understanding of this statistic.

Additionally, in your rebuttal you report a 1.7Mb N50, but in the paper it says "This yielded a contig assembly with a N50 of 1.86Mb". Please clarify which is correct.

"For the RNASeq mapping, we used Hisat2 with the default alignment score threshold of -30"

Why does the 91% in the rebuttal disagree with the original 84% in the text?

"Assembly collapse (i.e. repeat collapse) is not an issue using OLC (overlap layout consensus) assemblers, and MARVEL belongs to this family of assemblers"

We STRONGLY, STRONGLY disagree with this statement. This is nonsense. Repeat collapse is still an issue in OLC assemblers, particularly in regions such as segmental duplications, which may be too large for reads to span entirely across. With OLC, in highly repetitive regions, if your reads are not guaranteed to extend into unique flanking regions false overlaps can bridge widely separated regions

or regions that simply neighbor each other in a repeat array. In the latter case repeat collapse is a consequence.

SDA (Segmental Duplication Assembler) from the Eichler lab could be used to identify regions of collapse in assemblies, and we would strongly recommend this analysis, as copy number compression is likely to be the dominant source of error in the assembly.

Your transcript analysis is important, but does not address misassemblies that could occur in intergenic regions. There is also no prior paper that benchmarks MARVEL in terms of misassemblies, which makes it difficult to be confident in these results. Misassemblies are not the same as "misjoins" which you refer to later in your response.

"We developed an experimental detection tool that uses the HiC in order to detect contact arrangements that are indicative of mis-joins. When running this on the final contigs we detect 1583 potential mis-joins that would require extensive further manual validation to be eliminated"

How does this experimental detection tool work, in brief?

"as described below, Bionano is not an appropriate means for such validation." "We describe below how we determined the HiC coverage to use for scaffolding and address the Bionano validation."

Thank you for the explanation of the HiC coverage. This makes some sense and agrees with our experience. It might still be good to include the bionano as a confirmation on assembly agreement, even if it is unhelpful in guiding the assembly process. Specifically, using Hi-C as your scaffolding criteria and your validation criteria has the risk of circularity. Errors that are inherent to the linkage data may not be detected. Additionally, the vast size difference between contigs (N50~ =2Mb) and scaffolds (N50~ =1750Mb) leaves a lot of room for error in the scaffolding step, which is also unvalidated and previously unpublished for this method.

"Important to the scaffolding procedure is a misjoin detection tool in MARVEL that allows for visual inspection of the scaffolds by means of the contact map."

→ does this mean that the inspection is done manually after the fact? Or is the process automated? (48,505 joins is a lot!)

"260,478 out of 283,820 contigs align to the genome contigs using GMAP, while 253,781 align to the scaffolds."

20-30,000 transcriptome contigs not aligning back to the contigs/scaffolds seems like a lot. However, you mention the "well-known problems" of assembled transcriptomes. Can you please clarify the nature of the missing mapping fraction?

Supp Fig 25c,d is interesting -- " These plots show that the scaffolding shifted the coverage of many transcripts to 100%, validating the accuracy of the scaffolding." Encouraging results!

" N(X) plots for the contig assembly and the scaffolded genome have been added, see new Supplementary Figs. 20 and 22."

Thanks for including these. I see that they are actually Supp Figs 22 and 24, though. I think there may be some other inconsistencies in the figure numbering, worth double-checking. For example, there is another instance where the HiC workflow is cited as Supp Fig 21, and it ends up being 23.

" the heterozygosity is estimated to be only 0.0965-0.1278 %... We do not detect a secondary peak in the kmer data to estimate heterozygosity"

We appreciate the explanation. 0.1% is, if we understand correctly, around 1/1000 bases, i.e. similar to humans. The het peak in humans is easily detectable in a 30x illumina sample, so we are very surprised this is not the case here. Some comments on this in the paper and analysis would be appreciated.

"We computed a Q-value of 26 for the lungfish assembly"

Thanks. This is understandable, but not great, clearly.

"This is a very technical request that would require its own document. We are happy to provide a document as a separate publication, or better yet, discuss these details at an appropriate point."

We think a brief explanation would still be helpful.

"More recently, a colleague benchmarked FLYE, CANU, SHASTA, WTDBG2 and MARVEL using a high coverage Nanopore dataset of the drosophila genome aiming to achieve very accurate and complete assembly particularly of the highly repetitive piRNA cluster regions. CANU, which can treat repeats comparably to MARVEL did not finish within the given time frame. FLYE and WTDBG2 assemblies showed defects in chromosome 3 and 2 respectively. The assembly from SHASTA generally looked very good except the piRNA regions and was outcompeted for accuracy/contiguity in the piRNA cluster region by MARVEL. In conclusion MARVEL has so far provided the best performance in difficult-to-assemble, repeat-rich genomes. Specific metrics and figures are available upon request."

This comparison is useful. Were the piRNA regions the only regions where MARVEL outcompeted Shasta? Was analysis of misassemblies done for the whole genome? (i.e. using a tool like QUAST, which aligns the assembly to the reference and calculates regions where there are misalignments) We do think the inclusion of the scaffolder within MARVEL, in addition to these findings, also helps to justify the choice of MARVEL as an assembler.

"We completely agree with the surprising observation of at least 13Gb of 'nonrepetitive' sequence after doing the conventional analysis... In fact, we found that indeed a significant proportion of this initially unmasked DNA has repeat signatures. In total, we estimated that the lungfish genome is up to 90% repetitive."

This makes more sense; this additional analysis with double repeat masking is useful. We will be very surprised if the *vast*, *vast* majority of this sequence is not repetitive in origin.

" Further analyses of these candidate genes led us to sall1, an important gene for the distal branching of the limb and involved in human limb pathologies (Townes_Brocks syndrome)."

This is a great new insight. The inclusion of additional novel biological insights improves the quality of the manuscript.

"To have a closer look on the remaining fraction of the genome our paper provides the novel approach of double masking as a way to assess the repetitiveness in the remainder 13Gb. In fact, we found that indeed a significant proportion of this initially unmasked DNA has repeat signatures. In total, we estimated that the lungfish genome is up to 90% repetitive. In fact, we found that a substantial proportion of this DNA is repetitive, representing also many previously unrecoverable LINEs and unknown elements ... To be more conservative, we provided "at least 2/3" estimate in the text of the original submission. We rephrased the text to be more clear and have clarified the figure legend to explain that it refers to retroelements (LINE/LTR) and DNA transposons (DNA) as whole classes of repeat elements"

In the discussion/results I would first report whatever you think is the most accurate value for the proportion of repeats, and elaborate on the 2-step masking afterwards. The fact that the first run gives you 67% repeat content seems more like an artefact or inadequacy of the repeat thresholding/modeling than it is a feature of the genome that you are publishing on.

"These giant genome animals have proportionally larger cells and longer cell cycles as well, so the availability of components for DNA replication and 18 mitosis can likely accumulate through controlling gene expression."

This is very interesting, and perhaps merits its own investigation, but for this publication it may at least be worth citing a study regarding cell size/cycles.

Author Rebuttals to First Revision:

Point-by-point response

Referee #1 (Remarks to the Author):

I appreciate the thorough response to reviewers and the revisions to the manuscript and supplementary materials. The manuscript is much easier to read and does a better job justifying the analyses. There are a few points that still need addressing or correction.

There is a missed opportunity to analyze the raw reads for repeat elements, which are inevitably missed in the assembly. T-lex2 <https://pubmed.ncbi.nlm.nih.gov/25510498/> is an example of a program that can do this.

Response: We thank the reviewer for this suggestion. Indeed, a view from raw read data is important to verify the completeness of the repetitive landscape derived from the assembly. The omission of such an implementation was based on the lack of proper software that could analyze such a large quantity of raw data. However, following the reviewer's suggestion we have made an attempt by running DNAPipeTE (<https://academic.oup.com/gbe/article/7/4/1192/533768>). To the best of our knowledge this tool is more commonly used to generally quantify repeats in raw Illumina reads, whereas T-lex2 would be more appropriate for TE insertion/deletion polymorphism detection in population data.

We selected a subset of our raw Illumina read data, comprising a fastq file of 47 Gbytes (150bp reads, totaling 21 Gb, or about 0.5x coverage). DNAPipeTE pipeline was run with that coverage on a high memory (1Tb RAM) machine, resulting in an unfiltered, intermediary Trinity assembly of 816706 contigs. The pipeline was not able to complete the quantification of abundances due to large data quantities (several weeks runtime in sorting). To overcome this problem, we ran the abundance quantification tool kallisto, by using the assembly of repeats as reference and the sample genomic data as reads. This allowed us to identify 1000 most 'abundant'/repetitive contigs, 996 of those had 95%+ similarity hits to our repeat library. The remaining 4 were simple repeats that were captured by RepeatMasker. Similarly, taking all DNAPipeTE/Trinity contigs that are robustly repetitive, i.e., with 100 or more reads (25520 total), resulted in 25444 (99.7%) matches of 95% identity or higher. These results provide confidence that none of the major repeat element families were missed during our genome assembly, repeat detection, annotation, and masking.

The k-mer analysis for genome size is described incorrectly. Based on the code provided in the supplement, it is implemented correctly but described incorrectly (section starting line 1120) and the corresponding figure (Supplementary Fig.1) has an incorrect x-axis label and figure legend.

See <https://bioinformatics.uconn.edu/genome-size-estimation-tutorial/> for clarification. There is only one k-mer size that is used. It is recommended to explore multiple k-mer sizes for the analysis (this may also reveal a heterozygosity peak, although I agree that the coverage may be a limitation here). Also, the figure is truncated and a large bin (> largest point) will be informative and further demonstrate the repeat content of the genome.

Response: Thank you for these suggestions. We have corrected the description of the new Suppl. Fig. 1 to: *"The size of the lungfish genome was estimated using k-mer counts generated from the*

genome correction data using jellyfish⁷⁴. First, k-mers were counted separately for each fastq file using jellyfish count. The counts from individual files were then merged using jellyfish merge and the histogram was generated using jellyfish histo.”

We have also re-labeled the axes in the new Supplementary Figure 1 and modified the legend. In the new Supplementary Figure 1 (see below), we have performed the k-mer analyses with 7 different k-mer sizes ranging from 19-31, and we still do not see a heterozygosity peak. We agree with the reviewer that we are probably limited by coverage here.

We also provide the k-mer graph up to a k-mer abundance including a large bin that is >largest point as requested.

New Supplementary Figure 1. k-mer frequency analysis. Illumina dataset was used to generate the spectra of k-mer abundances using 7 different k-mer sizes.

The statement “This indicates that genome expansion similarly affects the genic and intergenic compartments of the genome” (line 292) needs additional context/development, what is the expectation (e.g. from Lynch expectations of genome size evolution), is this counter to an expectation?

Response: Thank you for this very interesting suggestion. In the revised manuscript (see line 312) we have indicated that the genome size expansion in lungfish is in line with Lynch’s expectations (Lynch and Conery 2003 Science).

Line 321, what were the results for metabolic rate vs non-metabolic rate?

Response: Response: We could do this analysis for developmental vs non-developmental genes because we had from our consortium an evidence-based, manually curated list of developmental genes. Such a resource is not available to us for metabolic rate genes. Thus, at present we are unfortunately not able to perform an analysis of intron sizes in metabolic rate determining genes vs the rest of the transcriptome. We are taking up this suggestion of the reviewer for our follow-up studies on the Australian lungfish genome combined with the other lungfish genomes that we are currently sequencing.

Positive selection section is weak. Is there a need / reason to include it? Given the phylogenetic distance between organisms there is likely saturation of sites. Is 158 genes

under positive selection more than expected given the number of tests? There is no correction for multiple tests. It seems the model comparing neutral to positive selection wasn't considered so those genes may just be neutrally evolving.

Response: The positive selection paragraph has been removed now from the main text to Supplementary information.

Please include a HiC scaffolding figure.

Response: We have included a new HiC scaffolding figure, please, see below (which is now the new Supplementary Fig. 25), see here:

Supplementary Fig. 25. HiC contact heat map of the scaffolded portion of the Lungfish genome assembly, ordered by scaffold length. Blue boxes indicate the scaffold boundaries. The four largest scaffolds represent both chromosome arms on a single scaffold. Remaining scaffolds are split into chromosome arms or represent microchromosomes.

Minor points:

The paragraph starting line 164 seems out of place.

Response: Thank you. We agree that this paragraph was somehow out of place and moved it to the section “Hallmarks of the giant lungfish genome” where we analyze the characteristics of those genomic features that are related to genome expansion. This provides a much better context for this paragraph than at the previous place.

The “Ur-vertebrate” is never defined.

Response: We have deleted this term and substituted by “ancestral Vertebrate”

The use of the word “novel” is unnecessary in places. The dataset is novel, but aren't all new

datasets novel?

Response: We have deleted the term “novel” in several places or replaced by “new”, where it appeared reasonable to point out that a finding, rather than the dataset itself, had not been made before.

Italicize species names.

Response: We have checked all manuscript files carefully and do hope that no species name that is not italicized has escaped our attention.

Why does genome size affect repeat masking?

Response: Currently available repeat masking software has been designed or tested to handle genome sizes within few Gb. How repeat masking performs on extremely large genomes so far has not been tested, due to the lack of well assembled genomes. In this paper, we asked whether the unmasked DNA using classical repeat masking approach contained any additional signatures of repetitive elements and indeed we found evidence for such. We also find that the relative gain in the second repeat masking round is proportional to the genome size, i.e. small genomes such as *Drosophila* do not gain any repetitive elements in the second masking. These results will certainly help guide development of future repeat masking approaches.

Line 219, are the reported percentages of the total or of the remaining genome?

Response: The percentages are of the total genome. This information is now included.

Figure 2C, not all lungfish chromosomes are shown, it would be informative to include a similar supplementary figure with all chicken and all lungfish chromosomes. Different from Supplementary Fig.6.

Response: We now provide in the new Supplementary Fig.6 a macrosynteny plot with all chicken chromosomes and kept the original Figure 2C since here we focus on micro-chromosomes.

What is “size” in supplementary Figure 6 and 8?

Response: We have clarified in the legends of previous Supplementary Figs. 6 and 8 (now Supplementary Figs. 5 and 7) that the “size” value refers to the number of shared orthologous gene families between homologous chromosomes.

Supplementary Fig.17 needs improvement.

Response: We have improved Supplementary Fig. 17 by spelling out the names of the species and have extended the figure legend to explain what the numbers (also now color-coded) above the branches are reporting.

The text jumps from expansion dynamics to *shh* expression back to expansion dynamics. (around lines 349-370). Also, why focus on *shh* and not other important regulators of lung development? **Response:** We are limited by the availability of Australian lungfish embryos and therefore had to be very selective how we use them for this study.

Violin plots may be more informative for Supp Fig 11.

Response: We have revised Supplementary Fig .11 and present the results now as violin plots.

Remove lines from Supp Fig 12 and only include dots since the x-axis is not ordered sequentially

Response: We have revised Supplementary Fig. 12 and present the results new as scatter plot.

Be sure to include github URL.

Response: Has been done.

Referee #2: we are glad that this referee is satisfied with our changes and revisions.

Referee #3 (Remarks to the Author):

Response to Lungfish Rebuttal

We originally reported in our review that this is a substantial and important body of work and continue to believe this. We find the manuscript to be improved, and the responses to the reviews to be mostly convincing, however, we still have some concerns and comments that we'd like to see addressed.

“The scaffold N50 was stated as 2.2 Gb, whereas it has been now finally determined to be 1.75 Gb” We appreciate the update to the scaffold N50. The inclusion of the table with the scaffold sizes later on in the discussion about karyotyping helps with the understanding of this statistic. Additionally, in your rebuttal you report a 1.7Mb N50, but in the paper it says “This yielded a contig assembly with a N50 of 1.86Mb”. Please clarify which is correct.

Response: The 1.7Mb (1.797.324) comes from the raw contigs, whereas the 1.86Mb (1.864.494) comes from the polished, RNA-seq corrected version of the contigs. This means that the predominant error type in the raw assembly were deletions.

“For the RNASeq mapping, we used Hisat2 with the default alignment score threshold of -30” Why does the 91% in the rebuttal disagree with the original 84% in the text?

Response: There is a misunderstanding and our text in the rebuttal was perhaps not clear enough. The overall mapping rate of the RNASeq data was 84%. Of that 84%, 91% of the reads mapped with less than 3 mismatches per 150bp and the remaining 9% of the 84% mapped with up to 6 mismatches per 150bp.

Here is the text we had provided in the manuscript in case reviewer#3 had missed that: *“The overall average alignment rate was 84%, of which 91% mapped with up to 3 mismatches per 150 bp.”*

“Assembly collapse (i.e. repeat collapse) is not an issue using OLC (overlap layout consensus) assemblers, and MARVEL belongs to this family of assemblers“

We STRONGLY, STRONGLY disagree with this statement. This is nonsense. Repeat collapse is still an issue in OLC assemblers, particularly in regions such as segmental duplications, which may be too large for reads to span entirely across. With OLC, in highly repetitive regions, if your reads are not guaranteed to extend into unique flanking regions false overlaps can bridge widely separated regions or regions that simply neighbor each other in a repeat array. In the latter case repeat collapse is a consequence.

SDA (Segmental Duplication Assembler) from the Eichler lab could be used to identify regions of collapse in assemblies, and we would strongly recommend this analysis, as copy number compression is likely to be the dominant source of error in the assembly. Your transcript analysis is important, but does not address misassemblies that could occur in intergenic regions. There is also no prior paper that benchmarks MARVEL in terms of misassemblies, which makes it difficult to be confident in these results. Misassemblies are not the same as “misjoins” which you refer to later in your response.

Response: We acknowledge our initial statement implied that ALL OLC assemblers are immune to repeat collapse, which is clearly an overstatement. Thank you for pointing out

that we were not clear enough in what we tried to convey. What we really meant to say, is that the MARVEL assembler is run under conditions where the assembly process stops when it cannot cross a repeat region, as is described next. Repeat collapse (or expansion) are due to incorrectly gauging the number of repeat instances during the assembly process and thereby resulting in contigs containing fewer (or more) repeat copies as actually present in the genome. Here the term repeat can either mean a “classical” repeat element or a cluster of highly similar sequences (e.g. a gene cluster) that are not differentiable under the error rate of the reads.

During our assembly process the reads were annotated with repeat information based on coverage anomalies followed by transitive transfer of this annotation (based on alignments), a process we refer to as repeat homogenization. This, in our experience, does result in an over-annotation of the reads. This is due to similarities in the error profile of divergent copies inducing matches to specific repeat instances with a highly similar error profile. In addition, in order not to fold similar stretches of sequence, we start annotating sequence as repetitive if the alignment depth goes above 1.7 the expected depth.

Furthermore, in our MARVEL assembly process only read overlaps that can be anchored in sufficient unique sequence are used during assembly graph construction resulting in contigs terminating in repeats if there is no read spanning it. This is why we asserted that (at least for MARVEL) repeat collapse is unlikely to be an issue. MARVEL does support heuristic repeat resolution by attempting repeat-decomposition, which can result in misjoins and repeat collapse, but this has not been used when assembling the lungfish genome. We ensured a very conservative assembly, due to the danger of assembly errors when working with such a highly repetitive genome.

“We developed an experimental detection tool that uses the HiC in order to detect contact arrangements that are indicative of mis-joins. When running this on the final contigs we detect 1583 potential mis-joins that would require extensive further manual validation to be eliminated”

How does this experimental detection tool work, in brief?

Response: The following (also see figure below) describes the procedure we used to quickly look whether the contigs have any internal misjoins. Contacts are binned (bin size based on contig length and expected digest site distance), projected onto the diagonal (i.e. 1D space) and decomposed if possible into separate components (i.e. stretches of non-zero bins along the diagonal). Components are then iteratively merged based on contacts linking them. If more than one component remains, a potential misjoin is reported. The rationale here is that misjoins will result in the contig separating into components, which then can't be merged due to insufficient contacts indicating that the genomic regions represented by the components are not close in sequence space. Whereas, for example, components due to repeat-induced contact depletion can be merged due to the presence of off-diagonal contacts. Below is a schema of the contig misjoin detection procedure.

If this reviewer feels that it would help to understand how the assembly was done we could also add this diagram as supplementary figure to the manuscript (but, as of now, we have only included here in the point-by-point response).

Figure R1. Schema illustrating the contig misjoin detection process. Hi-C contacts are binned along the diagonal (a, b). Points that are not crossed by a sufficient number of contacts are deemed potential misjoins and are thus separated (c, dotted line).

“as described below, Bionano is not an appropriate means for such validation.” “We describe below how we determined the HiC coverage to use for scaffolding and address the Bionano validation.”

Thank you for the explanation of the HiC coverage. This makes some sense and agrees with our experience. It might still be good to include the bionano as a confirmation on assembly agreement, even if it is unhelpful in guiding the assembly process. Specifically, using Hi-C as your scaffolding criteria and your validation criteria has the risk of circularity. Errors that are inherent to the linkage data may not be detected. Additionally, the vast size difference between contigs (N50~2Mb) and scaffolds (N50~1750Mb) leaves a lot of room for error in the scaffolding step, which is also unvalidated and previously unpublished for this method.

Response: Thank you for the suggestion to potentially include Bionano as a confirmation step. We are unfortunately lacking material to perform this step as we used up all of the tissues that can be processed for Bionano for our Nanopore and Illumina sequencing and HiC library preparation.

To further validate the HiC assembly process for huge genomes we have more deeply analyzed our assemblies from axolotl where both methods could be employed and compared the axolotl Bionano scaffolds with the axolotl HiC assembly which was performed on the contigs.

The axolotl Bionano scaffolds and the HiC scaffolding concur in 99.6% of the 10,775 total Bionano scaffolds that contained more than one contig. We should also mention that the lungfish genome was significantly easier to scaffold due to the 10x times greater contig N50 so we would expect an even greater concurrence if we had lungfish Bionano scaffolds.

For axolotl we examined the remaining 0.4% of Bionano scaffolds. We deemed these to be

Bionano misjoins because the inclusion of such scaffolds in the HiC assembly process induced a megabase or larger scale signal off the diagonal in the HiC maps. The HiC assembly process relies on the abundant subset of contacts close to the diagonal where we expect an exponential decrease in contact frequency with distance. If a contig from another region or indeed another chromosome is scaffolded into a region, then aberrant long-distance contacts start to be found off diagonal because that misplaced contig naturally has a high frequency of contacts to its natural neighbors that are in a distant part of the genome. This is what we found with a small number of the axolotl Bionano scaffolds, and we do not see another explanation there. An example of such misassembly and manual correction is given in Supplementary Figure 1 of Jebb D, et al. Nature. 2020. 583:578-584.

In our view the use of long-distance HiC contact information to confirm the scaffolding is not really circular because we are not using the assembled sequences to confirm assembly accuracy. Rather we are using the HiC contact map information at the large length scale to assess the accuracy of the assembly that was made based on short scale contact information.

“Important to the scaffolding procedure is a misjoin detection tool in MARVEL that allows for visual inspection of the scaffolds by means of the contact map.”

→ does this mean that the inspection is done manually after the fact? Or is the process automated? (48,505 joins is a lot!)

Response: We are sorry if a misunderstanding occurred here and want to clarify what we mean by misjoins and gaps. In the first rebuttal, we mentioned that “There are 48505 gaps at the contig joins in the scaffolds.” These are not misjoins but simply the gaps between the contigs in the assembly, and therefore do not need correction in the HiC process. Our statement “Important to the scaffolding procedure is a misjoin detection tool in MARVEL that allows for visual inspection of the scaffolds by means of the contact map. This allows the exclusion of larger structural defects” was meant to explain that this is a process where the MARVEL assembler outputs a contact map after a scaffolding step. Indeed, by manual inspection, it is very easy to spot problems in the scaffolding. Examples of such anomalies and manual correction are shown in Supplementary Figure 1 of Jebb D, et al. Nature. 2020. 583:578-584 and are very easy to handle manually.

“260,478 out of 283,820 contigs align to the genome contigs using GMAP, while 253,781 align to the scaffolds.”

20-30,000 transcriptome contigs not aligning back to the contigs/scaffolds seems like a lot. However, you mention the “well-known problems” of assembled transcriptomes. Can you please clarify the nature of the missing mapping fraction?

Response: We checked the size distribution of these 23,342 transcripts that did not map to the genome and over 18,000 of them are 300 bp or shorter. For establishing the axolotl genome we had, honestly, excluded transcripts less than 300 base-pairs as is often done and this had been overlooked here.

This leaves only 5,000 unmapped transcripts greater than 300 bp in size (~1.8%). Among these we find hits in the bacterial database and so these transcripts are very likely contamination from bacteria that could not be removed from the tissue before RNA preparation.

We have now added this information into the Materials and Methods section.

Supp Fig 25c,d is interesting -- “These plots show that the scaffolding shifted the coverage of many transcripts to 100%, validating the accuracy of the scaffolding.” Encouraging results!

Response: Thank you! Done

“N(X) plots for the contig assembly and the scaffolded genome have been added, see new Supplementary Figs. 20 and 22.”

Thanks for including these. I see that they are actually Supp Figs 22 and 24, though. I think there may be some other inconsistencies in the figure numbering, worth double-checking. For example, there is another instance where the HiC workflow is cited as Supp Fig 21, and it ends up being 23.

Response: We apologize for these mistakes in the point-by-point response letter. This happened during the process of stepwise revisions of the manuscript with information from several co-authors and adjusting several times the numbering of the Supplementary Figures.

“the heterozygosity is estimated to be only 0.0965-0.1278 %... We do not detect a secondary peak in the kmer data to estimate heterozygosity”

We appreciate the explanation. 0.1% is, if we understand correctly, around 1/1000 bases, i.e. similar to humans. The het peak in humans is easily detectable in a 30x illumina sample, so we are very surprised this is not the case here. Some comments on this in the paper and analysis would be appreciated.

Response: We have further explored the k-mer analysis by running the analysis with 7 different k-mer sizes as suggested by reviewer 1 (see above). This graph is provided in the response to reviewer 1 and now also as a new Supplementary figure 1. Nonetheless we do not see a heterozygosity peak. As reviewer 1 mentions, this is likely due to not enough coverage. We refer the reviewer to the Supplement from the original Genomescope publication (Vurture et al. 2017. *Bioinformatics*, 33:2202), where the 0.1% heterozygosity peak is barely visible on the shoulder using an idealized *Drosophila* dataset, at 100x coverage. Considering not only coverage, but also the enormous size and high repeat content of the lungfish genome, it is likely that the k-mer abundance spectrum does not allow the heterozygosity peak to be visible. The supplement equation 2 describes the modelling of these distributions under simple repeat scenarios, which can help the reviewer gauge how the massive presence of repeats may affect the k-mer count distribution.

“We computed a Q-value of 26 for the lungfish assembly” Thanks. This is understandable, but not great, clearly.

Response: Done

“This is a very technical request that would require its own document. We are happy to provide a document as a separate publication, or better yet, discuss these details at an appropriate point.” We think a brief explanation would still be helpful.

Response: Thanks for your interest in this procedure which we now describe in more detail in the

„Genome Assembly“ section of the Materials and Methods section.

“Broken alignments, due to two reads having a low-quality region where the alignment passes through, are detected and repaired (LAsitch). The approximate error of the reads is estimated by inspecting the level of identity achieved in the alignments to it (LAq). This error rate is then used to trim the reads (LAq). The repeats are detected based on coverage statistics (LArepeat).

Transitive transfer is used to homogenize the repeat annotation across the read-mass and ensure that repeats smaller than the minimum alignment length at the beginning and end of a read are annotated as well (TKhomogenize). This repeat annotation is then combined with the repeat masks used when aligning the reads initially (TKcombine). Larger regions in the reads not spanned by any read are detected and resolved, these regions are referred to as gaps (LAgap).

Resolving in this context entails selecting a group of overlaps from one of the sides of the gap and discarding the other. This then necessitates another round of trimming of the reads (LAq). The remaining alignments are then filtered to remove local and repeat-induced alignments (LAfilter). After this, only true overlaps should remain, which are then used to build the assembly graph (OGbuild). Optionally, at this stage the reads used for constructing the contigs, as selected during assembly graph touring (OGtour), can be corrected (LAcorrect) and used for constructing the contigs (tour2fasta). The sequence of command corresponding to this workflow is given below.

commands prefixed with [B] are executed for each read block

PLACEHOLDERS:

- {READ_BLOCK} - a single read block, e.g. LFISH.1, LFISH.321

- {ALL_READ_BLOCKS} - all read blocks, e.g. LFISH.1, LFISH.2, ... LFISH.n

- {LFISH_DB}, {LFISH_FIX_DB}, {LFISH_FIX_CORR_DB} - path to the database, e.g. LFISH.db

- {READ_BLOCK_NUMBER} - always used jointly with a {READ_BLOCK},

used to indicate the number after the database name,

e.g. {READ_BLOCK} = LFISH.4 then {READ_BLOCK_NUMBER} = 4

compute all alignments for a random subset of database blocks

derive a coarse repeat annotation to mask the worst repeat elements

transfer the annotation to all other reads based on the alignments

[B] daligner -A -t20 -T4 {READ_BLOCK} {ALL_READ_BLOCKS}

[B] LAMerge -n 16 {LFISH_DB} {READ_BLOCK}.las {READ_BLOCK_DIRECTORY}

[B] LArepeat -h 4.0 -l 3.5 -c 30 -t repeats_init -b {READ_BLOCK_NUMBER} {LFISH_DB} {READ_BLOCK}.las

[B] TKhomogenize -I {READ_BLOCK_NUMBER}.repeats_init -I hrepeats -b {READ_BLOCK_NUMBER}

{LFISH_DB} {READ_BLOCK}.las

after a repeat annotation for a random subset has been derived and

transferred to every other reads merge the resulting annotations

{READ_BLOCK_x_NUMBER} refers to the {READ_BLOCK_NUMBER} of each block used

in the TKhomogenize calls above

TKcombine {LFISH_DB} hrepeats_init {READ_BLOCK_x_NUMBER}.hrepeats_init ...

with the repeat annotation at hand we then perform the alignments

for all read block comparisons

[B] daligner -mhrepeats_init -t20 -T4 {READ_BLOCK_x} {ALL_READ_BLOCKS}

[B] LAMerge -n 16 {LFISH_DB} {READ_BLOCK}.las {READ_BLOCK}

derive read-quality quality information from the alignments

[B] LAq -b {READ_BLOCK_NUMBER} {LFISH_DB} {READ_BLOCK}.las

merge the block specific quality

information TKmerge -d {LFISH_DB} q

repair sequencing artifacts and low quality regions in the reads

[B] LAFix -g -l {LFISH_DB} {READ_BLOCK}.las {READ_BLOCK}.fasta

```
# create new read database
FA2db -x 4000 -c source
{LFISH_FIX_DB} *.fasta DBsplit -s 400
{LFISH_FIX_DB}

# repeat what has been done above for the initial repeat masking

[B] daligner -A -t20 -T4 {READ_BLOCK} {ALL_READ_BLOCKS}
[B] LAmerge -n 16 {LFISH_FIX_DB} {READ_BLOCK}.las {READ_BLOCK}
[B] LArepeat -h 4.0 -l 3.5 -c 30 -t repeats_init -b {READ_BLOCK_NUMBER}
{LFISH_FIX_DB}
{READ_BLOCK}.las
[B] TKhomogenize -I {READ_BLOCK_NUMBER}.repeats_init -I hrepeats -b
{READ_BLOCK_NUMBER}
{LFISH_FIX_DB} {READ_BLOCK}.las

# after a repeat annotation for a random subset has been derived and
# transferred to every other reads merge the resulting annotations
# {READ_BLOCK_x_NUMBER} refers to the {READ_BLOCK_NUMBER} of each block
used
# in the TKhomogenize calls above

TKcombine {LFISH_FIX_DB} hrepeats_init {READ_BLOCK_x_NUMBER}.hrepeats_init
...

# with the repeat annotation at hand we then perform the alignments
# for all read block comparisons

[B] daligner -mhrepeats_init -t20 -T4 {READ_BLOCK_x} {ALL_READ_BLOCKS}
[B] LAmerge -n 16 {LFISH_FIX_DB} {READ_BLOCK}.las
{READ_BLOCK_DIRECTORY}

# repair broken alignments
[B] LAstitch {LFISH_FIX_DB} {READ_BLOCK}.las {READ_BLOCK}.stitch.las

# compute read qualities and trimming information
[B] LAq -b {READ_BLOCK_NUMBER} {LFISH_FIX_DB} {READ_BLOCK}.stitch.las

# merge quality and trim
information TKmerge -d
{LFISH_FIX_DB} trim
TKmerge -d {LFISH_FIX_DB}
q

# compute repeat annotation
[B] LArepeat -c 30 -t repeats -b {READ_BLOCK_NUMBER} {LFISH_FIX_DB}
{READ_BLOCK}.stitch.las

# merge repeat annotation
TKmerge -d {LFISH_FIX_DB}
repeats
```

```
# transfer the annotation
[B] TKhomogenize -I repeats -b {READ_BLOCK_NUMBER} {LFISH_FIX_DB}
{READ_BLOCK}

# combine it with the initial repeat annotation
TKcombine {LFISH_FIX_DB} hrepeats #.hrepeats
TKcombine {LFISH_FIX_DB} repeats_all
hrepeats repeats

# resolve haplotypic breaks and breaks due to leftover weak regions in the reads
[B] LAgap -m 1 -L -t trim {LFISH_FIX_DB} {READ_BLOCK}.stitch.las
{READ_BLOCK}.gap.las

# update trimming information
[B] LAq -b {READ_BLOCK_NUMBER} -u -t trim -T trim2 {LFISH_FIX_DB}
{READ_BLOCK}.gap.las

# merge trim information
TKmerge -d
{LFISH_FIX_DB} trim2

# filter alignments
[B] LAfilter -p -n 300 -t trim2 -u 0 -T -r repeats_all {LFISH_FIX_DB}
{READ_BLOCK}.gap.las
{READ_BLOCK}.filter.las

# merge filtered alignments
LAMerge -n 16 {LFISH_FIX_DB} merged.filter.las *.filter.las
```

“More recently, a colleague benchmarked FLYE, CANU, SHASTA, WTDBG2 and MARVEL using a high coverage Nanopore dataset of the drosophila genome aiming to achieve very accurate and complete assembly particularly of the highly repetitive piRNA cluster regions. CANU, which can treat repeats comparably to MARVEL did not finish within the given time frame. FLYE and WTDBG2 assemblies showed defects in chromosome 3 and 2 respectively. The assembly from SHASTA generally looked very good except the piRNA regions and was outcompeted for accuracy/contiguity in the piRNA cluster region by MARVEL. In conclusion MARVEL has so far provided the best performance in difficult-to-assemble, repeat-rich genomes. Specific metrics and figures are available upon request.” This comparison is useful. Were the piRNA regions the only regions where MARVEL outcompeted Shasta? Was analysis of misassemblies done for the whole genome? (i.e. using a tool like QUASt, which aligns the assembly to the reference and calculates regions where there are misalignments) We do think the inclusion of the scaffold within MARVEL, in addition to these findings, also helps to justify the choice of MARVEL as an assembler.

Response: Thank you for the interest in the comparison of assemblers. Another aspect where MARVEL outperformed SHASTA was the ability assemble through heterozygous transposon insertions. Our colleague is working with a cell line in which there was a burst of about 200 heterozygous transposon insertions. When he examines the assemblies at those insertions, MARVEL reports two copies—one with and one without the transposon whereas SHASTA randomly assigns whether the transposon is present or not. With MARVEL it is possible to assemble through the transposon, whereas SHASTA breaks the assembly at the transposon boundaries.

Our colleague did run the QUASt program but it just reported several thousand “misassemblies” which actually represent places of transposon insertion that are different than the reference genome (which by the way is from a strain that is very different from the common lab strain and therefore has thousands of different transposon insertions).

“We completely agree with the surprising observation of at least 13Gb of ‘nonrepetitive’ sequence after doing the conventional analysis... In fact, we found that indeed a significant proportion of this initially unmasked DNA has repeat signatures. In total, we estimated that the lungfish genome is up to 90% repetitive.”

This makes more sense; this additional analysis with double repeat masking is useful. We will be very surprised if the *vast*, *vast* majority of this sequence is not repetitive in origin.

Response: Thank you. We are glad that we agree on the way this is now stated. Done.

“Further analyses of these candidate genes led us to *sall1*, an important gene for the distal branching of the limb and involved in human limb pathologies (Townes_Brocks syndrome).”

This is a great new insight. The inclusion of additional novel biological insights improves the quality of the manuscript.

Response: Thank you your praise. We very much appreciate that you like the additional biological insights we’ve incorporated into the revised manuscript.

“To have a closer look on the remaining fraction of the genome our paper provides the novel approach of double masking as a way to assess the repetitiveness in the remainder 13Gb. In fact, we found that indeed a significant proportion of this initially unmasked DNA has repeat signatures. In total, we estimated that the lungfish genome is up to 90% repetitive. In fact, we found that a substantial proportion of this DNA is repetitive, representing also many previously unrecoverable LINES and unknown elements ... To be more conservative, we provided “at least 2/3” estimate in the text of the original submission. We rephrased the text to be more clear and have clarified the figure legend to explain that it refers to retroelements (LINE/LTR) and DNA transposons (DNA) as whole classes of repeat elements”

In the discussion/results I would first report whatever you think is the most accurate value for the proportion of repeats, and elaborate on the 2-step masking afterwards. The fact that the first run gives you 67% repeat content seems more like an artefact or inadequacy of the repeat thresholding/modeling than it is a feature of the genome that you are publishing on.

Response: We have changed the discussion/results as suggested and now report first the most accurate value for the proportion of repeats, which comprises 90% of the total assembly size.

“These giant genome animals have proportionally larger cells and longer cell cycles as well, so the availability of components for DNA replication and 18 mitosis can likely accumulate through controlling gene expression.”

This is very interesting, and perhaps merits its own investigation, but for this publication it may at least be worth citing a study regarding cell size/cycles.

Response: We have now included this idea in the conclusions paragraph:

“We did not find in the genome signatures that would suggest that the lungfish has adapted to having a huge genome, e.g. with respect to DNA replication. An explanation may be that in organisms with very large genomes DNA replication may be regulated by other mechanism, e.g. by increasing the number of replication origins. Of note, giant genome animals have proportionally larger cells and longer cell cycles (for instance in somatic cells of the axolotl and in the embryo 6 hours versus 30 minutes in Xenopus). Thus, the availability of components for DNA replication and mitosis can likely accumulate through controlling gene expression.”

We now added three reference to that effect:

Rost F, Rodrigo Albors A, Mazurov V, et al. Accelerated cell divisions drive the outgrowth of the regenerating spinal cord in axolotls. *Elife*. 2016;5:e20357.

doi:10.7554/eLife.20357.

Hyrien O., Maric C., Mechali M. Transition in specification of embryonic metazoan DNA replication origins. *Science*. 1995;270:994–997. doi: 10.1126/science.270.5238.994.

Pagel, M. and R.F. Johnstone. 1992. Variation across species in the size of the nuclear genome supports the junk-DNA explanation for the C-value paradox. *Proc. Roy. Soc.* 249: 119-124.

Reviewer Reports on the Second Revision:

Referee #1 (Remarks to the Author):

The authors have adequately addressed my concerns about the study. This is an interesting study.

Final small comment: Figure 1 species names are not italicized and the font spacing varies among species.

Referee #3 (Remarks to the Author):

We thank the authors for addressing our comments in detail. Overall, we are mostly satisfied with the response, but request some additional analysis:

* The authors did not address our request to run SDA (Segmental Duplication Assembler) to identify the regions of collapse in the assembly. While the explanation of repeat processing in MARVEL is helpful, and describes a method that would perhaps prevent this issue, it has not yet been validated. SDA would help to support the claims that repeat collapse is not an issue using the MARVEL assembler by reevaluating coverage in the completed assembly. It is a 2 part pipeline that would also enable resolving repeats using an orthogonal method and comparing those results to your own.

* In addition to this, we recommend running winnowmap and manually reviewing the coverage in terms of deviations from the mean, and as a function of repetitiveness of each region.

* Regarding the comparison to other assemblers, we feel that MARVEL still does not have a thorough validation of the number of misassemblies that its assemblies may contain. However, the analysis that the misassemblies reported by QUILT for drosophila are largely places of transposon insertions is useful. We agree about the explanation of the behavior of the transposons with the Shasta assembler. Were there any additional metrics/figures from the drosophila genome analysis that can be shared with us to alleviate any remaining concerns?

* We appreciate the explanation of the lack of ability to do Bionano sequencing, and the comparison to the axolotl Bionano sequencing analysis is helpful.

* We would suggest including the HiC misjoin figure in the supplement, as mentioned in the point-by-point response.

The HiC map now included as Supp. Fig 25 has some unusual behavior with the contacts between the two ends of the scaffolds for the four largest scaffolds, at the corners of the boxes off the diagonal. Is there an explanation for this?

Author Rebuttals to Second Revision:**Responses to reviewer 3**

Referee #3 (Remarks to the Author):

We thank the authors for addressing our comments in detail. Overall, we are mostly satisfied with the response, but request some additional analysis:

* The authors did not address our request to run SDA (Segmental Duplication Assembler) to identify the regions of collapse in the assembly. While the explanation of repeat processing in MARVEL is helpful, and describes a method that would perhaps prevent this issue, it has not yet been validated. SDA would help to support the claims that repeat collapse is not an issue using the MARVEL assembler by reevaluating coverage in the completed assembly. It is a 2 part pipeline that would also enable resolving repeats using an orthogonal method and comparing those results to your own.

We apologize for the oversight in addressing the SDA suggestion in the last round. We have tried to implement SDA on the lungfish genome, including 4 adaptations of the software (in consultation with the developer via Github) to keep SDA from aborting on this large genome. Nonetheless, the algorithm has never finished or output any desired results. It apparently is not written in a way that is usable on a giant genome. In addition, we were in direct communication with the developer who reiterated that >40x coverage is required to obtain meaningful results (see Vollger et al Nat Methods, 2019, 16:88-94) and therefore questioned the validity of running this software in this context. We therefore implemented a similar approach to identify regions of unusually high coverage as suggested by the reviewer. In brief, we mapped the ONT reads to the genome using winnowmap2 and identified regions of the lungfish genome where the coverage exceeds 3 standard deviations from the mean (Figure R1).

Figure R1. Read coverage along the assembly showing a portion of scaffold 1. Red lines mark regions showing >3 SD from mean. Overall, these regions represent 0.09% of the genome.

There are 2248 such regions comprising about 34,405,000 bp (0.099% of the combined scaffolds length). Table R1 below describes the proportion of those regions that contain no repeats. In summary, only 9.3% out of the 0.099% portion of the scaffolds that show high coverage is NOT associated with repeats meaning that only 0.0092% of the assembly might be subject to the caveats of concern.

	Round1	Round1+2
Regions with 0 repeats	319 (14.2%)	208 (9.3%)
Total length of those regions	2,860,000 bp	1,940,000 bp
Mean length of the region	8.9kb	9.3kb

Table R1. Repeat association of the 2248 regions that show >3 standard deviations in ONT read mapping coverage. These 2248 regions covering 34,405,000 bp represent 0.099% of the scaffolds.

Figure R2 below shows a typical region of read pile-up. This region is entirely covered by one annotated repeat.

Figure R2. Representative region showing read pile up. The entire region is covered by a repeat sequence (red).

* In addition to this, we recommend running winnowmap and manually reviewing the coverage in terms of deviations from the mean, and as a function of repetitiveness of each region.

See comment above.

* Regarding the comparison to other assemblers, we feel that MARVEL still does not have a thorough validation of the number of misassemblies that its assemblies may contain. However, the analysis that the misassemblies reported by QUAST for drosophila are largely places of transposon insertions is useful. We agree about the explanation of the behavior of the transposons with the Shasta assembler. Were there any additional metrics/figures from the drosophila genome analysis that can be shared with us to alleviate any remaining concerns?

Here we provide additional metrics on the drosophila cell line genome assembly, including the BUSCO scores of the SHASTA and MARVEL generated assemblies and alignment to the drosophila reference.

Drosophila
BUSCO:
Shasta

C:97.8% [S:97.4%,D:0.4%],F:0.8%,M:1.4%,n:3285
3211 Complete BUSCOs (C)
3198 Complete and single-copy BUSCOs (S)
13 Complete and duplicated BUSCOs (D)
25 Fragmented BUSCOs (F)
49 Missing BUSCOs (M)
3285 Total BUSCO groups searched

MARVEL (polished)

C:96.2% [S:95.7%,D:0.5%],F:1.7%,M:2.1%,
n:3285
3160 Complete BUSCOs (C)
3144 Complete and single-copy BUSCOs (S)
16 Complete and duplicated BUSCOs (D)
56 Fragmented BUSCOs (F)
69 Missing BUSCOs (M)
3285 Total BUSCO groups searched

Figure R3. Alignment of drosophila cell line genomic DNA assembled with various assemblers and aligned to the reference. Figures at bottom depict the Flamenco piRNA cluster showing the distribution of reads.

* We appreciate the explanation of the lack of ability to do Bionano sequencing, and the comparison to the axolotl Bionano sequencing analysis is helpful.

Thank you

* We would suggest including the HiC misjoin figure in the supplement, as mentioned in the point-by-point response.

This figure is now part of a new Extended Data Figure 1e.

The HiC map now included as Suppl. Fig 25 has some unusual behavior with the contacts between the two ends of the scaffolds for the four largest scaffolds, at the corners of the boxes off the diagonal. Is there an explanation for this?

The previous Suppl. Fig. 25 is now Extended Data Fig. 1d. These are telomeric or subtelomeric regions that contact the other arm. Such contacts occur both in meiosis and mitosis (10.1104/pp.111.187161, doi.org/10.1111/j.0022-2720.2004.01324.x, 10.1104/pp.125.2.532, 10.7554/eLife.23623 Fig1, 10.1038/s41598-019-42967-4)

Reviewer Reports on the Third Revision:

Referee #3 (Remarks to the Author):

Thank you for your informative analysis on collapsed regions using coverage depth. We are now happy with the paper as is, but we have a few small outstanding questions and suggestions. These don't have to be addressed, in our view, for the paper to be accepted, but we pose them for sake of interest / clarification.

(1) As a quick clarification, for the collapsed regions, was this coverage computed without secondary alignments? Ideally only primaries and secondaries would be included to prevent artificial spikes in coverage from duplicated read segments. If not, it would be simple to filter the BAM to produce the desired output.

Your table is very helpful in understanding the state of the assembly in terms of collapses and we would suggest, in the spirit of full disclosure, that it is included in the supplement and referenced in the text. However, "round1" and "round1+2" column headers are not explained. These are the successive repeat masking rounds?

You wrote: "In summary, only 9.3% out of the 0.099% portion of the scaffolds that show high coverage is NOT associated with repeats meaning that only 0.0092% of the assembly might be subject to the caveats of concern". We believe regardless of the cause of collapse that this information is important to evaluating the assembly, and that the table should include statistics for repeat and non-repeat regions that exceed your coverage threshold. This repeat collapse analysis should appear in the paper somewhere (at least in the supplement).

The example repeat region plot would also be informative as a figure in the supplement (accompanying the table), especially with the addition of the mean and SD coverage as elements in the plot or the figure legend.

Can you clarify what "marvel-purged" means in Figure R3? (as opposed to "MARVEL (polished) mentioned earlier?) Do these methods apply to the lungfish assembly?

Author Rebuttals to Third Revision:

In the final manuscript we have addressed all remaining issues of reviewer 3 regarding the assembly. We have added an additional Ext. Data Figure 10, and an additional Supplementary Table 19, and more information about this aspect of the paper in the Supplementary Information especially regarding the assembly benchmarking. We added a new section in the Supplementary

Information. Assembly validation: to gauge repeat and segmental collapse we implemented an approach to SDA to identify regions of unusually high coverage. In brief, we mapped the ONT reads to the assembly using winnowmap2 and identified regions where the coverage exceeds 3 standard deviations from the mean (Extended Figure 10). We detected 2248 such regions comprising about 34,405,000 bp (0.099% of the combined scaffolds length). Only 9.3% out of the 0.099% portion of the scaffolds that show high coverage are not associated with annotated repeats, meaning that only 0.0092% of the assembly might be subject to the caveats of repeat and segmental collapse (Supplementary Table 19).

Furthermore, we validated the assembly and scaffolding in comparison to the chromosome-scale meiotic scaffolding from ref.43 and is available as described in ref.44.